# The Collusion of Memory and Nonlinearity in Stochastic Approximation With Constant Stepsize

**Dongyan Lucy Huo**[1]    **Yixuan Zhang**[2]    **Yudong Chen**[2]    **Qiaomin Xie**[2]

[1]Cornell University    [2]University of Wisconsin-Madison

dh622@cornell.edu

{yzhang2554,yudong.chen,qiaomin.xie}@wisc.edu

## Abstract

In this work, we investigate stochastic approximation (SA) with Markovian data and nonlinear updates under constant stepsize $\alpha > 0$. Existing work has primarily focused on either i.i.d. data or linear update rules. We take a new perspective and carefully examine the simultaneous presence of Markovian dependency of data and nonlinear update rules, delineating how the interplay between these two structures leads to complications that are not captured by prior techniques. By leveraging the smoothness and recurrence properties of the SA updates, we develop a fine-grained analysis of the correlation between the SA iterates $\theta_k$ and Markovian data $x_k$. This enables us to overcome the obstacles in existing analysis and establish for the first time the weak convergence of the joint process $(x_k, \theta_k)_{k \geq 0}$. Furthermore, we present a precise characterization of the asymptotic bias of the SA iterates, given by $\mathbb{E}[\theta_\infty] - \theta^* = \alpha(b_\mathrm{m} + b_\mathrm{n} + b_\mathrm{c}) + \mathcal{O}(\alpha^{3/2})$. Here, $b_\mathrm{m}$ is associated with the Markovian noise, $b_\mathrm{n}$ is tied to the nonlinearity of the SA operator, and notably, $b_\mathrm{c}$ represents a multiplicative interaction between the Markovian noise and the nonlinearity of the operator, which is absent in previous works. As a by-product of our analysis, we derive finite-time bounds on higher moment $\mathbb{E}[\|\theta_k - \theta^*\|^{2p}]$ and present non-asymptotic geometric convergence rates for the iterates, along with a Central Limit Theorem.

## 1   Introduction

Stochastic Approximation (SA) is an iterative scheme for solving fixed-point equations using noisy observations. Its application spans various domains including stochastic control [9, 40], reinforcement learning (RL) [4, 62] and stochastic optimization [43]. A typical SA algorithm takes the form $\theta_{k+1} = \theta_k + \alpha g(\theta_k, x_k)$, where $(x_k)_{k \geq 0}$ represents the underlying noisy data sequence and $\alpha > 0$ is a constant stepsize. The goal of SA is to approximate the target solution $\theta^*$ that solves $\mathbb{E}_{x \sim \pi}[g(\theta^*, x)] = 0$, with $\pi$ being the stationary distribution of the stochastic process $(x_k)_{k \geq 0}$.

SA subsumes many important algorithms. A prime example is stochastic gradient descent (SGD) for minimizing a function $J(\theta)$ given noisy estimates $g(\theta, x)$ of its gradient. Linear SA schemes include SGD for quadratic objective functions, as well as various RL algorithms such as linear TD-Learning (in which $g$ is not the gradient of any function and standard SGD results do not apply).

Of particular interest to us are SA updates given by a *nonlinear* function $g(\theta, x)$ of $\theta$. One motivating example is learning a Generalized Linear Model (GLM) $y \approx \sigma(z^\top \theta)$ with a nonlinear mean function $\sigma : \mathbb{R} \to \mathbb{R}$. A power approach, developed in [19, 36, 37, 64], uses a surrogate loss function, where the corresponding SGD update takes the form $\theta_{k+1} = \theta_k + \alpha(\sigma(w_k^\top \theta_k) - y_k)w_k$, where $x_k = (w_k, y_k)$ is the observed covariate-response pair. Common choices of $\sigma$ include the identity map for linear regression, the sigmoid function for logistic regression, as well as Rectified Linear Unit (ReLU) and its various smoothed versions (e.g., ELU and SoftPlus) for ReLU regression [7, 18, 19, 30, 36].

38th Conference on Neural Information Processing Systems (NeurIPS 2024).

Furthermore, we are interested in the setting where the data sequence $(x_k)_{k \geq 0}$ forms a *Markov chain*, going beyond the common i.i.d. data setting. The Markovian model captures a wide range of SA problems in machine learning where stochastic data exhibit serial dependence [5, 33, 38, 53].

Classical work on SA focuses on diminishing stepsizes [10, 58]. Constant stepsize schemes have recently gained popularity due to easy parameter tuning, fast initial convergence, and robust empirical performance. Non-asymptotic error bounds have been obtained for constant stepsize SA [17, 60]. Recent work further provides fine-grained characterization of the distributional and steady-state behaviors of the iterates [20, 33, 45, 67, 69]. Two recurring themes in these results are weak convergence of the distribution of $\theta_t$ and the presence of an asymptotic bias $\mathbb{E}[\theta_\infty] - \theta^* \propto \alpha$, both having important implications for iterate averaging, bias reduction and statistical inference [34].

Note that most previous work studied the nonlinear update setting and Markovian data setting *separately*—e.g., in [20, 67] for nonlinear SGD with i.i.d. data, and in [33, 34] for Markovian linear SA. The linearity or i.i.d. assumptions imposed in these prior works are restrictive, especially in the face of modern machine/reinforcement learning paradigms where nonlinear models are the norm and dependent data is common. Moreover, the absence of prior work dealing with Markovian nonlinear SA is not merely an overlook—as argued below, this setting is significantly more challenging.

**Our Contributions**  In this work, we study constant-stepsize SA with *both* Markovian data and nonlinear update. In Section 3, we elucidate the new challenges that arise from the simultaneous presence of these two structures, which break key steps in previous analyses of the i.i.d. or linear setting. Due to the interaction between these two structures, establishing weak convergence is far from obvious, and the asymptotic bias exhibits new behaviors. Consequently, analyzing the nonlinear Markovian setting requires more than simply combining previous techniques.

To address the above confounding complication, we exploit the smoothness and recurrence structures of the SA update, thereby developing a fine-grained analysis of the correlation of the parameter $\theta_k$ and data $x_k$. This allows us to establish for the first time the weak convergence of the joint process $(x_k, \theta_k)_{k \geq 0}$ to a unique invariant distribution, represented by the limiting random variable $(x_\infty, \theta_\infty)$. As a by-product of our analysis, we derive finite-time bounds on $\mathbb{E}[\|\theta_k - \theta^*\|^{2p}]$, the $2p$-th moments of the errors, generalizing the results in [17, 20, 60] to higher moments and to the nonlinear Markovian setting. In addition, we prove a Central Limit Theorem (CLT) for averaged iterates.

Moreover, we show that nonlinearity and Markovian structure contribute in a *multiplicative* way to the asymptotic bias of the SA iterates. We obtain the following bias characterization: $\mathbb{E}[\theta_\infty^{(\alpha)}] - \theta^* = \alpha(b_{\mathrm{m}} + b_{\mathrm{n}} + b_{\mathrm{c}}) + \mathcal{O}((\alpha\tau_\alpha)^{3/2})$. We provide explicit expressions for the vectors $b_{\mathrm{m}}, b_{\mathrm{n}}, b_{\mathrm{c}}$, which are independent of $\alpha$. $b_{\mathrm{m}}$ represents the bias component due to Markovian data (quantified by the mixing property of $x_k$), and $b_{\mathrm{n}}$ the bias due to the nonlinearity of $g$ (quantified by the second derivative $g''$). Importantly, we identify the compound term $b_{\mathrm{c}}$, which is absent in both nonlinear SA with i.i.d. data and linear SA with Markovian data. We explore the algorithmic implications of the above results on Polyak-Ruppert (PR) averaging [35, 56, 59] and Richardson-Romberg (RR) extrapolation [31]. We show that PR averaging reduces the variance but not the bias, whereas RR extrapolation eliminates the leading bias term $\alpha(b_{\mathrm{m}} + b_{\mathrm{n}} + b_{\mathrm{c}})$, reducing the asymptotic bias to a higher order of $\alpha$.

**Related Work**  Postponing a detailed literature review to Section 6, here we remark on the recent works most relevant to ours [1, 44, 45, 46], all studying Markovian nonlinear SA. The authors of [1] present an *upper bound* for the PR-averaged iterates and, similar to our results, demonstrate the effectiveness of RR-extrapolation in reducing bias, but lack weak convergence result for last iterates. In [44], the authors suggest adopting the ordinary differential equation framework to prove weak convergence of iterates and derive an *upper bound* for the asymptotic bias, which contrasts with our equality characterization with a closed-form solution for the leading-order bias. In [45], the authors prove weak convergence of $(x_t, \theta_t)$ using coupling, but only in the linear setting, *not* for nonlinear SA. In the latter setting, their weak convergence analysis is thwarted by challenges similar to what we elucidate in Section 3, the interplay between nonlinearity and Markovian data leading to "double recursions." The coupling technique differs as well: we couple two processes by sharing data $x_t = x'_t$, while in [45], they initialize two processes with different $x_0$ and $x'_0$, and analyze the stopping time $\tau$ when $\theta_\tau = \theta'_\tau$. Moreover, they only present an *upper bound* for asymptotic bias, while ours presents a fine-grained characterization in Theorem 4.6 *necessary* for justifying RR-extrapolation. Lastly, the paper [46] discusses stepsize selection and its impact on the asymptotic statistics of PR-averaged SA with both constant and diminishing stepsizes.

**Notations** The Euclidean norm is denoted by $\|\cdot\|$. The notation "$u \otimes v$" represents the tensor product between vectors $u$ and $v$, and "$u^{\otimes k}$" denotes the $k$-th tensor power of vector $u$. The ball with radius $\beta$ is $B(\beta) := \{\theta \in \mathbb{R}^d : \|\theta\| \leq \beta\}$. $\mathcal{L}(z)$ denotes the distribution of a random vector $z$ and $\mathrm{Var}(z)$ its covariance matrix. Let $\mathcal{P}_2(\mathbb{R}^d)$ be the space of square-integrable distributions on $\mathbb{R}^d$ and $\mathcal{P}_2(\mathcal{X} \times \mathbb{R}^d)$ be the space of distributions $\bar{\nu}$ on $\mathcal{X} \times \mathbb{R}^d$ with square-integrable second marginal on $\mathbb{R}^d$. The Wasserstein-2 between two probability measures $\mu$ and $\nu$ in $\mathcal{P}_2(\mathbb{R}^d)$ is defined as $W_2(\mu, \nu) = \inf_{\psi \in \Pi(\mu, \nu)} \left\{ \left(\mathbb{E}[\|\theta - \theta'\|^2]\right)^{\frac{1}{2}} : \mathcal{L}(\theta) = \mu, \mathcal{L}(\theta') = \nu \right\}$, where $\Pi(\mu, \nu)$ is the set of all couplings between $\mu$ and $\nu$. Extending to $\mathcal{X} \times \mathbb{R}^d$, we define the metric $\bar{d}\big((x, \theta), (x', \theta')\big) := \sqrt{\mathbb{1}\{x \neq x'\} + \|\theta - \theta'\|^2}$, and denote by $\bar{W}_2$ the extended Wasserstein-2 distance w.r.t. $\bar{d}$.

The lowercase letter $c$ and its derivatives $c'$, $c_0$, etc. denote universal numerical constants, whose value may change from line to line. We use $s \equiv s(\theta_0, \theta^*, \mu, L, R)$ and its derivatives to denote quantities (scalars, vectors, or matrices) that are independent of the stepsize $\alpha$ and the iteration index $k$, but may depend on the initialization $\theta_0$, SA primitives $\theta^*$, $\mu$ and $L$, and the coefficient $R$ for the geometric mixing rate of $(x_k)$ in Assumption 1. As we are primarily interested in dependence on $\alpha$ and $k$, we adopt the following big-O notation: $\|f\| = \mathcal{O}(h(\alpha, k))$ if it holds that $\|f\| \leq s \cdot \|h(\alpha, k)\|$.

## 2 Problem Setup and Preliminaries

Let $(x_k)_{k \geq 0}$ be a Markov chain on a general state space $\mathcal{X}$. Consider the following projected stochastic approximation (SA) iteration:

$$\theta_{k+1}^{(\alpha)} = \Pi_{B(\beta)} \left[ \theta_k^{(\alpha)} + \alpha \big( g(\theta_k^{(\alpha)}, x_k) + \xi_{k+1}(\theta_k^{(\alpha)}) \big) \right], \qquad (2.1)$$

where $g : \mathbb{R}^d \times \mathcal{X} \to \mathbb{R}^d$ is a deterministic function, $\{\xi_k\}_{k \geq 1}$ are i.i.d. zero-mean random fields, $\alpha > 0$ is a constant stepsize, and $\Pi_{B(\beta)}(\theta) := \arg\min_{z : \|z\| \leq \beta} \|z - \theta\|$ is the projection operator. We shall omit the superscript $^{(\alpha)}$ in $\theta_k$ when the dependence on $\alpha$ is clear from the context. In this work, we also consider the projection-free variant of the iteration (2.1) with $\beta = \infty$.

We denote by $\pi$ the stationary distribution of the Markov chain $(x_k)_{k \geq 1}$ and define the shorthand $\bar{g}(\theta) := \mathbb{E}_\pi[g(\theta, x)]$, where $\mathbb{E}_\pi[\cdot]$ denotes the expectation with respect to $x \sim \pi$. The algorithm (2.1) computes an estimation of the target vector $\theta^*$ that solves the steady-state equation $\mathbb{E}_\pi[g(\theta, x)] = 0$. Our general goal is to characterize the relationship between the iterate $\theta_k$ and the target solution $\theta^*$.

In the following, we state the assumptions needed for our main results. For a more detailed discussion of the assumptions, we refer readers to Appendix B.

**Assumption 1** (Uniform Ergodicity). *$(x_k)_{k \geq 0}$ is a uniformly ergodic Markov chain on a countable state space $(\mathcal{X}, \mathcal{B}(\mathcal{X}))$ with transition kernel $P$ and a unique stationary distribution $\pi$. That is, there exist constants $r \in [0, 1)$ and $R > 0$ such that $\|P^k(x, \cdot) - \pi\|_{\mathrm{TV}} \leq Rr^k, \forall x \in \mathcal{X}$.*

The countable state space ensures separability under the $\mathbb{1}\{x \neq x'\}$ metric, necessary for constructing a valid coupling in the invariance proof and establishing a well-defined $P^*$ for bias characterization. We keep the notation general to allow future extensions to general state space and broader applicability of our results. All irreducible, aperiodic, and finite state space Markov chains are uniformly ergodic. The uniform ergodicity assumption is common in prior work on SA with Markovian noise [6, 21, 24, 33, 48]. Relaxing this uniform ergodicity assumption, in the style of [45, 52, 60] is possible but orthogonal to our focus, and thus we do not pursue this direction in this work.

We allow the chain $(x_k)_{k \geq 0}$ to be arbitrarily initialized rather than from the stationary distribution $\pi$. An important quantity is the mixing time of the Markov chain, defined as follows.

**Definition 2.1.** *For $\epsilon \in (0, 1)$, the $\epsilon$-mixing time of $(x_k)_{k \geq 0}$, denoted by $\tau_\epsilon \geq 1$, is defined as $\tau_\epsilon := \min\left\{ k \geq 1 : \sup_{x \in X} \|P^k(x, \cdot) - \pi\|_{\mathrm{TV}} \leq \epsilon \right\}$.*

Under Assumption 1, the $\epsilon$-mixing time satisfies $\tau_\epsilon \leq K \log \frac{1}{\epsilon}$ for all $\epsilon \in (0, 1)$, where $K \geq 1$ is independent of $\epsilon$. In the sequel, unless otherwise specified, we always choose $\epsilon = \alpha$ and let $\tau \equiv \tau_\alpha$.

The following assumptions on the nonlinear function $g$ in (2.1) is standard in the literature [17, 20, 32, 45, 48]. A wide family of $g$ functions satisfies these assumptions, with the $L_2$-regularized logistic regression of GLM being a standard example.

**Assumption 2** (Differentiability and Linear Growth). *For each $x \in \mathcal{X}$, the function $g(\theta, x)$ is three times continuously differentiable in $\theta$ with uniformly bounded first to third derivatives, i.e., $\sup_{\theta \in \mathbb{R}^d} \|g^{(i)}(\theta, x)\| < +\infty$ for $i = 1, 2, 3$, $x \in \mathcal{X}$. Moreover, there exists a constant $L_1 > 0$ such that (1)$\|g^{(i)}(\theta, x) - g^{(i)}(\theta', x)\| \leq L_1$, for all $\theta, \theta' \in \mathbb{R}^d$, $i = 0, 1, 2$ and $x \in \mathcal{X}$, and (2) $\|g(0, x)\| \leq L_1$ for all $x \in \mathcal{X}$.*

The linear growth condition in Assumption 2 implies that $g(\theta, x)$ is $L_1$-Lipschitz w.r.t. $\theta$ uniformly in $x$. When $g$ is a linear function, i.e., $g(\theta, x) = A(x)\theta + b(x)$, this assumption is satisfied with $\sup_{x \in \mathcal{X}} \|A(x)\| < \infty$ and $\sup_{x \in \mathcal{X}} \|b(x)\| < \infty$, which are commonly assumed for linear SA. The above assumption immediately implies that the growth rate of $\|g\|$ and $\|\bar{g}\|$ will be at most linear in $\theta$, i.e., $\|g(\theta, x)\| \leq L_1(\|\theta - \theta^*\| + 1)$ and $\|\bar{g}(\theta)\| \leq L_1(\|\theta - \theta^*\| + 1)$.

**Assumption 3** (Strong Monotonicity). *There exists $\mu > 0$ such that $\langle \theta - \theta', \bar{g}(\theta) - \bar{g}(\theta') \rangle \leq -\mu\|\theta - \theta'\|^2$, $\forall \theta, \theta' \in \mathbb{R}^d$. Consequently, the target equation $\bar{g}(\theta) = 0$ has a unique solution $\theta^*$.*

When $g$ is a gradient field, Assumption 3 is equivalent to strong convexity. For notational simplicity, we assume the strong monotonicity parameter satisfies $\mu \leq 1 - r$, where $r$ is the convergence factor in Assumption 1. For general $\mu$, our results remain valid with $\mu$ replaced by $\min\{\mu, 1 - r\}$.

We next consider the noise. Denote by $\mathcal{F}_k$ the filtration generated by $\{x_t, \theta_t, \xi_{t+1}\}_{t=0}^{k-1} \cup \{x_k, \theta_k\}$.

**Assumption 4** (Noise Sequence). *Let $p \in \mathbb{Z}_+$ be given. The noise sequence $(\xi_k)_{k \geq 1}$ is a collection of i.i.d. random fields satisfying the following conditions with $L_{2,p} > 0$:*

$$\mathbb{E}[\xi_{k+1}(\theta)|\mathcal{F}_k] = 0 \quad and \quad \mathbb{E}^{1/(2p)}[\|\xi_1(\theta)\|^{2p}] \leq L_{2,p}(\|\theta - \theta^*\| + 1), \quad \forall \theta \in \mathbb{R}^d. \tag{2.2}$$

*Define $C(\theta) = \mathbb{E}[\xi_1(\theta)^{\otimes 2}]$ and assume that $C(\theta)$ is at least twice differentiable. There also exist $M_\epsilon, k_\epsilon \geq 0$ such that for $\theta \in \mathbb{R}^d$, we have $\max_{i=1,2} \|C^{(i)}(\theta)\| \leq M_\epsilon \{1 + \|\theta - \theta^*\|^{k_\epsilon}\}$.*

In the sequel, we set $L := L_1 + L_2$, and without loss of generality, we assume $L \geq 1$.

When $p = 1$, the second inequality in (2.2) only requires linear growth *in expectation*, which relaxes the almost sure linear growth condition in [17]. The constraint on the covariance matrix $C(\theta)$ is lenient and satisfied in most regular enough settings, as shown in [20].

## 3   Analytical Challenges and Techniques

In this section, we elaborate on the challenges and techniques in proving the above results.

Previous work has established weak convergence of $(x_k, \theta_k)$ separately for nonlinear SA with i.i.d. data, and for Markovian linear SA. The high-level approaches used in two representative prior works can be summarized as follows. The work [20] on nonlinear SGD leverages *local linearization* of $g$ through Taylor expansion. The work [33] on Markovian linear SA exploits the mixing property of the Markovian noise to regain approximate independence, particularly between $x_k$ and $\theta_{k-\tau}$ for sufficiently large $\tau$. It is tempting to expect that nonlinear SA can be analyzed by combining these two approaches. Perhaps surprisingly, such a simple combination would not work due to the interplay between nonlinearity and Markovian structures.

To demonstrate this challenge, let us seek to establish weak convergence in the Wasserstein distance $W_2$ via forward coupling [29], an approach employed by both [20, 33] as well as others [25]. Specifically, we consider two SA iterate sequences $(\theta_k^{[1]})_{k \geq 0}$ and $(\theta_k^{[2]})_{k \geq 0}$ from different initializations $\theta_0^{[1]}$ and $\theta_0^{[2]}$ coupled by sharing the data sequence $(x_k)_{k \geq 0}$: $\theta_{k+1}^{[1]} = \theta_k^{[1]} + \alpha g(\theta_k^{[1]}, x_k)$ and $\theta_{k+1}^{[2]} = \theta_k^{[2]} + \alpha g(\theta_k^{[2]}, x_k)$. To establish convergence in $W_2$, we consider the difference sequence

$$w_{k+1} := \theta_{k+1}^{[1]} - \theta_{k+1}^{[2]} = w_k + \alpha\big(g(\theta_k^{[1]}, x_k) - g(\theta_k^{[2]}, x_k)\big), \tag{3.1}$$

and it suffices to prove $w_k$ converges to 0 in mean square: $\mathbb{E}[\|w_{k+1}\|^2] \lesssim \rho^k \mathbb{E}[\|w_0\|^2]$ for $\rho < 1$.

With this goal in mind and following the idea from [20], one may first linearize the right-hand side of the difference dynamic (3.1) and obtain the approximation

$$w_{k+1} \approx w_k + \alpha g'(\theta_k^{[2]}, x_k)w_k. \tag{3.2}$$

Next, to analyze the drift of the Lyapunov function $\mathbb{E}[\|w_k\|^2]$ and handle the Markovian noise $(x_k)$, we use the conditioning technique from [33]. We condition on the information of $\tau$ steps before, denoted by $\mathcal{F}_{k-\tau} := \sigma\big((\theta_t^{[1]}, \theta_t^{[2]}, x_t) : t \leq k - \tau\big)$. Ignoring higher-order terms and assuming a one-dimensional problem for simplicity, we obtain that

$$\mathbb{E}[\|w_{k+1}\|^2] \approx \mathbb{E}\Big[\mathbb{E}\big[\|w_k\|^2\big(1 + 2\alpha g'(\theta_k^{[2]}, x_k)\big) \mid \mathcal{F}_{k-\tau}\big]\Big]$$

$$\approx \mathbb{E}\Big[\|w_{k-\tau}\|^2\big(1 + 2\alpha\mathbb{E}\big[g'(\theta_k^{[2]}, x_k) \mid \mathcal{F}_{k-\tau}\big]\big)\Big], \qquad (3.3)$$

where we use $w_k \approx w_{k-\tau}$ for small $\alpha$ (this argument, which is made precise in [33, 60], essentially exploits the fact that $x_k$ evolves faster than $\theta_k$).

To prove dynamic (3.3) converges, it boils down to showing the "gain matrix" $\mathbb{E}\big[g'(\theta_k^{[2]}, x_k) \mid \mathcal{F}_{k-\tau}\big]$ is negative/Hurwitz. To further simplify, we assume $k$ is large so that the chain $(x_k)$ is distributed per its stationary distribution $\pi$, in which case the gain matrix simplifies to $\mathbb{E}_{x_\infty \sim \pi}[g'(\theta_\infty^{[2]}, x_\infty)]$.

Analyzing this gain matrix is where our analysis diverges from previous work. If the SA update were *linear*, i.e., $g(\theta, x) = A(x)\theta$, then the gain $\mathbb{E}[g'(\theta_\infty^{[2]}, x_\infty)] = \mathbb{E}_\pi[A(x_\infty)]$ would be independent of $\theta_\infty^{[2]}$, and its Hurwitz property is a standard and necessary condition for proving convergence of linear SA. If the data sequence $(x_k)$ were *i.i.d.*, then $\theta_k$ would be *independent* of $x_k$ and hence the gain becomes $\mathbb{E}[g'(\theta_\infty^{[2]}, x_\infty)] = \mathbb{E}[\mathbb{E}[g'(\theta_\infty^{[2]}, x_\infty)|\theta_\infty^{[2]}]] = \mathbb{E}[\bar{g}'(\theta_\infty^{[2]})]$ with $\bar{g}(\cdot) := \mathbb{E}_{x \sim \pi}[g(\cdot, x)]$, where the Hurwitz property again follows from standard assumptions on $\bar{g}$.

However, both arguments fail for the *Markovian nonlinear* setting. Common assumptions for nonlinear SA only ensure Hurwitz $\mathbb{E}_{x \sim \pi}[g'(\theta, x)|\theta]$ given $\theta$. This does not imply the desired Hurwitz $\mathbb{E}[g'(\theta_\infty^{[2]}, x_\infty)]$, precisely owing to the simultaneous presence of (i) the *dependence* of $g'$ on both $\theta_\infty$ and $x_\infty$ (due to nonlinearity) and (ii) the *correlation* between $\theta_\infty$ and $x_\infty$ (due to Markovian).

**Our Approach** We overcome this challenge by carefully analyzing the properties of the above dependence and correlation. Therefore, for sufficiently large $\tau$, we further decompose (3.3) as

$$\mathbb{E}[\|w_{k+1}\|^2] \approx \mathbb{E}\Big[\|w_{k-\tau}\|^2\big(1 + 2\alpha\mathbb{E}\big[g'(\theta_k^{[2]}, x_k) \mid \mathcal{F}_{k-\tau}\big]\big)\Big]$$

$$= \mathbb{E}\Big[\|w_{k-\tau}\|^2\Big(1 + 2\alpha\underbrace{\mathbb{E}\big[g'(\theta_{k-\tau}^{[2]}, x_k) \mid \mathcal{F}_{k-\tau}\big]}_{\approx \mathbb{E}[g'(\theta_{k-\tau}^{[2]}, x_\infty)|\mathcal{F}_{k-\tau}] \text{ Hurwitz}} + 2\alpha\big(\mathbb{E}\big[g'(\theta_k^{[2]}, x_k) - g'(\theta_{k-\tau}^{[2]}, x_k) \mid \mathcal{F}_{k-\tau}\big]\big)\Big)\Big]$$

$$\lesssim \rho\mathbb{E}[\|w_{k-\tau}\|^2] + \alpha\mathbb{E}\Big[\mathbb{E}\big[\underbrace{\langle w_{k-\tau}, g(\theta_k^{[1]}, x_k) - g(\theta_k^{[2]}, x_k) - g(\theta_{k-\tau}^{[1]}, x_k) + g(\theta_{k-\tau}^{[2]}, x_k)\rangle}_{\spadesuit} \mid \mathcal{F}_{k-\tau}\big]\Big],$$

where we approximate $w_k \approx w_{k-\tau}$, $w_t g'(\theta_t^{[2]}, x_k) \approx (g(\theta_k^{[1]}, x_k) - g(\theta_k^{[2]}, x_k))$ for $t = k, k - \tau$ and obtain the second term in the last inequality. Next, we propose employing two different Taylor expansions to prove that $\spadesuit$ is of higher orders of $\alpha$. We first apply the Taylor expansion to $g(\theta_k^{[1]}, x_k) - g(\theta_k^{[2]}, x_k)$ and $g(\theta_{k-\tau}^{[1]}, x_k) - g(\theta_{k-\tau}^{[2]}, x_k)$. However, this only achieves $\spadesuit \lesssim \|w_k\|^2\big(\|w_k\| + \alpha\tau T_1\big)$, where $T_1 = \min(\|\theta_k^{[1]}\|, \|\theta_k^{[2]}\|, \|\theta_{k-\tau}^{[1]}\|, \|\theta_{k-\tau}^{[2]}\|) + 1$. When $\theta_k^{[1]}$ and $\theta_k^{[2]}$ are not close to each order, i.e., when $\|w_k\|$ is large, $\spadesuit$ is not necessarily of higher order. Therefore, we consider a second type of Taylor expansion on $g(\theta_k^{[1]}, x_k) - g(\theta_{k-\tau}^{[1]}, x_k)$ and $g(\theta_k^{[2]}, x_k) - g(\theta_{k-\tau}^{[2]}, x_k)$. The intuition for the second type of Taylor expansion is to analyze and bound $\spadesuit$ by the small distance between $\theta_k^{[j]}$ and $\theta_{k-\tau}^{[j]}$ for $j \in \{1, 2\}$, even when $\|w_k\|$ is large. This achieves $\spadesuit \lesssim \|w_k\|\alpha\tau T_1\big(\|w_k\| + \alpha\tau T_1\big)$. Simultaneously applying the two Taylor expansions will yield $\spadesuit \lesssim \alpha\tau\|w_k\|^2 T_1$. Finally, we overcome this challenge by carefully analyzing the boundness of $T_1$; see Theorem 4.1 and its proof.

In parallel to the above coupling approach, we also explore an alternative approach by verifying the joint Markov chain $(x_k, \theta_k)$ satisfies certain irreducibility and Lyapunov drift conditions, which in turn imply the chain is ergodic. To apply this approach, we exploit additional properties of the SA noise, namely miniorization, which is satisfied in many applications where additional randomness is injected to the SA update. While the high level strategy of this approach is well developed [23, 50],

carrying out the analysis of each step is technically involved. In particular, we need to translate the minorization property of the noise to the irreducibility of the joint chain $(x_k, \theta_k)$, which is nontrivial in the presence of Markovian noise and nonlinearity.

# 4   Main Results

## 4.1   Weak Convergence of Projected SA

Our first main result proves the ergodicity of the joint process $(x_k, \theta_k)_{k \geq 0}$ of the projected SA (2.1).

**Theorem 4.1** (Ergodicity of Projected SA). *Suppose that Assumption 1–4 ($p = 1$) hold. The projected SA (2.1) is applied with radius parameter $2\|\theta^*\| \leq \beta < \infty$. For stepsize $\alpha > 0$ that satisfies the constraint $\alpha\tau_\alpha \leq \frac{\mu}{(940+96\beta)L^2}$, the Markov chain $(x_k, \theta_k)_{k \geq 0}$ converges to a unique stationary distribution $\bar{\nu}_\alpha \in \mathcal{P}_2(\mathcal{X} \times \mathbb{R}^d)$. Let $\nu_\alpha := \mathcal{L}(\theta_\infty)$ be the second marginal of $\bar{\nu}_\alpha$. For $k \geq 2\tau_\alpha$, it holds that*

$$W_2(\mathcal{L}(\theta_k), \nu_\alpha) \leq \bar{W}_2(\mathcal{L}(x_k, \theta_k), \bar{\nu}_\alpha) \leq (1 - \alpha\mu)^{k/2} \cdot s(\theta_0, \theta^*, \mu, L, R).$$

Theorem 4.1 generalizes prior weak convergence results for constant stepsize SA/SGD either under i.i.d. noise [20, 67] or linear update [33, 45]. Our stepsize condition $\alpha\tau_\alpha \lesssim \mu/L^2$ coincides with [33, 60] on linear SA, a special case of our setting.

The proof of Theorem 4.1 highlights the stabilizing effect of the projection operation in (2.1). This effect, together with the smoothness of update function $g$, controls how the Markovian correlation propagates through the nonlinear update, allowing us to overcome the challenges discussed in Section 3. It is unclear whether our proof, which is based on Markov chain coupling, can be fully generalized to SA without projection. Nevertheless, we show that such a generalization is possible for a sub-family of nonlinear SA where $g$ possesses the additional structure termed "asymptotic linearity", which is satisfied by, e.g., SGD applied to certain settings of logistic regression. For a formal statement of this result and proof, we refer the readers to Appendix E.

As a by-product of our analysis, we establish the following non-asymptotic $2p$-th moment bound on the error $\theta_k - \theta^*$. Let $\theta_{t+1/2} := \theta_t + \alpha(g(\theta_t, x_t) + \xi_{t+1}(\theta_t))$ denote the pre-projection iterate.

**Proposition 4.2.** *Consider $(\theta_k)_{k \geq 0}$ of iteration (2.1) with $\beta \in [2\|\theta^*\|, \infty]$. Let Assumption 1–4($2p$) hold. If stepsize $\alpha$ satisfies $\alpha\tau_\alpha L^2 \leq c_p\mu$, with $c_p \leq 1$, the following holds for all $k \geq \tau_\alpha$,*

$$\mathbb{E}[\|\theta_{k+1} - \theta^*\|^{2p}] \leq \mathbb{E}[\|\theta_{k+1/2} - \theta^*\|^{2p}] \leq c_{p,1}(1 - \alpha\mu)^{k+1}\mathbb{E}[\|\theta_0 - \theta^*\|^{2p}] + c_{p,2}(\alpha\tau_\alpha)^p \cdot s(\theta_0, \theta^*, L, \mu).$$

Proposition 4.2 implies that $\mathbb{E}[\|\theta_k - \theta^*\|^{2p}] \lesssim (\alpha\tau)^p$ for sufficiently large $k$, generalizing the results of [17, 20, 60] to higher moments and the nonlinear Markovian setting. Notably, this result holds even without the projection operation in the SA update (2.1), i.e., $\beta = \infty$. Furthermore, Proposition 4.2 can be used to derive high-probability tail bounds using the Markov inequality.

## 4.2   Weak Convergence without Projection

Parallel to the coupling approach, we consider an alternative approach for establishing weak convergence via verifying irreducibility, positive Harris recurrence, and $V$-uniform ergodicity [50] of the Markov chain $(x_k, \theta_k)$. This approach applies to nonlinear SA even without projection. To verify irreducibility, we exploit the following additional noise structure.

**Assumption 5** (Noise Minorization). *For each $\theta \in \mathbb{R}^d$, the distribution of the random variable $\xi_1(\theta)$, denoted by $\zeta_\theta$, can be decomposed as $\zeta_\theta = \zeta_{1,\theta} + \zeta_{2,\theta}$, where the measure $\zeta_{1,\theta}$ has a density, denoted by $p_\theta$, which satisfies $\inf_{\theta \in C} p_\theta(t) > 0$ for any bounded set $C$ and any $t \in \mathbb{R}^d$.*

A similar assumption is considered in [5, 67]. This assumption is mild and satisfied by any continuous random field supported on $\mathbb{R}^d$. Introducing such (small) continuous noise is often part of the algorithm design for inducing privacy [2, 22] or exploration [28, 55]. Without Assumption 5, the chain may fail to be irreducible even when the other assumptions are satisfied; see [33] for a counterexample.

Under Assumption 5, we obtain the following ergodicity result paralleling Theorem 4.1.

**Theorem 4.3** (Ergodicity of SA – Minorization). *Suppose that Assumption 1–3, Assumption 4($p = 1$), and Assumption 5 hold. For stepsize $\alpha > 0$ that satisfies the constraint $\alpha\tau_\alpha L^2 < c_2\mu$, the Markov chain $(x_k, \theta_k)_{k\geq 0}$ of (2.1) with $\beta = \infty$ is $V$-uniformly ergodic with Lyapunov function $V(x, \theta) = \|\theta - \theta^*\|^2 + 1$ and a unique stationary distribution $\bar\nu_\alpha \in \mathcal{P}_2(\mathcal{X} \times \mathbb{R}^d)$. Moreover, defining the $V$-norm $\|\nu\|_V := \int |\nu(\mathrm{d}x)|V(x)$, we have*

$$\big\|\mathcal{L}(x_k, \theta_k) - \bar\nu_\alpha\big\|_V \leq \kappa\rho^k, \qquad \forall(x_0, \theta_0) \in \mathcal{X} \times \mathbb{R}^d, \forall k \geq 0, \tag{4.1}$$

*where the constants $\rho \in (0, 1)$ and $\kappa \in (0, \infty)$ may depend on $\alpha$.*

### 4.3 Non-Asymptotic Convergence Rate and Central Limit Theorem

In the sequel, let $(x_\infty, \theta_\infty^{(\alpha)})$ denote the random vector whose law is the stationary distribution $\bar\nu_\alpha$ given in Theorem 4.1. As a corollary, we have geometric convergence for the first 2 moments of $\theta_k$.

**Corollary 4.4** (Non-Asymptotic Convergence Rate). *Under the setting of Theorem 4.1, for any initialization of $\theta_0 \in \mathbb{R}^d$, we have*

$$\big\|\mathbb{E}[\theta_k] - \mathbb{E}[\theta_\infty^{(\alpha)}]\big\| \leq (1 - \alpha\mu)^{k/2} \cdot s'(\theta_0, \theta^*, \mu, L, R), \quad \text{and}$$
$$\big\|\mathbb{E}[\theta_k\theta_k^\top] - \mathbb{E}[\theta_\infty^{(\alpha)}(\theta_\infty^{(\alpha)})^\top]\big\| \leq (1 - \alpha\mu)^{k/2} \cdot s''(\theta_0, \theta^*, \mu, L, R).$$

Moreover, the convergence rate established in Theorem 4.1 is fast enough that we can use it to prove a Central Limit Theorem for the average iterates.

**Corollary 4.5** (Central Limit Theorem). *Under the setting of Theorem 4.1, as $k \to \infty$ we have $\frac{1}{\sqrt{k}}\sum_{t=0}^{k-1}\big(\theta_t - \mathbb{E}[\theta_\infty]\big) \Rightarrow \mathcal{N}(0, \Sigma^{(a)})$, where $\Sigma^{(\alpha)} := \lim_{k\to\infty}\frac{1}{k}\mathbb{E}\big[\big(\sum_{t=0}^{k-1}\big(\theta_t - \mathbb{E}[\theta_\infty^{(\alpha)}]\big)\big)^{\otimes 2}\big]$.*

Establishing the CLT sets the stage for using the SA iterates for statistical inference tasks such as confidence interval estimation. We discuss this in greater detail in Section 4.4 after characterizing the asymptotic bias, another important ingredient for using SA for inference.

### 4.4 Bias Characterization

In this subsection, we characterize the asymptotic bias $\mathbb{E}[\theta_\infty^{(\alpha)}] - \theta^*$. Understanding the bias structure has important algorithmic implications for bias reduction, which we explore in Section 4.5, as well as for more efficient statistical inference and confidence interval estimation [34].

**Theorem 4.6** (Bias Characterization). *Suppose Assumptions 1–4($p = 3$) hold. For each stepsize $\alpha > 0$ satisfying $\alpha\tau_\alpha L^2 < c_3\mu$, the following holds for some vector $b$ independent of $\alpha$:*

$$\mathbb{E}[\theta_\infty^{(\alpha)}] - \theta^* = \alpha b + \mathcal{O}\big((\alpha\tau_\alpha)^{3/2}\big). \tag{4.2}$$

*More specifically, the leading bias can be decomposed as $b = b_\mathrm{m} + b_\mathrm{n} + b_\mathrm{c}$, where*

$$b_\mathrm{m} = -(\bar{g}'(\theta^*))^{-1}\mathbb{E}[g'(\theta^*, x_\infty)h(\theta^*, x_\infty)], \tag{4.3}$$
$$b_\mathrm{n} = \frac{1}{2}(\bar{g}'(\theta^*))^{-1}\bar{g}''(\theta^*)A\Big(\mathbb{E}[g(\theta^*, x_\infty)^{\otimes 2}] + \mathbb{E}[(\xi_1(\theta^*))^{\otimes 2}]\Big), \tag{4.4}$$
$$b_\mathrm{c} = \frac{1}{2}(\bar{g}'(\theta^*))^{-1}\bar{g}''(\theta^*)A\Big(\mathbb{E}[g(\theta^*, x_\infty) \otimes h(\theta^*, x_\infty)] + \mathbb{E}[h(\theta^*, x_\infty) \otimes g(\theta^*, x_\infty)]\Big), \tag{4.5}$$

*with $A = (\bar{g}'(\theta^*) \otimes I + I \otimes \bar{g}'(\theta^*))^{-1}$ and $h(\theta^*, x) = \int_\mathcal{X}(I - P^* + \Pi)^{-1}(P^* - \Pi)(x, \mathrm{d}x')g(\theta^*, x')$, with the kernel $P^*$ being a regular conditional probability on $\mathcal{X}$ that satisfies $\int_B \pi(\mathrm{d}x)P^*(x, C) = \int_C \pi(\mathrm{d}y)P^*(y, B)$, for all $B, C \in \mathcal{B}(\mathcal{X})$.*

We defer the detailed proof to Appendix I. A few remarks are in order. First, we emphasize that (4.2) is essentially an equality, indicating a non-zero bias of order $\alpha$ whenever $b \neq 0$ (up to higher order terms). Notably, the Polyak-Ruppert averaging of the iterates cannot eliminate this bias. Note that the bias expansion in (4.2) applies to both weakly converged projected and non-projected SA. Our analysis shows that compared with the non-projected SA, the projection operator induces an extra bias term of the order $\mathcal{O}(\alpha^2\tau_\alpha^3)$, which is negligible relative to the main terms in in (4.2).

More importantly, Theorem 4.6 provides an explicit expression of the leading bias, which decomposes into three components: the Markovian part, the nonlinearity contribution, and a *compound* term,

which is unique in nonlinear Markovian SA. Specifically, $b_{\mathrm{m}}$ in (4.3) is associated with the Markovian multiplicative noise, where the matrix $P^* - \Pi$ in the $h$ function determines the mixing time of the data sequence $(x_k)_{k\geq}$. The term $b_{\mathrm{n}}$ in (4.4) is linked to nonlinearity, as reflected by the Hessian term $\bar{g}''(\theta^*) = \mathbb{E}[g''(\theta^*, x_\infty)]$, which quantifies the nonlinearity of $g$ and is equal to zero in the case of a linear $g$. Lastly, $b_{\mathrm{c}}$ in (4.5) is the *compound* term, due to its dependence on both the Markov noise ($h$ function) and the nonlinearity measure $\bar{g}''$. In particular, we note the following two special cases: (1) When $g$ is a linear function, $\bar{g}''(\theta^*) = 0$. Hence, $b_{\mathrm{n}} = b_{\mathrm{c}} = 0$, and $b_{\mathrm{m}}$ recovers the result in [33]; (2) When $(x_k)_{k\geq 0}$ is i.i.d. sampled from the stationary distribution $\pi$, we have $h(\theta^*, x) \equiv 0$ $\forall x \in \mathcal{X}$, for $P = P^* = \Pi$. As such, $b_{\mathrm{m}} = b_{\mathrm{c}} = 0$, recovering the result in [20]. The presence of the compound term $b_{\mathrm{c}}$ suggests that as the SA structure becomes more nonlinear and the underlying Markov chain mixes more slowly, the impact on the bias is *multiplicative* rather than simply additive, a surprising phenomenon not unveiled in previous studies. It is possible to improve the residual order from $\mathcal{O}(\alpha^{3/2})$ to $\mathcal{O}(\alpha^2)$ with a more refined characterization of the asymptotic second moment $\mathbb{E}[(\theta_\infty - \theta^*)^{\otimes 2}]$ by following a similar strategy as our current approach. We leave this refinement out of the scope of the current paper.

### 4.5 Algorithmic Implications

We examine the practical implications of our weak convergence and bias characterization results, particularly for Polyak-Ruppert (PR) tail averaging and Richardson-Romberg (RR) extrapolation. In this subsection, we focus on the dependence on the stepsize $\alpha$ and iteration index $k$, and make use of the big-O notation from Section 1. Recall that $b$ is the bias vector defined in Theorem 4.6.

PR averaging [56, 59] is a classical approach for reducing the variance and accelerating the convergence of SA. Here we consider the tail-averaging variant of PR averaging, defined as $\bar{\theta}_{k_0,k} := \frac{1}{k-k_0} \sum_{t=k_0}^{k-1} \theta_t$, for $k \geq k_0$, with a user-specified burn-in period $k_0 \geq 0$ (a common choice is $k_0 = k/2$). The following corollary, proved in Appendix J, provides a non-asymptotic bound on the mean squared error (MSE) for the averaged iterates $\bar{\theta}_{k_0,k}$.

**Corollary 4.7** (Tail Averaging). *Under the setting of Theorem 4.6, the tail-averaged iterates satisfy the following bounds for all $k > k_0 + 2\tau_\alpha$ and $k_0 \geq \tau_\alpha + \frac{1}{\alpha\mu} \log\left(\frac{1}{\alpha\tau_\alpha}\right)$,*

$$\mathbb{E}\left[\|\bar{\theta}_{k_0,k} - \theta^*\|^2\right] = \underbrace{\alpha^2\|b\|^2 + \mathcal{O}\left(\alpha \cdot (\alpha\tau)^{\frac{3}{2}}\right)}_{T_1:\ asymptotic\ squared\ bias} + \underbrace{\mathcal{O}\left(\frac{\tau_\alpha}{k-k_0}\right)}_{T_2:\ variance} + \underbrace{\mathcal{O}\left(\frac{(1-\alpha\mu)^{k_0/2}}{\alpha\,(k-k_0)^2}\right)}_{T_3:\ optimization\ error}.$$

Corollary 4.7 shows that the MSE can be decomposed into three terms and elucidates how these terms depend on $\alpha, k$, and other problem parameters. In particular, the term $T_1$ corresponds to the asymptotic squared bias $\|\mathbb{E}[\theta_\infty^{(\alpha)} - \theta^*]\|^2$, which is not affected by averaging. The term $T_2$ is associated with the variance $\mathrm{Var}(\bar{\theta}_{k_0,k})$, which decays at rate $1/k$ due to averaging. Lastly, the term $T_3$ represents the optimization error $\|\mathbb{E}\bar{\theta}_{k_0,k} - \theta_\infty\|^2$, which decays geometrically in $k_0$ thanks to the use of a constant stepsize $\alpha$ and the tail-averaging procedure.

Note that averaging does not affect the bias of order $\alpha$. With the precise bias characterization in Theorem 4.6, we can order-wise reduce the bias to $\mathcal{O}(\alpha^{3/2})$ by employing the RR extrapolation technique [61]. Let $\bar{\theta}_{k_0,k}^{(\alpha)}$ and $\bar{\theta}_{k_0,k}^{(2\alpha)}$ denote the tail-averaged iterates using two stepsizes $\alpha$ and $2\alpha$ with the same data $(x_k)_{k\geq 0}$. The RR extrapolated iterates are defined as $\widetilde{\theta}_{k_0,k}^{(\alpha)} = 2\bar{\theta}_{k_0,k}^{(\alpha)} - \bar{\theta}_{k_0,k}^{(2\alpha)}$.

**Corollary 4.8** (RR-Extrapolation). *Under the setting of Theorem 4.6, the RR-extrapolated iterates satisfy the following bounds for all $k > k_0 + 2\tau_\alpha$ and $k_0 \geq \tau_\alpha + \frac{1}{\alpha\mu} \log\left(\frac{1}{\alpha\tau_\alpha}\right)$,*

$$\mathbb{E}\left[\|\widetilde{\theta}_{k_0,k} - \theta^*\|^2\right] = \mathcal{O}\left((\alpha\tau_\alpha)^3\right) + \mathcal{O}\left(\frac{\tau_\alpha}{k-k_0}\right) + \mathcal{O}\left(\frac{(1-\alpha\mu)^{k_0/2}}{\alpha\,(k-k_0)^2}\right).$$

Backed by the CLT in Corollary 4.5, the iterates of constant-stepsize SA can be used to construct confidence intervals of $\theta^*$. For i.i.d. data or linear SA, this approach has been explored in [34, 47, 66, 67] along with an appropriate variance estimator [26, 66]. In our Markovian nonlinear setting, where the iterates are biased, it is crucial to use RR extrapolation for bias reduction. Once the bias is accounted for, the power of using constant stepsizes reveals itself as it leads to rapid mixing and

low correlation of the iterates. Together, they lead to efficient confidence interval estimation schemes using nonlinear Markovian SA; see the empirical results in [34] showing its efficacy. In contrast, the classical diminishing stepsize paradigm often suffers from high correlation [13] and in turn inaccurate variance estimation, resulting in unsatisfactory coverage probability with finite data [34].

## 4.6 Implications for Learning GLM

Generalized linear models (GLM) extend linear regression to the model $\mathbb{E}[Y|W] = \sigma(W^\top \theta^*)$, where $W$ is the covariate, $Y$ the response variable, and $\sigma$ is called the *mean function*. For any monotone (and potentially nonlinear) $\sigma$, the powerful framework developed in [19, 36, 37, 64] allows one to formulate the estimation of $\theta^*$ as minimizing an appropriate *convex* (surrogate) loss function. Applying SGD to this loss leads to a nonlinear SA update, to which our results are applicable. Below we discuss their applications in two concrete examples of GLMs.

**Logistic Regression**    Logistic regression uses a sigmoid mean function $\sigma(x) = \frac{1}{1+\exp(-x)}$. Suppose the covariate $w_k$ is sequentially sampled from a uniformly ergodic Markov chain with a bounded state space $\mathcal{W} \subset \mathbb{R}^d$, and conditioned on $w_k$ the response $y_k$ is Bernoulli distributed with parameter $(1 + \exp(-w_k^\top \theta^*))^{-1}$. SGD applied to the $L_2$-regularized negative log-likelihood function takes the form of the SA update $\theta_{k+1} = \theta_k + \alpha g(\theta_k, x_k)$, where $x_k = (w_k, y_k) \in \mathcal{W} \times \{0, 1\}$ and $g(\theta_k, x_k) = -w_k\big(\sigma(-w_k^\top \theta_k) - y_k\big) - \lambda \theta_k$. For simplicity, we do not consider $\xi$-perturbation, i.e., $\xi_{k+1}(\theta_k) \equiv 0$. It is easy to verify that this $g$ is strongly monotone and sufficiently smooth with at most linear growth in $|\theta|$, hence satisfying Assumption 1–3. Therefore, all the results in Sections 4.1–4.5 apply to logistic regression with constant stepsizes and Markovian data.

**Smooth ReLU Regression**    The mean function $\sigma$ can be interpreted as playing a similar role as the activation function in neural networks. Widely adopted is ReLU activation $\sigma(x) = \max(0, x)$ as well as its various smooth approximations [7, 30]. The problem of learning $\theta^*$ in this setting, sometimes called ReLU Regression, has been studied in the last decade and recently regained attention [19, 36, 37, 64]. Unlike linear or logistic regression, the least squares and maximum likelihood formulation associated with such nonlinear mean functions $\sigma$ is non-convex. Nevertheless, the convex surrogate loss framework in [19, 64] still applies. As an example, we focus on the SoftPlus activation $\sigma(x) = \log(1 + \exp(\iota x))/\iota$ with a temperature parameter $\iota > 0$ [30]. With $L_2$-regularization the resulting SGD iteration is $\theta_{k+1} = \theta_k - \alpha\big(w_k\big(\frac{1}{\iota} \log(1 + \exp(\iota w_k^\top \theta_k)) - y_k\big) + \lambda \theta_k\big)$, where the covariate-response pair $(\theta_k, x_k)$ is as before. This problem can again be cast as nonlinear SA with a strongly monotone and smooth $g$, satisfying Assumptions 1–3. All results in Sections 4.1–4.5 apply.

# 5    Numerical Experiments

In this section, we provide numerical experiment results to verify our theoretical results. We run SGD on $L_2$-regularized logistic regression with Markovian data and constant stepsizes, where the covariate $x_t \in \mathbb{R}$ is sequentially sampled from an autoregressive (AR) model of order 1; specifically, $x_{t+1} = 0.9x_t + \zeta_{t+1}$ with $\zeta_t$ i.i.d. following $N(0, 1)$, and the stationary distribution is $x_\infty \sim N(0, 1/(1-0.9^2))$. The binary dependent variable $y_t$ is sampled from Bernoulli$(1/(1+\exp(-w^* x_t))$ with $w^* = 1$. The regularized parameter is set to $\lambda = 0.0001$.[1]

To examine the asymptotic bias, we run the experiment for an episode length of $10^7$, with Markovian data as well as i.i.d. data sampled from $N(0, 1/(1 - 0.9^2))$. We plot the errors (distance to $\theta^*$) of the PR averaged iterates and the RR extrapolated iterates, for different stepsizes $\alpha$. Figure 1(a) verifies the presence of an asymptotic bias approximately proportional to the stepsize $\alpha$, and illustrates the effectiveness of RR extrapolation in reducing this bias. In Figure 1(b), we compare the bias under Markovian data ($x_{t+1} \sim P(\cdot|x_t)$) and i.i.d. data ($x_t \sim x_\infty$). Interestingly, Figure 1(b) reveals that Markovian data does not necessarily lead to a larger bias than i.i.d. data. This is consistent with our theory, as the three bias terms $b_m, b_n, b_c$ may have opposite signs leading to cancellation. This result suggests that in the presence of nonlinearity, one should not avoid Markovian data simply for the sake of reducing bias. Rather, RR extrapolation may be more effective for bias reduction.

---

[1] The experiments were conducted on two sockets of Intel(R) Xeon(R) Gold 6154 CPU @ 3.00GHz with 566Gb of RAM. Implementation details and code are available at https://github.com/lucyhuodongyan/nonlinear-sa-bias.

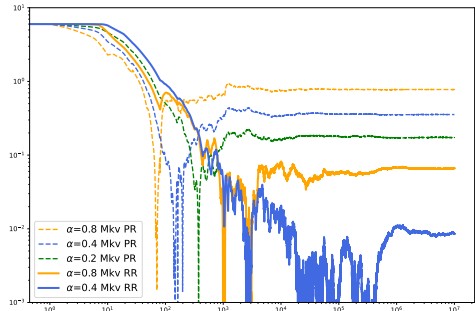
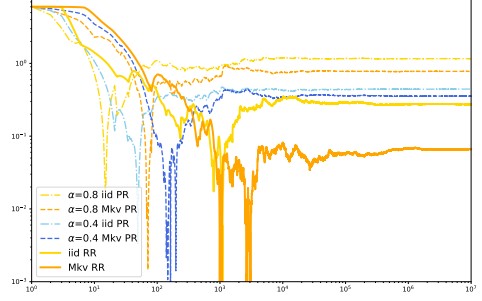

(a) This plot shows the errors of PR-averaged iterates and RR-extrapolated iterates, all generated using Markovian data.

(b) This plot compares the errors under Markovian data $(x_t \sim P(\cdot|x_{t-1}))$ and iid data $(x_t \sim \pi)$, where all other settings are the same.

Figure 1: Experiment results to illustrate the properties of asymptotic bias.

To verify the CLT in Corollary 4.5, we repeat the experiment 1000 times with an episode length of $10^6$ and stepsize $\alpha = 0.8$. We compute the PR averaged iterates and plot the histogram and the quantile-quantile (QQ) plot in Figure 2 in Appendix C. The close alignment between the histogram and the normal curve in Figure 2(a) and the linearity of the points along the 45-degree reference line in the QQ plot in Figure 2(b) confirm that the empirical distribution follows a normal distribution.

## 6 Related Work

**General SA and SGD.** SA and SGD can be traced back to the seminal work of [58]. Classical work assumes a diminishing stepsize sequence, and has shown almost sure asymptotic convergence to $\theta^*$ [8, 58]. Subsequent works propose the iterate averaging technique, now known as Polyak-Ruppert (PR) averaging, to reduce variance and accelerate convergence [56, 59], and also establish a Central Limit Theorem for the asymptotic normality of the averaged iterates [57]. The asymptotic convergence theory of SA and SGD is well developed and extensively addressed in many exemplary textbooks, see [3, 40, 65]. There are also recent works studying the non-asymptotic convergence with diminishing stepsizes [12, 14]. The recent work [15] establishes the high probability bound on the estimation error of contractive SA with diminishing stepsize.

**SA and SGD with Constant Stepsizes.** There has been an increasing interest in studying SA with constant stepsize. Many works in this line provide non-asymptotic upper bounds on mean squared error (MSE) $\mathbb{E}[\|\theta_t - \theta^*\|^2]$. Works in [25, 42, 51] study linear SA (LSA) under i.i.d. data. Recent works extend the analysis of the MSE to LSA with Markovian data, such as [24, 52, 60]. There are also works providing upper bounds of MSE for general contractive SA with Markovian noise [14, 17].

In addition to non-asymptotic guarantees, some works focus on the asymptotic behavior of SA iterates. Recent works have shown that when using constant stepsize, one loses the almost sure convergence guarantee in the diminishing stepsize sequence regime, and at best can achieve distributional convergence, as demonstrated in [16, 20, 25, 33, 66, 67, 69]. The presence of asymptotic bias is also a recurring theme in recent literature, with precise characterization given in [20] for strongly-convex SGD with i.i.d. data and in [33] for LSA with Markovian data. Works in [34, 51, 66, 67, 69] also establish Central Limit Theorems for averaged SA iterates with constant stepsizes.

## 7 Conclusion

We provide the first weak convergence and steady-state analysis for constant-stepsize SA with both nonlinear update and Markovian data. Our analysis elucidates the compound effect of nonlinearity and memory, which leads to new analytical challenges and behaviors. A limitation of our results is the use of a projection step or the noise minorization assumption. Whether they can be removed is worth investigating. Other future directions include refining the dimension dependence in our results, as well as a theoretical investigation of statistical inference.

**Acknowledgement**    Y. Chen is partially supported by National Science Foundation (NSF) grants CCF-1704828 and NSF CCF-2233152. Y. Zhang and Q. Xie are supported in part by NSF grants CNS-1955997, EPCN-2339794 and EPCN-2432546. Y. Zhang is also supported in part by NSF Award DMS-2023239.

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

# A  Additional Notations

**General Probability**  We write $z_1 \perp\!\!\!\perp z_2 \mid z_3$ if random variables $z_1$ and $z_2$ are conditionally independent given $z_3$.

Recall that we define the metric $\bar{d}\big((x,\theta),(x',\theta')\big) := \sqrt{\mathbb{1}\{x \neq x'\} + \|\theta - \theta'\|^2}$ for the space $\mathcal{X} \times \mathbb{R}^d$. Thus, for $\bar{\mu}$ and $\bar{\nu}$ in $\mathcal{P}_2(\mathcal{X} \times \mathbb{R}^d)$, the Wasserstein-2 distance w.r.t. $\bar{d}$ is computed as

$$\bar{W}_2(\bar{\mu}, \bar{\nu}) = \inf \left\{ \left(\mathbb{E}[\mathbb{1}\{x \neq x'\} + \|\theta - \theta'\|^2]\right)^{\frac{1}{2}} : \mathcal{L}\big((x,\theta)\big) = \bar{\mu}, \mathcal{L}\big((x',\theta')\big) = \bar{\nu} \right\}.$$

**General State Space Markov Chains**  Throughout the paper, we assume that $\mathcal{X}$ is a Borel space. Let $P$ denote the transition kernel. We call $\pi$ the stationary distribution of $P$ if it satisfies $\int_{\mathcal{X}} \pi(dx)P(x,B) = \pi(B)$, for $B \in \mathcal{B}(\mathcal{X})$. Define the $\pi$-weighted inner product $\langle f, g \rangle_{L^2(\pi)} = \int_{\mathcal{X}} \pi(dx)f^\top(x)g(x)$ and the induced norm $\|f\|_{L^2(\pi)} = (\langle f, f \rangle_{L^2(\pi)})^{1/2}$. Let $L^2(\pi) = \{f : \|f\|_{L^2(\pi)} < \infty\}$ denote the corresponding Hilbert space of $\mathbb{R}^d$-valued, square-integrable and measurable functions on $\mathcal{X}$. For an operator $T : L^2(\pi) \to L^2(\pi)$, its operator norm is defined as $\|T\|_{L^2(\pi)} = \sup_{\|f\|_{L^2(\pi)}=1} \|Tf\|_{L^2(\pi)}$. The transition kernel is a bounded linear operator on $L^2(\pi)$, in particular with norm $\|P\|_{L^2(\pi)=1}$. Also, we define the kernel/operator $\Pi = 1 \otimes \pi$ by $\Pi(x, \cdot) = \pi$.

Throughout the paper, we assume that $\mathcal{X}$ is a Borel space. Let $P$ denote the transition kernel. We call $\pi$ the stationary distribution of $P$ if it satisfies $\int_{\mathcal{X}} \pi(dx)P(x,B) = \pi(B)$, for $B \in \mathcal{B}(\mathcal{X})$. There exists a kernel $P^*$ as a regular conditional probability that satisfies $\int_A \pi(ds)P(x,B) = \int_B \pi(dy)P^*(y,A)$, for $A, B \in \mathcal{B}(\mathcal{X})$ [27, Chapter 21.4, Theorem 19], and $P^*$ defines the probability law for the time-reversed chain of $(x_k)_{k \geq 0}$.

# B  Additional Discussion on Assumptions

In this section, we provide a more detailed discussion of the assumptions taken in this work.

**Projection and Minorization**  Projection steps have a longstanding presence in SA literature for tractability in convergence theory, as seen in many analyses of SGD [6, 11, 39, 41, 54]. Although not an algorithmic proposal, this additional projection step does not incur much computational cost in practice, as it only involves rescaling the iterates, and the projection radius can be estimated a priori. Before our work, no studies had proven weak convergence for non-linear SA with Markovian data and constant stepsize, with or without the projection. Thus, our result is the first to prove detailed weak convergence in this setting. Nonetheless, we provide an alternative proof of weak convergence in Theorem 4.3 using the Drift and Minorization technique, which does not require a projection.

**Differentiability**  The differentiability condition of the SA update operator $g$ in Assumption 2 ensures controlled evolution of the iterates $\theta$. This differentiability assumption supports a third-order Taylor expansion of $g$ with a bounded remainder, which is crucial for both analyzing the ♠ term in the convergence proof as discussed in Section 3 and for bias characterization. Moreover, some form of differentiability assumption is standard in SA literature, such as [1, 20, 45], particularly when one seeks a fine-grained characterization of the iterates' distributional property. Such an assumption is satisfied by many GLMs, such as logistic regression and Poisson regression. When $g$ is not differentiable, the bias of SA iterates behaves drastically different [68], which is beyond the scope of this paper.

**Strong Monotonicity**  The strong monotonicity assumption is common in SA literature. Together with smoothness, it allows us to establish geometric distributional convergence. While some GLMs by themselves do not satisfy this condition, it is a common practice to apply $L_2$-regularization (equivalently, weight decay) to ensure strong convexity and improve statistical performance. It is a standard calculation that one can appropriately choose the regularization parameter to derive tight results for non-strongly-convex functions.

## C  Additional Plots

In this section, we present the plots from numerical experiments in Section 5 that verify the Central Limit Theorem (CLT) in Corollary 4.5. In these plots, we plot the centered and scaled PR-averaged iterates, i.e, $\sqrt{T}\left(\bar{\theta}_T^{(k)} - \sum_{l=1}^{1000} \bar{\theta}_T^{(l)}\right)$, where $\bar{\theta}^{(k)} = \left(\sum_{t=1}^{T} \theta_t\right)/T$ with $T = 10^6$ is the PR-averaged iterates for the $k$-th repeat.

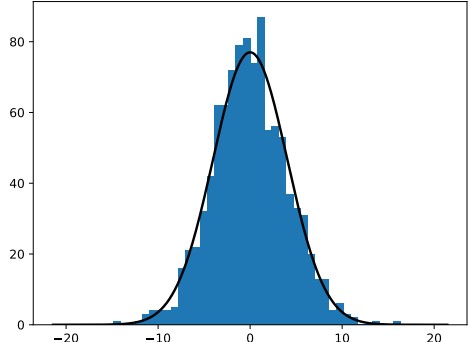

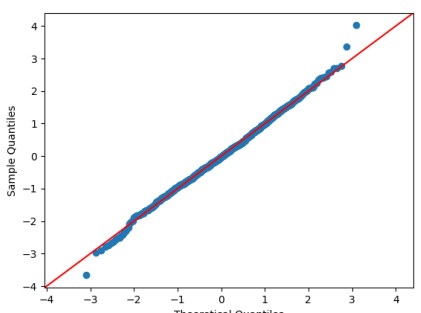

(a) Histogram of Centered and Scaled PR-Averaged Iterates with Fitted Normal Density. This histogram displays the distribution of the PR-averaged iterates from the experiment. Overlaid on the histogram is a fitted normal density curve. The close alignment indicates that the empirical distribution of the iterates closely follows the normal distribution.

(b) QQ Plot of Centered and Scaled PR-Averaged Iterates. This QQ plot compares the empirical distribution of the PR-averaged iterates from the $L_2$-regularized logistic regression experiment with the theoretical normal distribution. The linearity of the points along the 45-degree reference line indicates that the empirical distribution closely follows the normal distribution.

Figure 2: Experiment results to verify the Central Limit Theorem.

## D  Proof of Pilot Results (Proposition 4.2)

In this section, we prove the pilot result, namely Proposition 4.2. We prove the desired moments for $\beta = \infty$, i.e., without any projection. It is easy to see that when the projection radius $\beta \in [2\|\theta^*\|, \infty]$,

$$\mathbb{E}[\|\theta_{t+1} - \theta^*\|^{2p}] \leq \mathbb{E}[\|\theta_{t+1/2} - \theta^*\|^{2p}],$$

where $\theta_{t+1/2}$ denotes the iterate before projection. The term on the right hand side can be further bounded by the moment bounds for iteration without projection. Therefore, it suffices for us to prove the respective moment bounds without any projection.

Given Assumption 4 hold for $2p$-th moment, with $p \geq 1$, we prove the moment bound in Proposition 4.2 for $n$ with $1 \leq n \leq p$ by induction.

### D.1  Base Case

In this section, we prove the base case of Proposition 4.2, i.e., with $n = 1$. The base case gives the desired mean squared error (MSE) convergence bound, which will subsequently be used in the proof of weak convergence.

We start by noting the following decomposition,

$$
\begin{aligned}
&\mathbb{E}[\|\theta_{k+1} - \theta^*\|^2] - \mathbb{E}[\|\theta_k - \theta^*\|^2] \\
&= 2\alpha\mathbb{E}[\langle\theta_k - \theta^*, g(\theta_k, x_k)\rangle] + \alpha^2\mathbb{E}[\|g(\theta_k, x_k)\|^2] + \alpha^2\mathbb{E}[\|\xi_{k+1}(\theta_k)\|^2] \\
&= 2\alpha\mathbb{E}[\langle\theta_k - \theta^*, g(\theta_k, x_k) - \bar{g}(\theta_k)\rangle] + 2\alpha\mathbb{E}[\langle\theta_k - \theta^*, \bar{g}(\theta_k)\rangle] \\
&\quad + \alpha^2\mathbb{E}[\|g(\theta_k, x_k)\|^2] + \alpha^2\mathbb{E}[\|\xi_{k+1}(\theta_k)\|^2].
\end{aligned}
$$

It is easy to see that under Assumption 3, we have

$$\langle\theta_k - \theta^*, \bar{g}(\theta_k)\rangle = \langle\theta_k - \theta^*, \bar{g}(\theta_k) - \bar{g}(\theta^*)\rangle \leq -\mu\|\theta_k - \theta^*\|^2. \tag{D.1}$$

Additionally, under Assumption 2 and 4, we have the following upper bound

$$\alpha^2 \Big( \mathbb{E}[\|g(\theta_k, x_k)\|^2] + \mathbb{E}[\|\xi_{k+1}(\theta_k)\|^2] \Big)$$

$$\leq \alpha^2 \Big( L_1^2 \mathbb{E}[(\|\theta_k - \theta^*\| + 1)^2] + L_2^2 \mathbb{E}[(\|\theta_k - \theta^*\| + 1)^2] \Big)$$

$$\leq 2\alpha^2 L^2 \Big( \mathbb{E}[\|\theta_k - \theta^*\|^2] + 1 \Big). \tag{D.2}$$

Therefore, the key to analyze the remaining inner product $\langle \theta_k - \theta^*, g(\theta_k, x_k) - \bar{g}(\theta_k) \rangle$.

Consider the following decomposition

$$\langle \theta_k - \theta^*, g(\theta_k, x_k) - \bar{g}(\theta_k) \rangle = \langle \theta_k - \theta_{k-\tau}, g(\theta_k, x_k) - \bar{g}(\theta_k) \rangle \tag{D.3}$$

$$+ \langle \theta_{k-\tau} - \theta^*, g(\theta_{k-\tau}, x_k) - \bar{g}(\theta_{k-\tau}) \rangle \tag{D.4}$$

$$+ \langle \theta_{k-\tau} - \theta^*, g(\theta_k, x_k) - g(\theta_{k-\tau}, x_k) \rangle \tag{D.5}$$

$$+ \langle \theta_{k-\tau} - \theta^*, \bar{g}(\theta_k) - \bar{g}(\theta_{k-\tau}) \rangle. \tag{D.6}$$

Hence, we need some upper bound on $\|\theta_k - \theta_{k-\tau}\|$.

We next note the following technical Lemma, which is adapted from [17, 60] for the updated unbounded i.i.d. noise assumption in Assumption 4. The proof of the technical Lemma is delayed to Appendix D.1.1.

**Lemma D.1.** *For $16\alpha\tau \leq \mu/(4L^2)$, we have*

$$\mathbb{E}[\|\theta_k - \theta_{k-\tau}\| | \mathcal{F}_{k-\tau}] \leq 2\alpha\tau L \|\theta_{k-\tau} - \theta^*\| + 2\alpha\tau L \tag{D.7}$$

$$\mathbb{E}[\|\theta_k - \theta_{k-\tau}\| | \mathcal{F}_{k-\tau}] \leq 4\alpha\tau L \mathbb{E}[\|\theta_k - \theta^*\| | \mathcal{F}_{k-\tau}] + 4\alpha\tau L \tag{D.8}$$

$$\mathbb{E}[\|\theta_k - \theta_{k-\tau}\|^2 | \mathcal{F}_{k-\tau}] \leq 8\alpha^2\tau^2 L^2 \|\theta_{k-\tau} - \theta^*\|^2 + 8\alpha^2\tau^2 L^2 \tag{D.9}$$

$$\mathbb{E}[\|\theta_k - \theta_{k-\tau}\|^2 | \mathcal{F}_{k-\tau}] \leq 32\alpha^2\tau^2 L^2 \mathbb{E}[\|\theta_k - \theta^*\|^2 | \mathcal{F}_{k-\tau}] + 32\alpha^2\tau^2 L^2. \tag{D.10}$$

Given (D.10), we additionally note that

$$\|\theta_{k-\tau} - \theta^*\|^2 + 1 = \mathbb{E}[\|\theta_{k-\tau} - \theta^*\|^2 | \mathcal{F}_{k-\tau}] + 1$$

$$\leq 2\Big( \mathbb{E}[\|\theta_k - \theta_{k-\tau}\|^2 | \mathcal{F}_{k-\tau}] + \mathbb{E}[\|\theta_k - \theta^*\|^2 | \mathcal{F}_{k-\tau}] \Big) + 1$$

$$\leq 2\Big( 32\alpha^2\tau^2 L^2 (\mathbb{E}[\|\theta_k - \theta^*\|^2 | \mathcal{F}_{k-\tau}] + 1) + \mathbb{E}[\|\theta_k - \theta^*\|^2 | \mathcal{F}_{k-\tau}] \Big) + 1$$

$$\leq 4(\mathbb{E}[\|\theta_k - \theta^*\|^2 | \mathcal{F}_{k-\tau}] + 1). \tag{D.11}$$

We next use the above four technical inequalities to analyze the four terms in (D.3)–(D.4).

To bound (D.3), we first note that

$$\|\mathbb{E}[\langle \theta_k - \theta_{k-\tau}, g(\theta_k, x_k) - \bar{g}(\theta_k) \rangle | \mathcal{F}_{k-\tau}]\|$$

$$\leq \mathbb{E}[\|\theta_k - \theta_{k-\tau}\| \cdot 2L(\|\theta_k - \theta^*\| + 1) | \mathcal{F}_{k-\tau}]$$

$$\overset{(i)}{\leq} 2L \sqrt{\mathbb{E}[\|\theta_k - \theta_{k-\tau}\|^2 | \mathcal{F}_{k-\tau}]} \sqrt{\mathbb{E}[(\|\theta_k - \theta^*\| + 1)^2 | \mathcal{F}_{k-\tau}]}$$

$$\overset{(ii)}{\leq} 2L \sqrt{32\alpha^2\tau^2 L^2 (\mathbb{E}[\|\theta_k - \theta^*\|^2 | \mathcal{F}_{k-\tau}] + 1)} \sqrt{2(\mathbb{E}[\|\theta_k - \theta^*\|^2 | \mathcal{F}_{k-\tau}] + 1)}$$

$$\leq 16\alpha\tau L^2 (\mathbb{E}[\|\theta_k - \theta^*\|^2 | \mathcal{F}_{k-\tau}] + 1),$$

where (i) holds for the Cauchy-Schwarz inequality and (ii) holds for (D.10).

To bound (D.4), we next note that

$$\|\mathbb{E}[\langle \theta_{k-\tau} - \theta^*, g(\theta_{k-\tau}, x_k) - \bar{g}(\theta_{k-\tau}) \rangle | \mathcal{F}_{k-\tau}]\|$$

$$= \|\langle \theta_{k-\tau} - \theta^*, \mathbb{E}[g(\theta_{k-\tau}, x_k) - \bar{g}(\theta_{k-\tau}) | \mathcal{F}_{k-\tau}] \rangle\|$$

$$\leq \|\theta_{k-\tau} - \theta^*\| \|\mathbb{E}[g(\theta_{k-\tau}, x_k) - \bar{g}(\theta_{k-\tau}) | \mathcal{F}_{k-\tau}]\|$$

$$\overset{(iii)}{\leq} \|\theta_{k-\tau} - \theta^*\| \cdot \Big( \alpha L(\|\theta_{k-\tau} - \theta^*\| + 1) \Big)$$

$$\leq 2\alpha L(\|\theta_{k-\tau} - \theta^*\|^2 + 1)$$

$$\overset{(iv)}{\leq} 8\alpha L(\mathbb{E}[\|\theta_k - \theta^*\|^2 | \mathcal{F}_{k-\tau}] + 1)$$
$$\leq 8\alpha\tau L^2(\mathbb{E}[\|\theta_k - \theta^*\|^2 | \mathcal{F}_{k-\tau}] + 1),$$

where (iii) holds due to the mixing property of Markov chain $(x_k)_{k\geq 0}$ and (iv) holds for (D.11).

To bound (D.5), we have

$$\|\mathbb{E}[\langle\theta_{k-\tau} - \theta^*, g(\theta_k, x_k) - g(\theta_{k-\tau}, x_k)\rangle | \mathcal{F}_{k-\tau}]\|$$
$$= \|\langle\theta_{k-\tau} - \theta^*, \mathbb{E}[g(\theta_k, x_k) - g(\theta_{k-\tau}, x_k) | \mathcal{F}_{k-\tau}]\rangle\|$$
$$\leq L\|\theta_{k-\tau} - \theta^*\| \cdot \mathbb{E}[\|\theta_k - \theta_{k-\tau}\| | \mathcal{F}_{k-\tau}]$$
$$\overset{(v)}{\leq} L\|\theta_{k-\tau} - \theta^*\| \cdot 2\alpha\tau L(\|\theta_{k-\tau} - \theta^*\| + 1)$$
$$\leq 4\alpha\tau L^2(\|\theta_{k-\tau} - \theta^*\|^2 + 1)$$
$$\overset{(vi)}{\leq} 16\alpha\tau L^2(\mathbb{E}[\|\theta_k - \theta^*\|^2 | \mathcal{F}_{k-\tau}] + 1),$$

where (v) holds for (D.7) and (vi) holds for (D.11).

Lastly, to bound (D.6), we apply the similar technique used in bounding the third term in (D.5) and obtain a similar result

$$\|\mathbb{E}[\langle\theta_{k-\tau} - \theta^*, \bar{g}(\theta_k) - \bar{g}(\theta_{k-\tau})\rangle | \mathcal{F}_{k-\tau}]\| \leq 16\alpha\tau L^2(\mathbb{E}[\|\theta_k - \theta^*\|^2 | \mathcal{F}_{k-\tau}] + 1).$$

Combining all analyses above, we have

$$\|2\alpha\mathbb{E}[\langle\theta_k - \theta^*\rangle, g(\theta_k, x_k) - \bar{g}(\theta_k) | \mathcal{F}_{k-\tau}]\|$$
$$\leq 2\alpha(16\alpha\tau L^2 + 8\alpha\tau L^2 + 32\alpha\tau L^2)(\mathbb{E}[\|\theta_k - \theta^*\|^2 | \mathcal{F}_{k-\tau}] + 1)$$
$$\leq 112\alpha^2\tau L^2(\mathbb{E}[\|\theta_k - \theta^*\|^2 | \mathcal{F}_{k-\tau}] + 1). \tag{D.12}$$

Hence, making use of (D.1), (D.2), and (D.12), we obtain the following

$$\mathbb{E}[\|\theta_{k+1} - \theta^*\|^2 | \mathcal{F}_{k-\tau}] - \mathbb{E}[\|\theta_k - \theta^*\|^2 | \mathcal{F}_{k-\tau}]$$
$$\leq -2\alpha\mu\mathbb{E}[\|\theta - \theta^*\|^2 | \mathcal{F}_{k-\tau}] + 112\alpha^2\tau L^2(\mathbb{E}[\|\theta_k - \theta^*\|^2 | \mathcal{F}_{k-\tau}] + 1)$$
$$+ 2\alpha^2 L^2(\mathbb{E}[\|\theta_k - \theta^*\|^2 | \mathcal{F}_{k-\tau}] + 1)$$
$$\leq -2\alpha\mu\mathbb{E}[\|\theta - \theta^*\|^2 | \mathcal{F}_{k-\tau}] + 114\alpha^2\tau L^2\mathbb{E}[\|\theta_k - \theta^*\|^2 | \mathcal{F}_{k-\tau}] + 114\alpha^2\tau L^2$$
$$= -2\alpha(\mu - 57\alpha\tau L^2)\mathbb{E}[\|\theta_k - \theta^*\|^2 | \mathcal{F}_{k-\tau}] + 114\alpha^2\tau L^2.$$

Therefore, when we have $\alpha$ satisfying the constraint, i.e., $\alpha\tau L^2 < c_{2,1}\mu$, we obtain

$$\mathbb{E}[\|\theta_{k+1} - \theta^*\|^2 | \mathcal{F}_{k-\tau}] \leq (1 - \alpha\mu)\mathbb{E}[\|\theta_k - \theta^*\|^2 | \mathcal{F}_{k-\tau}] + 114\alpha^2\tau L^2.$$

Recursively, we get

$$\mathbb{E}[\|\theta_k - \theta^*\|^2] \leq (1 - \alpha\mu)^{k-\tau}\mathbb{E}[\|\theta_\tau - \theta^*\|^2] + \frac{114\alpha\tau L^2}{\mu}$$
$$\leq 2(1 - \alpha\mu)^{k-\tau}\left(\mathbb{E}[\|\theta_0 - \theta^*\|^2] + \mathbb{E}[\|\theta_\tau - \theta_0\|^2]\right) + \frac{114\alpha\tau L^2}{\mu}$$
$$\leq 2(1 - \alpha\mu)^{k-\tau}\left(\mathbb{E}[\|\theta_0 - \theta^*\|^2] + 8\alpha^2\tau^2 L^2\left(\mathbb{E}[\|\theta_0 - \theta^*\|^2] + 1\right) + \frac{114\alpha\tau L^2}{\mu}\right)$$
$$\leq 4(1 - \alpha\mu)^{k-\tau}\mathbb{E}[\|\theta_0 - \theta^*\|^2] + \frac{122\alpha\tau L^2}{\mu}.$$

Lastly, we note that

$$\frac{1}{(1 - \alpha\mu)^\tau} \overset{(i)}{\leq} \frac{1}{1 - \alpha\tau\mu} \overset{(ii)}{\leq} \frac{1}{1 - \alpha\tau L} \overset{(iii)}{\leq} 2, \tag{D.13}$$

where (i) holds by the Bernoulli inequality, that $(1 + x)^r \geq 1 + rx$ for $x \geq -1$ and $r \geq 1$; (ii) holds for $\mu \leq L$; (iii) holds for $\alpha\tau L < \mu/(114L) < \frac{1}{2}$.

Hence, for $k \geq \tau$, we have

$$\mathbb{E}[\|\theta_k - \theta^*\|^2] \leq c_{2,1}(1 - \alpha\mu)^k \|\theta_0 - \theta^*\|^2 + c_{2,2}\alpha\tau_\alpha \frac{L^2}{\mu},$$

for $c_{2,1}$ and $c_{2,2}$ some universal constants. As such, we have completed the proof of base case for Proposition 4.2.

### D.1.1  Proof of Lemma D.1

In this section, we provide the proofs of the four technical inequalities in Lemma D.1.

**Proof of** (D.7).

$$\mathbb{E}[\|\theta_k - \theta_{k-\tau}\| \,|\, \mathcal{F}_{k-\tau}] \leq 2\alpha\tau L \|\theta_{k-\tau} - \theta^*\| + 2\alpha\tau L.$$

*Proof.* Note that

$$\|\theta_k - \theta_{k-\tau}\| \leq \sum_{t=k-\tau}^{k-1} \|\theta_{t+1} - \theta_t\|,$$

so we start with analyzing $\|\theta_{t+1} - \theta_t\|$.

$$
\begin{aligned}
\|\theta_{t+1} - \theta^*\| - \|\theta_t - \theta^*\| &\leq \|\theta_{t+1} - \theta_t\| = \alpha\|g(\theta_t, x_t) + \xi_{t+1}(\theta_t)\| \\
&\leq \alpha\|g(\theta_t, x_t)\| + \alpha\|\xi_{t+1}(\theta_t)\| \leq \alpha L_1(\|\theta_t - \theta^*\| + 1) + \alpha\|\xi_{t+1}(\theta_t)\| \\
\|\theta_{t+1} - \theta^*\| &\leq (1 + \alpha L_1)\|\theta_t - \theta^*\| + \alpha L_1 + \alpha\|\xi_{t+1}(\theta_t)\|.
\end{aligned}
$$

Recall that we assume

$$\mathbb{E}^{1/2}[\|\xi_{t+1}(\theta_t)\|^2 \,|\, \mathcal{F}_t] \leq L_2(\|\theta_t\| + 1),$$

then we have for $k - \tau \leq t \leq k$,

$$
\begin{aligned}
\mathbb{E}[\|\xi_{t+1}(\theta_t)\| \,|\, \mathcal{F}_{k-\tau}] &= \mathbb{E}[\mathbb{E}[\|\xi_{t+1}(\theta_k)\| \,|\, \mathcal{F}_t] \,|\, \mathcal{F}_{k-\tau}] \leq \mathbb{E}[L_2(\|\theta_k\| + 1) \,|\, \mathcal{F}_{k-\tau}] \\
\mathbb{E}[\|\theta_{t+1} - \theta_t\| \,|\, \mathcal{F}_{k-\tau}] &\leq \alpha L(\mathbb{E}[\|\theta_t - \theta^*\| \,|\, \mathcal{F}_{k-\tau}] + 1) \\
\mathbb{E}[\|\theta_{k+1} - \theta^*\| \,|\, \mathcal{F}_{k-\tau}] &\leq (1 + \alpha L)\mathbb{E}[\|\theta_k - \theta^*\| \,|\, \mathcal{F}_{k-\tau}] + \alpha L.
\end{aligned}
$$

Hence, for $0 \leq n \leq \tau$,

$$
\begin{aligned}
\mathbb{E}[\|\theta_{k-\tau+n} - \theta^*\| \,|\, \mathcal{F}_{k-\tau}] &\leq (1 + \alpha L)^n \mathbb{E}[\|\theta_{k-\tau} - \theta^*\| \,|\, \mathcal{F}_{k-\tau}] + \alpha L \sum_{l=0}^{n-1} (1 + \alpha L)^l \\
&= (1 + \alpha L)^n \|\theta_{k-\tau} - \theta^*\| + ((1 + \alpha L)^n - 1).
\end{aligned}
$$

We next note that

$$(1 + x)^y = e^{y\log(1+x)} \leq e^{xy} \leq 1 + 2xy, \quad xy \in [0, 1/2].$$

Hence, at this stage, if we require $\alpha\tau L < \mu/(4L) < 1/4$, we have the following upper bound

$$(1 + \alpha L)^n \leq (1 + \alpha L)^\tau \leq 1 + 2\alpha\tau L \leq 2.$$

Therefore, for $0 \leq n \leq \tau$,

$$
\begin{aligned}
\mathbb{E}[\|\theta_{k-\tau+n} - \theta^*\| \,|\, \mathcal{F}_{k-\tau}] &\leq (1 + 2\alpha\tau L)\|\theta_{k-\tau} - \theta^*\| + 2\alpha\tau L \\
&\leq 2\|\theta_{k-\tau} - \theta^*\| + 2\alpha\tau L.
\end{aligned}
$$

As such, we have

$$
\begin{aligned}
\mathbb{E}[\|\theta_k - \theta_{k-\tau}\| \,|\, \mathcal{F}_{k-\tau}] &\leq \sum_{t=k-\tau}^{k-1} \mathbb{E}[\|\theta_{t+1} - \theta_t\| \,|\, \mathcal{F}_{k-\tau}] \\
&\leq \alpha L \sum_{t=k-\tau}^{k-1} \mathbb{E}[\|\theta_t - \theta^*\| \,|\, \mathcal{F}_{k-\tau}] + \alpha\tau L \\
&\leq \alpha\tau L(2\|\theta_{k-\tau} - \theta^*\| + 2\alpha\tau L) + \alpha\tau L \\
&\leq 2\alpha\tau L \|\theta_{k-\tau} - \theta^*\| + 2\alpha\tau L,
\end{aligned}
$$

and prove the desired inequality. $\qquad\square$

**Proof of (D.8).**

$$\mathbb{E}[\|\theta_k - \theta_{k-\tau}\| | \mathcal{F}_{k-\tau}] \leq 4\alpha\tau L \mathbb{E}[\|\theta_k - \theta^*\| | \mathcal{F}_{k-\tau}] + 4\alpha\tau L.$$

*Proof.* We prove this inequality based on the claim that we have just shown,

$$\mathbb{E}[\|\theta_k - \theta_{k-\tau}\| | \mathcal{F}_{k-\tau}] \leq 2\alpha\tau L \|\theta_{k-\tau} - \theta^*\| + 2\alpha\tau L.$$

We simply note that

$$
\begin{aligned}
\|\theta_{k-\tau} - \theta^*\| &= \mathbb{E}[\|\theta_{k-\tau} - \theta^*\| | \mathcal{F}_{k-\tau}] \\
&\leq \mathbb{E}[\|\theta_k - \theta_{k-\tau}\| | \mathcal{F}_{k-\tau}] + \mathbb{E}[\|\theta_k - \theta^*\| | \mathcal{F}_{k-\tau}].
\end{aligned}
$$

Hence,

$$\mathbb{E}[\|\theta_k - \theta_{k-\tau}\| | \mathcal{F}_{k-\tau}] \leq 2\alpha\tau L(\mathbb{E}[\|\theta_k - \theta_{k-\tau}\| | \mathcal{F}_{k-\tau}] + \mathbb{E}[\|\theta_k - \theta^*\| | \mathcal{F}_{k-\tau}] + 1)$$
$$(1 - 2\alpha\tau L)\mathbb{E}[\|\theta_k - \theta_{k-\tau}\| | \mathcal{F}_{k-\tau}] \leq 2\alpha\tau L \mathbb{E}[\|\theta_k - \theta^*\| | \mathcal{F}_{k-\tau}] + 2\alpha\tau L.$$

Therefore, we obtain

$$\mathbb{E}[\|\theta_k - \theta_{k-\tau}\| | \mathcal{F}_{k-\tau}] \leq 4\alpha\tau L \mathbb{E}[\|\theta_k - \theta^*\| | \mathcal{F}_{k-\tau}] + 4\alpha\tau L.$$

$\square$

**Proof of (D.9).**

$$\mathbb{E}[\|\theta_k - \theta_{k-\tau}\|^2 | \mathcal{F}_{k-\tau}] \leq 32\alpha^2\tau^2 L^2 \mathbb{E}[\|\theta_k - \theta^*\|^2 | \mathcal{F}_{k-\tau}] + 32\alpha^2\tau^2 L^2.$$

*Proof.* To analyze $\mathbb{E}[\|\theta_k - \theta_{k-\tau}\|^2 | \mathcal{F}_{k-\tau}]$, we consider the following attempt.

$$
\begin{aligned}
\mathbb{E}[\|\theta_k - \theta_{k-\tau}\|^2 | \mathcal{F}_{k-\tau}] &\leq \tau \sum_{t=k-\tau}^{k-1} \mathbb{E}[\|\theta_{t+1} - \theta_t\|^2 | \mathcal{F}_{k-\tau}] \\
&= \alpha^2\tau \sum_{t=k-\tau}^{k-1} \mathbb{E}[(\|g(\theta_t, x_t)\| + \|\xi_{t+1}(\theta_t)\|)^2 | \mathcal{F}_{k-\tau}] \\
&\leq 2\alpha^2\tau \sum_{t=k-\tau}^{k-1} \Big( \mathbb{E}[\|g(\theta_t, x_t)\|^2 | \mathcal{F}_{k-\tau}] + \mathbb{E}[\|\xi_{t+1}(\theta_t)\|^2 | \mathcal{F}_{k-\tau}] \Big) \\
&\leq 2\alpha^2\tau L^2 \sum_{t=k-\tau}^{k-1} \mathbb{E}[(\|\theta_t - \theta^*\| + 1)^2 | \mathcal{F}_{k-\tau}] \\
&\leq 4\alpha^2\tau L^2 \sum_{t=k-\tau}^{k-1} \mathbb{E}[\|\theta_t - \theta^*\|^2 | \mathcal{F}_{k-\tau}] + 4\alpha^2\tau^2 L^2.
\end{aligned}
$$

Next, we study $\mathbb{E}[\|\theta_t - \theta^*\|^2 | \mathcal{F}_{k-\tau}]$. We start with the following, for $k - \tau \leq t < k$,

$$
\begin{aligned}
&\mathbb{E}[\|\theta_{t+1} - \theta^*\|^2 | \mathcal{F}_{k-\tau}] \\
&= \mathbb{E}[\|\theta_t - \theta^*\|^2 | \mathcal{F}_{k-\tau}] + 2\alpha \mathbb{E}[\langle \theta_t - \theta^*, g(\theta_t, x_t) \rangle | \mathcal{F}_{k-\tau}] + \alpha^2 \mathbb{E}[\|g(\theta_t, x_t) + \xi_{t+1}(\theta_t)\|^2 | \mathcal{F}_{k-\tau}] \\
&\leq \mathbb{E}[\|\theta_t - \theta^*\|^2 | \mathcal{F}_{k-\tau}] + 2\alpha^2 \Big( \mathbb{E}[\|g(\theta_t, x_t)\|^2 | \mathcal{F}_{k-\tau}] + \mathbb{E}[\|\xi_{t+1}(\theta_t)\|^2 | \mathcal{F}_{k-\tau}] \Big) \\
&\quad + 2\alpha \mathbb{E}[\|\theta_t - \theta^*\| \|g(\theta_t, x_t)\| | \mathcal{F}_{k-\tau}].
\end{aligned}
$$

We note that

$$
\begin{aligned}
2\mathbb{E}[\|\theta_t - \theta^*\| \|g(\theta_t, x_t)\| | \mathcal{F}_{k-\tau}] &\leq 2\sqrt{\mathbb{E}[\|\theta_t - \theta^*\|^2 | \mathcal{F}_{k-\tau}] \mathbb{E}[\|g(\theta_t, x_t)\|^2 | \mathcal{F}_{k-\tau}]} \\
&\leq 2\sqrt{\mathbb{E}[\|\theta_t - \theta^*\|^2 | \mathcal{F}_{k-\tau}] \mathbb{E}[(L(\|\theta_t - \theta^*\| + 1))^2 | \mathcal{F}_{k-\tau}]} \\
&\leq 2\sqrt{\mathbb{E}[\|\theta_t - \theta^*\|^2 | \mathcal{F}_{k-\tau}] 2L^2 \mathbb{E}[\|\theta_t - \theta^*\|^2 + 1 | \mathcal{F}_{k-\tau}]}
\end{aligned}
$$

$$\leq 4L(\mathbb{E}[\|\theta_t - \theta^*\|^2 | \mathcal{F}_{k-\tau}] + 1).$$

Substituting the above inequality back, we obtain

$$\mathbb{E}[\|\theta_{t+1} - \theta^*\|^2 | \mathcal{F}_{k-\tau}]$$
$$\leq \mathbb{E}[\|\theta_t - \theta^*\|^2 | \mathcal{F}_{k-\tau}] + 4\alpha^2 L^2 \Big(\mathbb{E}[\|\theta_t - \theta^*\|^2 | \mathcal{F}_{k-\tau}] + 1\Big) + 4\alpha L \mathbb{E}[\|\theta_t - \theta^*\|^2 | \mathcal{F}_{k-\tau}] + 4\alpha L$$
$$\leq (1 + 4\alpha^2 L^2 + 4\alpha L)\mathbb{E}[\|\theta_t - \theta^*\|^2 | \mathcal{F}_{k-\tau}] + (4\alpha^2 L^2 + 4\alpha L).$$

We further recall that

$$4\alpha^2 L^2 \leq 4\alpha L(\alpha \tau L) \leq \alpha L,$$

and hence we obtain the following upper bound

$$\mathbb{E}[\|\theta_{t+1} - \theta^*\|^2 | \mathcal{F}_{k-\tau}] \leq (1 + 5\alpha L)\mathbb{E}[\|\theta_t - \theta^*\|^2 | \mathcal{F}_{k-\tau}] + 5\alpha L.$$

Then, recursively, for $0 \leq n \leq \tau$, we have

$$\mathbb{E}[\|\theta_{k-\tau+n} - \theta^*\|^2 | \mathcal{F}_{k-\tau}] \leq (1 + 5\alpha L)^n \|\theta_{k-\tau} - \theta^*\|^2 + 5\alpha L \sum_{l=0}^{n-1} (1 + 5\alpha L)^l.$$

As such, under the assumption that $4\alpha\tau L < \mu/(4L) < 1/4$, then for $k - \tau \leq t \leq k$, we have

$$\mathbb{E}[\|\theta_t - \theta^*\|^2 | \mathcal{F}_{k-\tau}] \leq (1 + 10\alpha\tau L)\|\theta_{k-\tau} - \theta^*\|^2 + 10\alpha\tau L$$
$$\leq 2\|\theta_{k-\tau} - \theta^*\|^2 + 10\alpha\tau L.$$

Combining all the analyses above, we have

$$\mathbb{E}[\|\theta_k - \theta_{k-\tau}\|^2 | \mathcal{F}_{k-\tau}] \leq 4\alpha^2\tau L^2 \sum_{t=k-\tau}^{k-1} \mathbb{E}[\|\theta_t - \theta^*\|^2 | \mathcal{F}_{k-\tau}] + 4\alpha^2\tau^2 L^2$$
$$\leq 4\alpha^2\tau^2 L^2 \Big(2\|\theta_{k-\tau} - \theta^*\|^2 + 10\alpha\tau L\Big) + 4\alpha^2\tau^2 L^2$$
$$\leq 8\alpha^2\tau^2 L^2 \|\theta_{k-\tau} - \theta^*\|^2 + 8\alpha^2\tau^2 L^2.$$

$\square$

**Proof of (D.10).**

$$\mathbb{E}[\|\theta_k - \theta_{k-\tau}\|^2 | \mathcal{F}_{k-\tau}] \leq 32\alpha^2\tau^2 L^2 \mathbb{E}[\|\theta_k - \theta^*\|^2 | \mathcal{F}_{k-\tau}] + 32\alpha^2\tau^2 L^2.$$

*Proof.* This inequality simply extends the result from (D.9), i.e.,

$$\mathbb{E}[\|\theta_k - \theta_{k-\tau}\|^2 | \mathcal{F}_{k-\tau}] \leq 8\alpha^2\tau^2 L^2 \|\theta_{k-\tau} - \theta^*\|^2 + 8\alpha^2\tau^2 L^2.$$

We first note that

$$\|\theta_{k-\tau} - \theta^*\|^2 = \mathbb{E}[\|\theta_{k-\tau} - \theta^*\|^2 | \mathcal{F}_{k-\tau}]$$
$$\leq 2\mathbb{E}[\|\theta_k - \theta_{k-\tau}\|^2 | \mathcal{F}_{k-\tau}] + 2\mathbb{E}[\|\theta_k - \theta^*\|^2 | \mathcal{F}_{k-\tau}].$$

Hence,

$$\mathbb{E}[\|\theta_k - \theta_{k-\tau}\|^2 | \mathcal{F}_{k-\tau}] \leq 8\alpha^2\tau^2 L^2 (2\mathbb{E}[\|\theta_k - \theta_{k-\tau}\|^2 | \mathcal{F}_{k-\tau}] + 2\mathbb{E}[\|\theta_k - \theta^*\|^2 | \mathcal{F}_{k-\tau}] + 1)$$
$$(1 - 16\alpha^2\tau^2 L^2)\mathbb{E}[\|\theta_k - \theta_{k-\tau}\|^2 | \mathcal{F}_{k-\tau}] \leq 16\alpha^2\tau^2 L^2 \mathbb{E}[\|\theta_k - \theta^*\|^2 | \mathcal{F}_{k-\tau}] + 8\alpha^2\tau^2 L^2.$$

Again, under the assumption that $16\alpha\tau L^2 < \mu/4$, we can conclude that

$$\mathbb{E}[\|\theta_k - \theta_{k-\tau}\|^2 | \mathcal{F}_{k-\tau}] \leq 32\alpha^2\tau^2 L^2 \mathbb{E}[\|\theta_k - \theta^*\|^2 | \mathcal{F}_{k-\tau}] + 32\alpha^2\tau^2 L^2.$$

$\square$

## D.2 Induction Step

In this step, assume that the moment bound in Proposition 4.2 has been proven for $k \leq n - 1$, we now proceed to show that the desired moment convergence holds for $n$ with $2 \leq n \leq p$.

We start with the following decomposition of $\|\theta_{k+1} - \theta^*\|^{2n}$

$$\|\theta_{k+1} - \theta^*\|^{2n}$$
$$= \left( \|\theta_k - \theta^*\|^2 + 2\alpha\langle\theta_k - \theta^*, g(\theta_k, x_k) + \xi_{k+1}(\theta_k)\rangle + \alpha^2\|g(\theta_k, x_k) + \xi_{k+1}(\theta_t)\|^2 \right)^n$$
$$= \sum_{\substack{i,j,l \\ i+j+l=n}} \binom{n}{i,j,l} \|\theta_k - \theta^*\|^{2i} \left( 2\alpha\langle\theta_k - \theta^*, g(\theta_k, x_k) + \xi_{k+1}(\theta_k)\rangle \right)^j \left( \alpha\|g(\theta_k, x_k) + \xi_{k+1}(\theta_k)\| \right)^{2l}$$

We note the following cases.

1. $i = n$, $j = l = 0$. In this case, the summand is simply $\|\theta_k - \theta^*\|^{2i}$.

2. When $i = n - 1$, $j = 1$ and $l = 0$. In this case, the summand is of order $\alpha$, i.e., $\alpha 2n\langle\theta_k - \theta^*, g(\theta_k, x_k) + \xi_{k+1}(\theta_k)\rangle^j \|\theta_k - \theta^*\|^{2(n-1)}$. We can further compose it as

$$2n\alpha\langle\theta_k - \theta^*, g(\theta_k, x_k) + \xi_{k+1}(\theta_k)\rangle\|\theta_k - \theta^*\|^{2(n-1)}$$
$$= \underbrace{2n\alpha\langle\theta_k - \theta^*, g(\theta_k, x_k) - \bar{g}(\theta_k) + \xi_{k+1}(\theta_k)\rangle\|\theta_k - \theta^*\|^{2(n-1)}}_{T_1}$$
$$+ \underbrace{2n\alpha\langle\theta_k - \theta^*, \bar{g}(\theta_k)\rangle\|\theta_k - \theta^*\|^{2(n-1)}}_{T_2}.$$

Note that, when $(x_k)$ is i.i.d. or from a martingale noise sequence, we have

$$\mathbb{E}[T_1|\theta_k] = 0.$$

However, when $(x_k)$ is Markovian, the above equality then does not hold and $T_1$ requires a careful analysis.

Nonetheless, under the strong monotonicity assumption, we have

$$T_2 \leq -2n\alpha\mu\|\theta_k - \theta^*\|^{2n}.$$

3. For the remaining terms, we see that they are of higher orders of $\alpha$. Therefore, when $\alpha$ is selected sufficiently small, these terms do not raise concern.

Therefore, to prove the desired moment bound, we spend the remaining section analyzing $T_1$. Immediately, we note that

$$\mathbb{E}[T_1|\mathcal{F}_{k-\tau}] = \mathbb{E}\left[ 2n\alpha\langle\theta_k - \theta^*, g(\theta_k, x_k) - \bar{g}(\theta_k) + \mathbb{E}[\xi_{k+1}(\theta_k)|\theta_k]\rangle\|\theta_k - \theta^*\|^{2(n-1)}|\mathcal{F}_{k-\tau} \right]$$
$$= \mathbb{E}\left[ \underbrace{2n\alpha\langle\theta_k - \theta^*, g(\theta_k, x_k) - \bar{g}(\theta_k)\rangle\|\theta_k - \theta^*\|^{2(n-1)}}_{T_1'}|\mathcal{F}_{k-\tau} \right].$$

Subsequently, we focus on analyzing $T_1'$.

We start with the following decomposition of $T_1'$.

$$2n\alpha\langle g(\theta_k, x_k) - \bar{g}(\theta_k), \theta_k - \theta^*\rangle\|\theta_k - \theta^*\|^{2(n-1)}$$
$$\leq 2n\alpha\|g(\theta_{k-\tau}, x_k) - \bar{g}(\theta_{k-\tau})\|\|\theta_{k-\tau} - \theta^*\|^{2n-1} \tag{D.14}$$
$$+ 2n\alpha\|g(\theta_k, x_k) - g(\theta_{k-\tau}, x_k)\|\|\theta_{k-\tau} - \theta^*\|^{2n-1} \tag{D.15}$$
$$+ 2n\alpha\|\bar{g}(\theta_{k-\tau}) - \bar{g}(\theta_k)\|\|\theta_{k-\tau} - \theta^*\|^{2n-1} \tag{D.16}$$
$$+ 2n\alpha\|g(\theta_k, x_k) - \bar{g}(\theta_k)\|\|\theta_k - \theta^*\|^{2(n-1)}\|\theta_k - \theta_{k-\tau}\| \tag{D.17}$$
$$+ 2n\alpha\|g(\theta_k, x_k) - \bar{g}(\theta_k)\|\|\theta_{k-\tau} - \theta^*\| \cdot \left( \|\theta_k - \theta^*\|^{2(n-1)} - \|\theta_{k-\tau} - \theta^*\|^{2(n-1)} \right). \tag{D.18}$$

We note the following technical lemma, which will offer significant help in the analysis of $T_1'$. We postpone the proof of the lemma to the end of this subsection.

**Lemma D.2.** *For $\tilde{c}_n \alpha \tau \leq \mu/(4L^2)$, where $\tilde{c}_n$ denotes some constant dependent of the higher-moment $2n$, we have*

$$\mathbb{E}[\|\theta_k - \theta_{k-\tau}\|^{2n}|\mathcal{F}_{k-\tau}] \leq c_n \alpha^{2n} \tau^{2n} L^{2n}(\|\theta_{k-\tau} - \theta^*\|^{2n} + 1).$$

Following the lemma, we observe that a natural consequence is for any $m \leq 2n$, we have

$$\mathbb{E}[\|\theta_k - \theta_{k-\tau}\|^m|\mathcal{F}_{k-\tau}] \leq \left( \mathbb{E}[\|\theta_k - \theta_{k-\tau}\|^{2n}|\mathcal{F}_{k-\tau}] \right)^{\frac{m}{2n}}$$

$$\leq \left( c_n \alpha^{2n} \tau^{2n} L^{2n}(\|\theta_{k-\tau} - \theta^*\|^{2n} + 1) \right)^{\frac{m}{2n}}$$

$$\leq c_m \alpha^m \tau^m L^m \left( \|\theta_{k-\tau} - \theta^*\|^m + 1 \right),$$

where we use the inequality $a^p + b^p > (a+b)^p$ for $a, b > 0$, $p \in (0,1)$ to obtain the final inequality.

Now, we are ready to analyze (D.14)–(D.18). Firstly, for (D.14), we make use of the mixing assumption of $\tau$, and have that

$$\mathbb{E}[|(D.14)||\mathcal{F}_{k-\tau}] \leq 2n\alpha\|\theta_{k-\tau} - \theta^*\|^{2n-1}\mathbb{E}[\|g(\theta_{k-\tau}, x_k) - \bar{g}(\theta_{k-\tau})\||\mathcal{F}_{k-\tau}]$$

$$\leq 2n\alpha^2 L\|\theta_{k-\tau} - \theta^*\|^{2n-1}(\|\theta_{k-\tau} - \theta^*\| + 1)$$

$$\leq 2n\alpha^2 L\|\theta_{k-\tau} - \theta^*\|^{2n} + 2n\alpha^2 L\|\theta_{k-\tau} - \theta^*\|^{2n-1}$$

$$\leq 3n\alpha^2 L\|\theta_{k-\tau} - \theta^*\|^{2n} + n\alpha^2 L\|\theta_{k-\tau} - \theta^*\|^{2(n-1)},$$

where we make use of the inequality $2|x|^3 \leq x^2 + x^4$ to obtain the final step.

Next, we proceed to analyze (D.15). It is easy to see that

$$\mathbb{E}[|(D.15)||\mathcal{F}_{k-\tau}] = 2n\alpha\|\theta_{k-\tau} - \theta^*\|^{2n-1}\mathbb{E}[\|g(\theta_k, x_k) - g(\theta_{k-\tau}, x_k)\||\mathcal{F}_{k-\tau}]$$

$$\leq 2n\alpha\|\theta_{k-\tau} - \theta^*\|^{2n-1}\mathbb{E}[\|\theta_k - \theta_{k-\tau}\||\mathcal{F}_{k-\tau}]$$

$$\leq 2n\alpha\|\theta_{k-\tau} - \theta^*\|^{2n-1}\left( 2\alpha\tau L(\|\theta_{k-\tau} - \theta^*\| + 1) \right)$$

$$\leq 4n\alpha^2 \tau L\|\theta_{k-\tau} - \theta^*\|^{2n} + 4n\alpha^2 \tau L\|\theta_{k-\tau} - \theta^*\|^{2n-1}$$

$$\leq 6n\alpha^2 \tau L\|\theta_{k-\tau} - \theta^*\|^{2n} + 2n\alpha^2 \tau L\|\theta_{k-\tau} - \theta^*\|^{2(n-1)}.$$

The term in (D.16) can be analyzed in a similar fashion as the (D.15).

For (D.17), we first derive the following

$$\mathbb{E}[|(D.17)||\mathcal{F}_{k-\tau}]$$

$$\leq 2n\alpha\mathbb{E}\Big[ 2L\big(\|\theta_k - \theta^*\| + 1\big)\|\theta_k - \theta_{k-\tau}\|\|\theta_k - \theta^*\|^{2(n-1)}|\mathcal{F}_{k-\tau}\Big]$$

$$= \underbrace{4n\alpha L\mathbb{E}[\|\theta_k - \theta_{k-\tau}\|\|\theta_k - \theta^*\|^{2n-1}|\mathcal{F}_{k-\tau}]}_{T_a} + \underbrace{4n\alpha L\mathbb{E}[\|\theta_k - \theta_{k-\tau}\|\|\theta_k - \theta^*\|^{2(n-1)}|\mathcal{F}_{k-\tau}]}_{T_b}.$$

We next analyze the two terms $T_a$ and $T_b$ respectively. Starting with $T_a$, we have

$$4n\alpha L\mathbb{E}[\|\theta_k - \theta_{k-\tau}\|\|\theta_k - \theta^*\|^{2n-1}|\mathcal{F}_{k-\tau}] \tag{D.19}$$

$$\leq 4n\alpha L\mathbb{E}\Big[ \|\theta_k - \theta_{k-\tau}\|\big(\|\theta_k - \theta_{k-\tau}\| + \|\theta_{k-\tau} - \theta^*\|\big)^{2n-1}|\mathcal{F}_{k-\tau}\Big] \tag{D.20}$$

$$\leq 2^{2(n-1)}4n\alpha L\mathbb{E}[\|\theta_k - \theta_{k-\tau}\|(\|\theta_k - \theta_{k-\tau}\|^{2n-1} + \|\theta_{k-\tau} - \theta^*\|^{2n-1}|\mathcal{F}_{k-\tau}] \tag{D.21}$$

$$= 4^n n\alpha L\Big( \underbrace{\mathbb{E}[\|\theta_k - \theta_{k-\tau}\|^{2n}|\mathcal{F}_{k-\tau}]}_{\text{by Lemma D.2}} + \|\theta_{k-\tau} - \theta^*\|^{2n-1}\mathbb{E}[\|\theta_k - \theta_{k-\tau}\||\mathcal{F}_{k-\tau}]\Big) \tag{D.22}$$

$$\leq 4^n n\alpha L\Big( \underbrace{c_n \alpha^{2n} \tau^{2n} L^{2n}}_{\leq 2\alpha\tau L}(\|\theta_{k-\tau} - \theta^*\|^{2n} + 1) + \|\theta_{k-\tau} - \theta^*\|^{2n-1}(2\alpha\tau L(\|\theta_{k-\tau} - \theta^*\| + 1))\Big)$$

$$\tag{D.23}$$

$$\leq 4^n n\alpha L\Big( 4\alpha\tau L\|\theta_{k-\tau} - \theta^*\|^{2n} + 2\alpha\tau L\|\theta_{k-\tau} - \theta^*\|^{2n-1} + c_n \alpha^{2n} \tau^{2n} L^{2n}\Big) \tag{D.24}$$

$$\leq 4^n n\alpha L\Big( 5\alpha\tau L\|\theta_{k-\tau} - \theta^*\|^{2n} + \alpha\tau L\|\theta_{k-\tau} - \theta^*\|^{2(n-1)} + c'_n \alpha^{2n-1} \tau^{2n-1} L^{2n-1}\Big). \tag{D.25}$$

For $T_b$, we have

$$4n\alpha L \mathbb{E}[\|\theta_k - \theta_{k-\tau}\|\|\theta_k - \theta^*\|^{2(n-1)}|\mathcal{F}_{k-\tau}]$$

$$\leq 4n\alpha L \mathbb{E}\Big[\|\theta_k - \theta_{k-\tau}\|\big(\|\theta_k - \theta_{k-\tau}\| + \|\theta_{k-\tau} - \theta^*\|\big)^{2(n-1)}|\mathcal{F}_{k-\tau}\Big]$$

$$\leq 2^{2n-1}n\alpha L \mathbb{E}[\|\theta_k - \theta_{k-\tau}\|(\|\theta_k - \theta_{k-\tau}\|^{2(n-1)} + \|\theta_{k-\tau} - \theta^*\|^{2(n-1)}|\mathcal{F}_{k-\tau}]$$

$$= 2^{2n-1}n\alpha L \Big(\underbrace{\mathbb{E}[\|\theta_k - \theta_{k-\tau}\|^{2n-1}|\mathcal{F}_{k-\tau}]}_{\text{by Lemma D.2}} + \|\theta_{k-\tau} - \theta^*\|^{2(n-1)}\mathbb{E}[\|\theta_k - \theta_{k-\tau}\||\mathcal{F}_{k-\tau}]\Big)$$

$$\leq 2^{2n-1}n\alpha L \Big(\underbrace{c_{n-1}\alpha^{2n-1}\tau^{2n-1}L^{2n-1}}_{\leq 2\alpha\tau L}(\|\theta_{k-\tau} - \theta^*\|^{2n-1} + 1)$$

$$\qquad\qquad + \|\theta_{k-\tau} - \theta^*\|^{2(n-1)}(2\alpha\tau L(\|\theta_{k-\tau} - \theta^*\| + 1))\Big)$$

$$\leq 2^{2n-1}n\alpha L \Big(4\alpha\tau L\|\theta_{k-\tau} - \theta^*\|^{2n-1} + 2\alpha\tau L\|\theta_{k-\tau} - \theta^*\|^{2(n-1)} + c_{n-1}\alpha^{2n-1}\tau^{2n-1}L^{2n-1}\Big)$$

$$\leq 2^{2n-1}n\alpha L \Big(2\alpha\tau L\|\theta_{k-\tau} - \theta^*\|^{2n} + 4\alpha\tau L\|\theta_{k-\tau} - \theta^*\|^{2(n-1)} + c_{n-1}\alpha^{2n-1}\tau^{2n-1}L^{2n-1}\Big).$$

Combining the analyses of the two terms, we get the following upper bound to (D.17)

$$\mathbb{E}[\|(\text{D.17})\||\mathcal{F}_{k-\tau}]$$

$$\leq 4^n n\alpha L \Big(5\alpha\tau L\|\theta_{k-\tau} - \theta^*\|^{2n} + \alpha\tau L\|\theta_{k-\tau} - \theta^*\|^{2(n-1)} + c'_n\alpha^{2n-1}\tau^{2n-1}L^{2n-1}\Big)$$

$$+ 2^{2n-1}n\alpha L \Big(2\alpha\tau L\|\theta_{k-\tau} - \theta^*\|^{2n} + 4\alpha\tau L\|\theta_{k-\tau} - \theta^*\|^{2(n-1)} + c_{n-1}\alpha^{2n-1}\tau^{2n-1}L^{2n-1}\Big)$$

$$= 2^{2n-1}n\alpha L \Big(12\alpha\tau L\|\theta_{k-\tau} - \theta^*\|^{2n} + 6\alpha\tau L\|\theta_{k-\tau} - \theta^*\|^{2(n-1)} + c''_{n-1}\alpha^{2n-1}\tau^{2n-1}L^{2n-1}\Big).$$

Lastly, we analyze (D.18). We first make use of the mean-value theorem, with $a \in [0,1]$, we have

$$\|\theta_k - \theta^*\|^{2(n-1)} - \|\theta_{k-\tau} - \theta^*\|^{2(n-1)}$$

$$= \|\theta_k - \theta_{k-\tau}\| \cdot 2(n-1)\|a(\theta_k - \theta^*) + (1-a)(\theta_{k-\tau} - \theta^*)\|^{2n-3}$$

$$= \|\theta_k - \theta_{k-\tau}\| \cdot 2(n-1)\|a(\theta_k - \theta_{k-\tau}) + \theta_{k-\tau} - \theta^*\|^{2n-3}$$

$$\leq 2^{2n-3}(n-1)\|\theta_k - \theta_{k-\tau}\|\Big(\|\theta_k - \theta_{k-\tau}\|^{2n-3} + \|\theta_{k-\tau} - \theta^*\|^{2n-3}\Big)$$

Substituting the above upper bound back into (D.18), we obtain

$$\mathbb{E}[\|(\text{D.18})\||\mathcal{F}_{k-\tau}]$$

$$\leq 2^{2n-1}n(n-1)\alpha L\|\theta_{k-\tau} - \theta^*\|$$

$$\qquad \mathbb{E}\Big[(\|\theta_k - \theta^*\| + 1)\|\theta_k - \theta_{k-\tau}\|\Big(\|\theta_k - \theta_{k-\tau}\|^{2n-3} + \|\theta_{k-\tau} - \theta^*\|^{2n-3}\Big)|\mathcal{F}_{k-\tau}\Big]$$

$$\leq 2^{2n-1}n(n-1)\alpha L$$

$$\qquad \Big(\|\theta_{k-\tau} - \theta^*\|\mathbb{E}[\|\theta_k - \theta_{k-\tau}\|^{2n-1}|\mathcal{F}_{k-\tau}] + \|\theta_{k-\tau} - \theta^*\|^2\mathbb{E}[\|\theta_k - \theta_{k-\tau}\|^{2n-2}|\mathcal{F}_{k-\tau}]$$

$$\qquad\quad + \|\theta_{k-\tau} - \theta^*\|\mathbb{E}[\|\theta_k - \theta_{k-\tau}\|^{2n-2}|\mathcal{F}_{k-\tau}] + \|\theta_{k-\tau} - \theta^*\|^{2n-2}\mathbb{E}[\|\theta_k - \theta_{k-\tau}\|^2|\mathcal{F}_{k-\tau}]$$

$$\qquad\quad + \|\theta_{k-\tau} - \theta^*\|^{2n-1}\mathbb{E}[\|\theta_k - \theta_{k-\tau}\||\mathcal{F}_{k-\tau}] + \|\theta_{k-\tau} - \theta^*\|^{2n-2}\mathbb{E}[\|\theta_k - \theta_{k-\tau}\||\mathcal{F}_{k-\tau}]\Big)$$

$$\leq 2^{2n-1}n(n-1)\alpha L \Big(c_n\alpha\tau L\|\theta_{k-\tau} - \theta^*\|^{2n} + c_{n-1}\alpha\tau L\|\theta_{k-\tau} - \theta^*\|^{2(n-1)} + c_{n-1}\alpha^{2n-1}\tau^{2n-1}L^{2n-1}\Big).$$

Combining the analyses above, we have the following bound for $T_1$,

$$\mathbb{E}[|T_1||\mathcal{F}_{k-\tau}]$$

$$\leq \mathbb{E}[\|(\text{D.14})\||\mathcal{F}_{k-\tau}] + \mathbb{E}[\|(\text{D.15})\||\mathcal{F}_{k-\tau}] + \mathbb{E}[\|(\text{D.16})\||\mathcal{F}_{k-\tau}]$$

$$\quad + \mathbb{E}[\|(\text{D.17})\||\mathcal{F}_{k-\tau}] + \mathbb{E}[\|(\text{D.18})\||\mathcal{F}_{k-\tau}]$$

$$\leq c_{n,1}\alpha^2\tau L^2\|\theta_{k-\tau} - \theta^*\|^{2n} + c_{n,2}\alpha^2\tau L^2\|\theta_{k-\tau} - \theta^*\|^{2(n-1)} + c_{n,3}\alpha^{2n}\tau^{2n-1}L^{2n},$$

where $c_{n,1}$, $c_{n,2}$ and $c_{n,3}$ are some constants that depend on $n$.

Additionally, we note that

$$\|\theta_{k-\tau} - \theta^*\|^{2n} = \mathbb{E}[\|\theta_{k-\tau} - \theta^*\|^{2n}|\mathcal{F}_{k-\tau}]$$
$$\leq \mathbb{E}\Big[\Big(\|\theta_k - \theta_{k-\tau}\| + \|\theta_k - \theta^*\|\Big)^{2n}|\mathcal{F}_{k-\tau}\Big]$$
$$\leq 2^{2n-1}\mathbb{E}[\|\theta_k - \theta_{k-\tau}\|^{2n}|\mathcal{F}_{k-\tau}] + 2^{2n-1}\mathbb{E}[\|\theta_k - \theta^*\|^{2n}|\mathcal{F}_{k-\tau}]$$
$$\leq c_n\alpha^{2n}\tau^{2n}L^{2n}(\|\theta_{k-\tau} - \theta^*\|^{2n} + 1) + 2^{2n-1}\mathbb{E}[\|\theta_k - \theta^*\|^{2n}|\mathcal{F}_{k-\tau}].$$

Therefore, for sufficiently small $\alpha\tau L < \mu/(c_n' L)$, we have

$$(1 - c_n'\alpha^{2n}\tau^{2n}L^{2n})\|\theta_{k-\tau} - \theta^*\|^{2n} \leq c_n''\mathbb{E}[\|\theta_k - \theta^*\|^{2n}|\mathcal{F}_{k-\tau}] + c_n\alpha^{2n}\tau^{2n}L^{2n}$$
$$\Rightarrow \quad \|\theta_{k-\tau} - \theta^*\|^{2n} \leq 2c_n''\mathbb{E}[\|\theta_k - \theta^*\|^{2n}|\mathcal{F}_{k-\tau}] + 2c_n\alpha^{2n}\tau^{2n}L^{2n}.$$

As such, for sufficiently small $\alpha$, we have

$\mathbb{E}[|T_1||\mathcal{F}_{k-\tau}]$

$\leq c_{n,1}\alpha^2\tau L^2\Big(c_n\mathbb{E}[\|\theta_k - \theta^*\|^{2n}|\mathcal{F}_{k-\tau}] + c_n'\alpha^{2n}\tau^{2n}L^{2n}\Big)$

$+ c_{n,2}\alpha^2\tau L^2\Big(c_{n-1}\mathbb{E}[\|\theta_k - \theta^*\|^{2(n-1)}|\mathcal{F}_{k-\tau}] + c_{n-1}'\alpha^{2(n-1)}\tau^{2(n-1)}L^{2(n-1)}\Big) + c_{n,3}\alpha^{2n}\tau^{2n-1}L^{2n}$

$= c_{n,1}\alpha^2\tau L^2\mathbb{E}[\|\theta_k - \theta^*\|^{2n}|\mathcal{F}_{k-\tau}] + c_{n,2}\alpha^2\tau L^2\mathbb{E}[\|\theta_k - \theta^*\|^{2(n-1)}|\mathcal{F}_{k-\tau}] + c_{n,3}\alpha^{2n}\tau^{2n-1}L^{2n}.$

Hence, up til this point, we have obtained

$\mathbb{E}[\|\theta_{k+1} - \theta^*\|^{2n}|\mathcal{F}_{k-\tau}]$

$\leq (1 - 2n\alpha\mu)\mathbb{E}[\|\theta_k - \theta^*\|^{2n}|\mathcal{F}_{k-\tau}]$

$+ c_{n,1}\alpha^2\tau L^2\mathbb{E}[\|\theta_k - \theta^*\|^{2n}|\mathcal{F}_{k-\tau}] + c_{n,2}\alpha^2\tau L^2\mathbb{E}[\|\theta_k - \theta^*\|^{2(n-1)}|\mathcal{F}_{k-\tau}] + c_{n,3}\alpha^{2n}\tau^{2n-1}L^{2n}$

$\leq (1 - 2n\alpha(\mu - c_{n,1}'\alpha\tau L^2))\mathbb{E}[\|\theta_k - \theta^*\|^{2n}|\mathcal{F}_{k-\tau}]$

$+ c_{n,2}\alpha^2\tau L^2\mathbb{E}[\|\theta_k - \theta^*\|^{2(n-1)}]|\mathcal{F}_{k-\tau}] + c_{n,3}\alpha^{2n}\tau^{2n-1}L^{2n}.$

Following the induction hypothesis, when $k$ is sufficiently large, we have

$$\mathbb{E}[\|\theta_k - \theta^*\|^{2(n-1)}|\mathcal{F}_{k-\tau}] \leq c_{n-1}\alpha^{n-1}\tau^{n-1}s(\theta_0, L, \mu).$$

Substituting the above upper bound back into our analysis of the $2n$-th moment bound, we obtain

$$\mathbb{E}[\|\theta_{k+1} - \theta^*\|^{2n}|\mathcal{F}_{k-\tau}] \leq (1 - 2n\alpha(\mu - c_{n,1}'\alpha\tau L^2))\mathbb{E}[\|\theta_k - \theta^*\|^{2n}|\mathcal{F}_{k-\tau}]$$
$$+ \alpha^{n+1}\tau^n L^2 c_{n,2} \cdot c_{n-1}s(\theta_0, L, \mu) + c_{n,3}\alpha^{2n}\tau^{2n-1}L^{2n}.$$

Subsequently, if we set $\alpha$ sufficiently small, such that

$$\alpha\tau L^2 < c_n \cdot \mu,$$

we obtain

$$\mathbb{E}[\|\theta_{k+1} - \theta^*\|^{2n}|\mathcal{F}_{k-\tau}] \leq (1 - \alpha\mu)\mathbb{E}[\|\theta_k - \theta^*\|^{2n}|\mathcal{F}_{k-\tau}] + \alpha^{n+1}\tau^n c_{n,2}' \cdot s(\theta_0, L, \mu),$$

where $s(\theta_0, L, \mu)$ is some constant that may depend on the initialization $\theta_0$ and the problem primitives $\mu$ and $L$ but is independent of $\alpha$.

Recursively, we get

$$\mathbb{E}[\|\theta_k - \theta^*\|^{2n}] \leq (1 - \alpha\mu)^{k-\tau}\mathbb{E}[\|\theta_\tau - \theta^*\|^{2n}] + \alpha^n\tau^n \cdot s(\theta_0, L, \mu).$$

Lastly, we recall that

$$\mathbb{E}[\|\theta_\tau - \theta^*\|^{2n}] \leq 2^{2n-1}\mathbb{E}[\|\theta_\tau - \theta_0\|^{2n}] + 2^{2n-1}\mathbb{E}[\|\theta_0 - \theta^*\|^{2n}]$$
$$\leq c_{n,1}\alpha^{2n}\tau^{2n}L^{2n}(\mathbb{E}[\|\theta_0 - \theta^*\|^{2n}] + 1) + c_{n,2}\|\theta_0 - \theta^*\|^{2n}$$
$$\leq c_{n,1}\mathbb{E}[\|\theta_0 - \theta^*\|^{2n}] + c_{n,2}\alpha^{2n}\tau^{2n}L^{2n}.$$

Substituting back, we obtain for sufficiently large $k$,

$$\mathbb{E}[\|\theta_k - \theta^*\|^{2n}] \leq c_{n,1}(1 - \alpha\mu)^{k-\tau}\mathbb{E}[\|\theta_0 - \theta^*\|^{2n}] + \alpha^{2n}\tau^{2n}s(\theta_0, L, \mu).$$

As such, we have proven the desired $n$-th moment bound.

### D.2.1 Proof of Lemma D.2

We now come back to Lemma D.2 and provide the complete proof.

*Proof.* The proof follows a similar strategy as (D.9) and (D.10) in Section D.1.1.
We start with the following relaxation and obtain that

$$
\mathbb{E}[\|\theta_k - \theta_{k-\tau}\|^{2n}|\mathcal{F}_{k-\tau}] \leq \mathbb{E}\Big[\Big(\sum_{t=k-\tau}^{k-1}\|\theta_{t+1} - \theta_t\|\Big)^{2n}|\mathcal{F}_{k-\tau}\Big]
$$

$$
\leq \tau^{2n-1}\sum_{t=k-\tau}^{k-1}\mathbb{E}[\|\theta_{t+1} - \theta_t\|^{2n}|\mathcal{F}_{k-\tau}]
$$

$$
= \alpha^{2n}\tau^{2n-1}\sum_{t=k-\tau}^{k-1}\mathbb{E}[\|g(\theta_t, x_t) + \xi_{t+1}(\theta_t)\|^{2n}|\mathcal{F}_{k-\tau}]
$$

$$
\leq 2^{2n-1}\alpha^{2n}\tau^{2n-1}\sum_{t=k-\tau}^{k-1}\Big(\mathbb{E}[\|g(\theta_t, x_t)\|^{2n}|\mathcal{F}_{k-\tau}] + \mathbb{E}[\|\xi_{t+1}(\theta_t)\|^{2n}|\mathcal{F}_{k-\tau}]\Big)
$$

$$
\leq 2^{2n-1}\alpha^{2n}\tau^{2n-1}\sum_{t=k-\tau}^{k-1}\Big(L_1^{2n}\mathbb{E}[(\|\theta_t - \theta^*\| + 1)^{2n}|\mathcal{F}_{k-\tau}] + L_2^{2n}(\mathbb{E}[\|\theta_t - \theta^*\||\mathcal{F}_{k-\tau}] + 1)^{2n}\Big)
$$

$$
\leq 4^{2n-1}\alpha^{2n}\tau^{2n-1}L^{2n}\sum_{t=k-\tau}^{k-1}\mathbb{E}[\|\theta_t - \theta^*\|^{2n}|\mathcal{F}_{k-\tau}] + 4^{2n-1}\alpha^{2n}\tau^{2n}L^{2n}.
$$

Next, in order to obtain a bound on $\|\theta_t - \theta^*\|^{2n}$, we study the following term.

$$
\mathbb{E}[\|\theta_{t+1} - \theta^*\|^{2n}|\mathcal{F}_{k-\tau}] \leq \mathbb{E}\Big[\Big(\|\theta_{t+1} - \theta_t\| + \|\theta_t - \theta^*\|\Big)^{2n}|\mathcal{F}_{k-\tau}\Big]
$$

$$
= \sum_{i=0}^{2n}\binom{2n}{i}\mathbb{E}[\|\theta_{t+1} - \theta_t\|^i\|\theta_t - \theta^*\|^{2n-i}|\mathcal{F}_{k-\tau}]
$$

$$
= \mathbb{E}[\|\theta_t - \theta^*\|^{2n}|\mathcal{F}_{k-\tau}] + \sum_{i=1}^{2n}\alpha^i\mathbb{E}[\|g(\theta_t, x_t) + \xi_{t+1}(\theta_t)\|^i\|\theta_t - \theta^*\|^{2n-i}|\mathcal{F}_{k-\tau}]
$$

Note that

$$
\mathbb{E}[\|g(\theta_t, x_t) + \xi_{t+1}(\theta_t)\|^i\|\theta_t - \theta^*\|^{2n-i}|\mathcal{F}_{k-\tau}]
$$

$$
\leq 2^{i-1}\mathbb{E}\Big[\Big(\|g(\theta_t, x_t)\|^i + \|\xi_{t+1}(\theta_t)\|^i\Big)\|\theta_t - \theta^*\|^{2n-i}|\mathcal{F}_{k-\tau}\Big]
$$

$$
= 2^{i-1}\mathbb{E}\Big[\mathbb{E}[(\|g(\theta_t, x_t)\|^i + \|\xi_{t+1}(\theta_t)\|^i)|\theta_t]\|\theta_t - \theta^*\|^{2n-i}|\mathcal{F}_{k-\tau}\Big]
$$

$$
\leq 2^{i-1}L^i\mathbb{E}\Big[(\|\theta_t - \theta^*\| + 1)^{2n}|\mathcal{F}_{k-\tau}\Big]
$$

$$
\leq 2^{2(n-1)}2^iL^i\Big(\mathbb{E}[\|\theta_t - \theta^*\||\mathcal{F}_{k-\tau}] + 1\Big)
$$

Substituting back, we obtain

$$
\sum_{i=1}^{2n}\alpha^i\mathbb{E}[\|g(\theta_t, x_t) + \xi_{t+1}(\theta_t)\|^i\|\theta_t - \theta^*\|^{2n-i}|\mathcal{F}_{k-\tau}]
$$

$$
\leq 2^{2(n-1)}\sum_{i=1}^{2n}2^i\alpha^iL^i\Big(\mathbb{E}[\|\theta_t - \theta^*\||\mathcal{F}_{k-\tau}] + 1\Big)
$$

$$
= 2^{2(n-1)}\Big(\mathbb{E}[\|\theta_t - \theta^*\||\mathcal{F}_{k-\tau}] + 1\Big)\cdot 2\alpha L(1 + 2\alpha L)^{2n-1}
$$

$$\leq 4^{2n-1}\alpha L\Big(\mathbb{E}[\|\theta_t - \theta^*\| | \mathcal{F}_{k-\tau}] + 1\Big)$$

Consolidating the terms, we have

$$\mathbb{E}[\|\theta_{t+1} - \theta^*\|^{2n} | \mathcal{F}_{k-\tau}] \leq (1 + 4^{2n-1}\alpha L)\Big(\mathbb{E}[\|\theta_t - \theta^*\| | \mathcal{F}_{k-\tau}] + 1\Big)$$

Recursively, for $0 \leq l \leq \tau$, we have

$$\mathbb{E}[\|\theta_{k-\tau+l} - \theta^*\|^{2n} | \mathcal{F}_{k-\tau}] \leq (1 + 4^{2n-1}\alpha L)^l \|\theta_{k-\tau} - \theta^*\|^{2n} + 4^{2n-1}\alpha L \sum_{i=0}^{l-1}(1 + 4^{2n-1}\alpha L)^i$$

$$= (1 + 4^{2n-1}\alpha L)^l \|\theta_{k-\tau} - \theta^*\|^{2n} + (1 + 4^{2n-1}\alpha L)^l$$

Then, for

$$4^{2n-1}\alpha\tau L \leq \mu/4L < 1/4,$$

we have for $k - \tau \leq t \leq k$,

$$\mathbb{E}[\|\theta_t - \theta^*\|^{2n} | \mathcal{F}_{k-\tau}] \leq (1 + 2^{4n-1}\alpha\tau L)\|\theta_{k-\tau} - \theta^*\|^{2n} + 2^{4n-1}\alpha\tau L$$
$$\leq 2\|\theta_{k-\tau} - \theta^*\|^{2n} + 2^{4n-1}\alpha\tau L$$

Finally, we have

$$\mathbb{E}[\|\theta_k - \theta_{k-\tau}\|^{2n} | \mathcal{F}_{k-\tau}] \leq 4^{2n-1}\alpha^{2n}\tau^{2n-1}L^{2n}\sum_{t=k-\tau}^{k-1}\mathbb{E}[\|\theta_t - \theta^*\|^{2n} | \mathcal{F}_{k-\tau}] + 4^{2n-1}\alpha^{2n}\tau^{2n}L^{2n}$$

$$\leq 4^{2n-1}\alpha^{2n}\tau^{2n}L^{2n}\Big(2\|\theta_{k-\tau} - \theta^*\|^{2n} + 2^{4n-1}\alpha\tau L + 1\Big)$$

$$\leq 2^{4n-1}\alpha^{2n}\tau^{2n}L^{2n}\Big(\|\theta_{k-\tau} - \theta^*\|^{2n} + 1\Big).$$

As such, we have completed the proof. $\qquad\square$

# E  Proof of Theorem 4.1

In this section, we prove the weak convergence result in Theorem 4.1. In fact, the proof of the projected SA weak convergence result can be seen as a special case of unprojected SA with the asymptotic linearity condition, which we have briefly discussed in Section 4. Therefore, the proof proceeds in the following two subsections. First, we formally define the asymptotic linearity condition and present our weak convergence result for unprojected SA under this additional assumption. Next, we relate this result for unprojected SA to projected SA and specialize the proof to obtain Theorem 4.1.

## E.1  Asymptotic Linearity

In this subsection, we formally introduce the asymptotic linearity condition, which is crucial for establishing weak convergence in the context of unprojected SA ($\beta = \infty$). Additionally, we explore the implications of this condition.

**Assumption 6** (Asymptotic Linearity)**.** *The noise sequence $(\xi_k)_{k\geq 1}$ is a collection of i.i.d. random fields satisfying the following conditions: (1) $\mathbb{E}[\xi_{k+1}(\theta)|\mathcal{F}_k] = 0$, (2) there exists a constant $L_3 > 0$ such that $\xi_1$ is $L_3$-Lipschitz, i.e., $\|\xi_1(\theta) - \xi_1(\theta')\| \leq L_3\|\theta - \theta'\|$, for all $\theta, \theta' \in \mathbb{R}^d$, and (3) $\|\xi_1(0)\| \leq L_3$.*

*Moreover, there exists a function $G(\cdot) : \mathcal{X} \to \mathbb{R}^{d\times d}$ such that given $\epsilon > 0$, define*

$$\delta(\epsilon) := \min\big\{\delta : \|g'(\theta, x) - G(x)\| \leq \epsilon, \forall x \in \mathcal{X} \quad and \quad \forall\theta \in \{\theta : \|\theta\| \geq \delta\}\big\},$$

*and we have $\lim_{\epsilon\to 0}\epsilon\delta(\epsilon) = 0$.*

The first part of Assumption 6 states that the random field grows at most linearly in $\theta$. The second part of Assumption 6 implies that $g'(\theta, x)$ converges to a limit $G(x)$ when $\|\theta\| \to \infty$ for all $x \in \mathcal{X}$, which shows the asymptotic linearity of $g(\theta, x)$. Furthermore, Assumption 6 also requires how fast

$g'(\theta, x)$ converges to $G(x)$. A sufficient condition under which the second part of Assumption 6 holds is that there exists $\omega > 0$ such that $\|\theta\|^{1+\omega}\|g'(\theta, x) - G(x)\| < \infty$ for $\forall \theta \in \mathbb{R}^d$ and $x \in \mathcal{X}$. We can verify that to ensure $\|g'(\theta, x) - G(x)\| < \epsilon$, we can set $\|\theta\| \in \Theta(\epsilon^{-\frac{1}{1+\omega}})$, which can ensure $\epsilon\delta(\epsilon) \in \mathcal{O}(\epsilon^{\frac{\omega}{1+\omega}}) \to 0$ as $\epsilon \to 0$. This sufficient condition implies that $g'(\theta, x)$ uniformly converge to $G(x)$ with convergence rate of $\mathcal{O}(\|\theta\|^{-(1+w)})$. By definition, we conclude that the structure of linear SA is also asymptotic linear. Besides that, the 1-dimensional logistic regression also satisfies Assumption 6. For 1-dimensional logistic regression, we have $g(\theta, x, y) = x\left(\frac{1}{1+e^{-\theta x}} - y\right) + \lambda\theta$, where $(x, y)$ presents the data. Therefore, we have $g'(\theta, x, y) = \frac{x^2 e^{-\theta x}}{(1+e^{-\theta x})^2} + \lambda$ and $g'(\theta, x, y)$ uniformly converges to $\lambda$ with geometric convergence rate, thereby satisfies the Assumption 6.

### E.2  Proof Under Assumption 6

With the asymptotic linearity condition now formally defined, we proceed to prove the weak convergence for unprojected SA. For convenient reference, we state the theorem below.

**Theorem E.1** (Ergodicity of SA–Asymptotic Linearity). *Suppose that Assumption 1–Assumption 4 hold. Additionally, assume 6. For stepsize $\alpha > 0$ that satisfies the constraint $\alpha\tau_\alpha L^2 < \min(c_2\mu, \kappa_\mu)$, with $c_2$ formalized in Proposition 4.2 and $\kappa_\mu$ defined in (E.1), the Markov chain $(x_k, \theta_k)_{k \geq 0}$ converges to a unique stationary distribution $\bar{\nu}_\alpha \in \mathcal{P}_2(\mathcal{X} \times \mathbb{R}^d)$.*

*Moreover, there exist $\kappa_\mu > 0$ and some universal constant $c'$ such that*

$$\epsilon\delta(\epsilon) \leq c'\mu, \quad \forall \epsilon \leq \kappa_\mu. \tag{E.1}$$

*We let $\nu_\alpha := \mathcal{L}(\theta_\infty)$ be the second marginal of $\bar{\nu}_\alpha$. For $k \geq 2\tau_\alpha$, it holds that*

$$W_2(\mathcal{L}(\theta_k), \nu_\alpha) \leq \bar{W}_2(\mathcal{L}(x_k, \theta_k), \bar{\nu}_\alpha) \leq (1 - \alpha\mu)^{k/2} \cdot s(\theta_0, L, \mu). \tag{E.2}$$

The proof of Theorem E.1 consists of two major steps. Firstly, we assume that $x_0 \sim \pi$, and show that $(x_k, \theta_k)_{k \geq 0}$ converges to a unique limiting invariant distribution. Next, we relax the assumption of $x_0 \sim \pi$, and prove that for arbitrary initialization $(x_0, \theta_0) \in \mathcal{X} \times \mathbb{R}^d$, the Markov chain will converge to the same limit.

**Step 1: Initialization with $x_0 \sim \pi$.**  To prove the convergence of the Markov chain, we consider the following coupling construction. We have a pair of Markov chains $(x_k, \theta_k^{[1]})_{k \geq 0}$ and $(x_k, \theta_k^{[2]})_{k \geq 0}$ sharing the same underlying process and noise, i.e., $(x_k, \xi_{k+1})_{k \geq 0}$, i.e.,

$$\begin{aligned} \theta_{k+1}^{[1]} &= \theta_k^{[1]} + \alpha(g(\theta_k^{[1]}, x_k) + \xi_{k+1}(\theta_k^{[1]})), \\ \theta_{k+1}^{[2]} &= \theta_k^{[2]} + \alpha(g(\theta_k^{[2]}, x_k) + \xi_{k+1}(\theta_k^{[2]})). \end{aligned} \tag{E.3}$$

We assume that the initial iterates $\theta_0^{[1]}$ and $\theta_0^{[2]}$ may depend on each other and on $x_0$, but are independent of subsequent $(x_k)_{k \geq 1}$ given $x_0$. For the iterates difference $\theta_k^{[1]} - \theta_k^{[2]}$, we have the following Proposition E.2, whose proof is given at the end of this subsection.

**Proposition E.2.** $\forall k \geq \tau$ and $\alpha\tau \leq \min(\frac{\mu}{908L^2}, \frac{\kappa_\mu}{L^2})$, we have

$$\mathbb{E}[\|\theta_k^{[1]} - \theta_k^{[2]}\|^2] \leq 4(1 - \mu\alpha)^{k-\tau}\mathbb{E}[\|\theta_0^{[1]} - \theta_0^{[2]}\|^2],$$

where $\kappa_\mu > 0$ and $\epsilon\delta(\epsilon) \leq \frac{\mu}{768}, \forall \epsilon \leq \kappa_\mu$.

By Proposition E.2 and the definition of $W_2$ and $\bar{W}_2$, we have

$$\begin{aligned} W_2^2\left(\mathcal{L}(\theta_k^{[1]}), \mathcal{L}(\theta_k^{[2]})\right) &\overset{(i)}{\leq} \bar{W}_2^2\left(\mathcal{L}(x_k, \theta_k^{[1]}), \mathcal{L}(x_k, \theta_k^{[2]})\right) \\ &\overset{(ii)}{\leq} \mathbb{E}[\|\theta_k^{[1]} - \theta_k^{[2]}\|^2] \\ &\overset{(iii)}{\leq} 4(1 - \mu\alpha)^{k-\tau}\mathbb{E}[\|\theta_0^{[1]} - \theta_0^{[2]}\|^2], \end{aligned} \tag{E.4}$$

where (i) and (ii) hold by the definition of $W_2$ and $\bar{W}_2$ and (iii) holds by applying Proposition E.2.

Note that equation (E.4) always holds for any joint distribution of initial iterates $(x_0, \theta_0^{[1]}, \theta_0^{[2]})$. Recall that $P^*$ represents the transition kernel for the time-reversed Markov chain of $\{x_k\}_{k\geq 0}$, and the initial distribution of $x_0$ is assumed to be mixed already. Given a specific $x_0$, we sample $x_{-1}$ from $P^*(\cdot \mid x_0)$. Additionally, we use $\theta_{-1}^{[2]}$ to denote the random varible that satisfies $\theta_{-1}^{[2]} \stackrel{d}{=} \theta_0^{[1]}$ and is independent of $\{x_k\}_{k\geq 0}$. Finally, we set $\theta_0^{[2]}$ as

$$\theta_0^{[2]} = \theta_{-1}^{[2]} + \alpha(g(x_{-1}, \theta_{-1}^{[2]}) + \xi_0(\theta_{-1}^{[2]})).$$

By the property of time-reversed Markov chain, we have $\{x_k\}_{k\geq -1} \stackrel{d}{=} \{x_k\}_{k\geq 0}$. Given that $\theta_{-1}^{[2]} \stackrel{d}{=} \theta_0^{[1]}$ and $\theta_{-1}^{[2]}$ is independent with $\{x_k\}_{k\geq -1}$, we can prove $(x_k, \theta_k^{[2]}) \stackrel{d}{=} (x_{k+1}, \theta_{k+1}^{[1]})$ for $k \geq 0$. We thus have for all $k \geq \tau$:

$$\bar{W}_2^2\left(\mathcal{L}\left(x_k, \theta_k^{[1]}\right), \mathcal{L}\left(x_{k+1}, \theta_{k+1}^{[1]}\right)\right) = \bar{W}_2^2\left(\mathcal{L}\left(x_k, \theta_k^{[1]}\right), \mathcal{L}\left(x_k, \theta_k^{[2]}\right)\right)$$
$$\stackrel{(i)}{\leq} 4(1-\mu\alpha)^{k-\tau}\mathbb{E}[\|\theta_0^{[1]} - \theta_0^{[2]}\|^2],$$

where (i) holds by inequality (E.4). Then, we have

$$\sum_{k=0}^{\infty} \bar{W}_2^2\left(\mathcal{L}\left(x_k, \theta_k^{[1]}\right), \mathcal{L}\left(x_{k+1}, \theta_{k+1}^{[1]}\right)\right)$$
$$\leq \sum_{k=0}^{t_\alpha - 1} \bar{W}_2^2\left(\mathcal{L}\left(x_k, \theta_k^{[1]}\right), \mathcal{L}\left(x_{k+1}, \theta_{k+1}^{[1]}\right)\right) + 4\mathbb{E}[\|\theta_0^{[1]} - \theta_0^{[2]}\|^2]\sum_{k=0}^{\infty}(1-\mu\alpha)^k$$
$$< \infty.$$

Consequently, $\{\mathcal{L}(x_k, \theta_k^{[1]})\}_{k\geq 0}$ forms a Cauchy sequence w.r.t. the metric $\bar{W}_2$. Since the space $\mathcal{P}_2(\mathcal{X} \times \mathbb{R}^d)$ endowed with $\bar{W}_2$ is a Polish space, every Cauchy sequence converges [63, Theorem 6.18]. Furthermore, convergence in Wasserstein 2-distance also implies weak convergence [63, Theorem 6.9]. Therefore, we conclude that the sequence $(\mathcal{L}(x_k, \theta_k^{[1]}))_{k\geq 0}$ converges weakly to a limit distribution $\bar{\mu} \in \mathcal{P}_2(\mathcal{X} \times \mathbb{R}^d)$.

Now that we have established the existence of a limiting distribution, we next proceed to show the uniqueness. We prove this by contradiction. Note that we currently assume that $x_0 \sim \pi$, hence to show that the limit $(x_\infty, \theta_\infty)$ is unique, we only need to show that the limit is independent of the initial distribution of $\theta_0$, which can be correlated to $x_0$.

Consider two Markov chains $(x_k, \theta_k)_{k\geq 0}$ and $(x_k, \theta_k')_{k\geq 0}$, sharing $(x_k, \xi_{k+1})_{k\geq 0}$ but with arbitrary initialization of $\theta_0$ and $\theta_0'$. For the sake of contradiction, we assume that $(x_0, \theta_0) \Rightarrow (x_\infty, \theta_\infty)$ and $(x_0, \theta_0') \Rightarrow (x_\infty, \theta_\infty')$ respectively. Then, by the triangle inequality, we have that

$$\bar{W}_2\Big((x_\infty, \theta_\infty), (x_\infty, \theta_\infty')\Big)$$
$$\leq \bar{W}_2\Big((x_\infty, \theta_\infty), (x_k, \theta_k)\Big) + \bar{W}_2\Big((x_k, \theta_k), (x_k, \theta_k')\Big) + \bar{W}_2\Big((x_k, \theta_k'), (x_\infty, \theta_\infty')\Big)$$
$$\to 0.$$

As such, we have shown that the limit $\bar{\nu}$ is unique.

Lastly, we prove that $\bar{\nu}$ is invariant. Suppose that we initialize the joint process at its limit, i.e., $(x_0, \theta_0) \sim \bar{\nu}$. We first apply the triangle inequality, and we obtain

$$\bar{W}_2\Big((x_1, \theta_1), (x_0, \theta_0)\Big) \leq \bar{W}_2\Big((x_1, \theta_1), (x_{k+1}, \theta_{k+1})\Big) + \bar{W}_2\Big((x_{k+1}, \theta_{k+1}), (x_0, \theta_0)\Big).$$

Clearly, as $k \to \infty$, $\bar{W}_2\Big((x_{k+1}, \theta_{k+1}), (x_0, \theta_0)\Big) \to 0$. To bound $\bar{W}_2\Big((x_1, \theta_1), (x_{k+1}, \theta_{k+1})\Big)$, we need the following lemma.

**Lemma E.3.** *Consider two copies of the SA trajectory, where $\mathcal{L}(x_0, \theta_0) = \bar{\nu}$ and $\mathcal{L}(x_0', \theta_0')$ is allowed to be arbitrary.*

$$\bar{W}_2^2\Big(\mathcal{L}(x_1, \theta_1), \mathcal{L}(x_1', \theta_1')\Big) \leq \rho_1 \cdot \bar{W}_2^2\Big(\mathcal{L}(x_0, \theta_0), \mathcal{L}(x_0', \theta_0')\Big) + \rho_2 \cdot \sqrt{\bar{W}_2^2\Big(\mathcal{L}(x_0, \theta_0), \mathcal{L}(x_0', \theta_0')\Big)},$$

*where*

$$\rho_1 := 1 + 2(1 + \alpha L)^2 + 16\alpha^2 L^2 < \infty \quad and \quad \rho_2 := 16\alpha^2 L^2 \sqrt{\mathbb{E}[\|\theta_0\|^4]} < \infty$$

*are independent of* $\mathcal{L}(x_0', \theta_0')$.

*Proof.* Consider the following coupling between the two processes $(x_k, \theta_0)_{k \geq 0}$ and $(x_k', \theta_k')_{k \geq 0}$

$$\bar{W}_2^2\Big(\mathcal{L}(x_0, \theta_0), \mathcal{L}(x_0', \theta_0')\Big) = \mathbb{E}\Big[d_0(x_0, x_0') + \|\theta_0 - \theta_0'\|^2\Big] \quad \text{and}$$
$$x_{k+1} = x_{k+1}' \quad \text{if } x_k = x_k', \quad \forall k \geq 0.$$

Then, it is clear that

$$\bar{W}_2^2\Big(\mathcal{L}(x_1, \theta_1), \mathcal{L}(x_1', \theta_1')\Big) \leq \mathbb{E}\Big[d_0(x_1, x_1') + \|\theta_1 - \theta_1'\|^2\Big].$$

Recall the metric $d_0(x, x') = \mathbb{1}\{x \neq x'\}$ and hence, we have

$$g(\theta_0, x_0) = g(\theta_0, x_0') + d_0(x_0', x_0)(g(\theta_0, x_0) - g(\theta_0, x_0')).$$

Therefore, it is easy to see that

$$\begin{aligned}
\theta_1 - \theta_1' &= \theta_0 - \theta_0' + \alpha(g(\theta_0, x_0) - g(\theta_0', x_0')) + \alpha(\xi_1(\theta_0) - \xi_1(\theta_0')) \\
&= \theta_0 - \theta_0' + \alpha(g(\theta_0, x_0) \mp g(\theta_0, x_0') - g(\theta_0', x_0')) + \alpha(\xi_1(\theta_0) - \xi_1(\theta_0')) \\
&= \theta_0 - \theta_0' + \alpha(g(\theta_0, x_0') - g(\theta_0', x_0')) + \alpha(\xi_1(\theta_0) - \xi_1(\theta_0')) \\
&\quad + \alpha d_0(x_0', x_0)(g(\theta_0, x_0) - g(\theta_0, x_0')),
\end{aligned}$$

whence

$$\begin{aligned}
\|\theta_1 - \theta_1'\| &\leq (1 + \alpha L)\|\theta_0 - \theta_0'\| + \alpha d_0(x_0', x_0)\|g(\theta_0, x_0) - g(\theta_0, x_0')\| \\
&\leq (1 + \alpha L)\|\theta_0 - \theta_0'\| + \alpha d_0(x_0', x_0) \cdot 2L(\|\theta_0\| + 1).
\end{aligned}$$

As such, we see that

$$\begin{aligned}
\mathbb{E}[d_0(x_1, x_1') + \|\theta_1 - \theta_1'\|^2] &\leq \mathbb{E}[d_0(x_0, x_0')] + 2(1 + \alpha L)^2 \cdot \mathbb{E}[\|\theta_0 - \theta_0'\|^2] \\
&\quad + 16\alpha^2 L^2 \cdot \mathbb{E}[d_0(x_0', x_0)(\|\theta_0\|^2 + 1)].
\end{aligned}$$

Next, we make use of Cauchy-Schwarz inequality and obtain

$$\mathbb{E}[d_0(x_0', x_0) \cdot \|\theta_0\|^2] \leq \sqrt{\mathbb{E}[d_0(x_0', x_0)]}\sqrt{\mathbb{E}_{\theta_0 \sim \mu}[\|\theta_0\|^4]}.$$

Because Assumption 6 implies Assumption 4($p = 2$), by Proposition 4.2 and Fatou's lemma, we have

$$\mathbb{E}[\|\theta_\infty - \theta^*\|^4] \leq \liminf_{k \to \infty} \mathbb{E}[\|\theta_k - \theta^*\|_\infty^4] < \infty,$$

which implies $\mathbb{E}[\|\theta_\infty\|^4] < \infty$. Hence, the desired inequality follows through. $\qquad\square$

By Lemma E.3, we can set $\mathcal{L}(x_0', \theta_0') = \mathcal{L}(x_k, \theta_k)$, then

$$\bar{W}_2^2\left(\mathcal{L}(x_1, \theta_1), \mathcal{L}(x_{k+1}, \theta_{k+1})\right) \leq \rho_1 \bar{W}_2^2\left(\bar{\mu}, \mathcal{L}(x_k, \theta_k)\right) + \rho_2\sqrt{\bar{W}_2^2\left(\bar{\nu}, \mathcal{L}(x_k, \theta_k)\right)}.$$

Therefore, $\bar{W}_2^2\left(\mathcal{L}(x_1, \theta_1), \mathcal{L}(x_{k+1}, \theta_{k+1})\right) \to 0$ as $k \to 0$, which implies $\bar{W}_2\Big((x_1, \theta_1), (x_0, \theta_0)\Big) = 0$. As such, we have proved the joint sequence $(x_k, \theta_k)_{k \geq 0}$ converges weakly to the unique invariant distribution $\bar{\nu} \in \mathcal{P}_2(\mathcal{X} \times \mathbb{R}^d)$. As a result, $\{\theta_k\}_{k \geq 0}$ converges weakly to $\mu \in \mathcal{P}_2(\mathbb{R}^d)$, where $\mu$ is the second marginal of $\bar{\mu}$ over $\mathbb{R}^d$.

Lastly, before proceeding to the next step, in which we remove the assumption $x_0 \sim \pi$, we first derive the convergence rate of $\{\theta_k\}_{k \geq 0}$ under $x_0 \sim \pi$ as presented in the following lemma. This lemma will help us to establish the convergence rate without $x_0 \sim \pi$.

**Lemma E.4.** *Under $x_0 \sim \pi$, Assumption 1–4 and 6 and the same setting as Proposition E.2,*

$$W_2^2 \left( \mathcal{L}(\theta_k), \nu_\alpha \right) \leq \bar{W}_2^2 \left( \mathcal{L}(x_k, \theta_k), \bar{\nu}_\alpha \right) \leq 16(1 - \mu\alpha)^k \cdot \left( \mathbb{E} \left[ \|\theta_0^{[1]} - \theta^*\|^2 \right] + c_{2,2}' \right).$$

*Proof.* Let us consider the coupled processes defined as equation (E.3). Suppose that the initial iterate $(x_0, \theta_0^{[2]})$ follows the stationary distribution $\bar{\nu}$, thus $\mathcal{L}(x_k, \theta_k^{[2]}) = \bar{\nu}$ and $\mathcal{L}(\theta_k^{[2]}) = \nu$ for all $k \geq 0$. By equation (E.4), we have for all $k \geq \tau$ :

$$
\begin{aligned}
W_2^2 \left( \mathcal{L}(\theta_k^{[1]}), \mu \right) &= W_2^2 \left( \mathcal{L}(\theta_k^{[1]}), \mathcal{L}(\theta_k^{[2]}) \right) \\
&\leq \bar{W}_2^2 \left( \mathcal{L}(x_k, \theta_k^{[1]}), \mathcal{L}(x_k, \theta_k^{[2]}) \right) \\
&\leq 4(1 - \mu\alpha)^{k-\tau} \mathbb{E}[\|\theta_0^{[1]} - \theta_0^{[2]}\|^2] \\
&\leq 8(1 - \mu\alpha)^{k-\tau} \cdot \left( \mathbb{E} \left[ \|\theta_0^{[1]} - \theta^*\|^2 \right] + \mathbb{E} \left[ \|\theta_\infty - \theta^*\|^2 \right] \right) \\
&\leq 16(1 - \mu\alpha)^k \cdot \left( \mathbb{E} \left[ \|\theta_0^{[1]} - \theta^*\|^2 \right] + \mathbb{E} \left[ \|\theta_\infty - \theta^*\|^2 \right] \right),
\end{aligned}
\tag{E.5}
$$

where we make use of the derivation in (D.13) to obtain the last inequality.

We note that

$$\mathbb{E}[\|\theta_\infty - \theta^*\|^2] \leq \lim_{k \to \infty} \inf \mathbb{E}[\|\theta_k - \theta^*\|_\infty^2] \leq c_{2,2} \cdot \alpha\tau L^2/\mu \leq c_{2,2}',$$

the last inequality holds for $\alpha\tau \leq \min(\frac{\mu}{908 L^2}, \frac{\kappa_\mu}{L^2})$. Therefore, we prove the desired inequality

$$W_2^2 \left( \mathcal{L}(\theta_k^{[1]}), \mu \right) \leq 16(1 - \mu\alpha)^k \cdot \left( \mathbb{E} \left[ \|\theta_0^{[1]} - \theta^*\|^2 \right] + c_{2,2}' \right).$$

$\square$

**Step 2: Arbitrary Initialization for** $(x_0, \theta_0)$. In this step, we remove the assumption of $x_0 \sim \pi$ needed in the previous step. We need the following lemma to prove our result.

**Lemma E.5.** *Consider two trajectories $(x_k, \theta_k)_{k \geq 0}$ and $(x_k', \theta_k')_{k \geq 0}$. Suppose that $\theta_0 = \theta_0'$, $x_0' \sim \pi$ and $x_0$ is initialized from some arbitrary distribution that satisfies $\|\mathcal{L}(x_0) - \pi\|_{\mathrm{TV}} = \epsilon$. Then for $k \geq \tau$, we have*

$$\bar{W}_2(\mathcal{L}(x_k, \theta_k), \mathcal{L}(x_k', \theta_k')) \leq \epsilon \Big( 4c_{2,1}(1 - \alpha\mu)^k \mathbb{E}[\|\theta_0 - \theta^*\|^2] + 4c_{s,2}\alpha\tau \cdot \frac{L^2}{\mu} + 1 \Big)^{1/2}.$$

*Proof.* We consider the following coupling between two joint processes $(x_k, \theta_k)_{k \geq 0}$ and $(x_k', \theta_k')_{k \geq 0}$. We first apply the maximal coupling on $x_0$ and $x_0'$ such that

$$\mathbb{P}(x_0 \neq x_0') = \|\mathcal{L}(x_0) - \mathcal{L}(x_0')\|_{\mathrm{TV}} = \epsilon.$$

For the case $x_0 = x_0'$, we can couple the two Markov chains $\{x_k\}_{k \geq 0}$ and $\{x_k'\}_{k \geq 0}$ such that

$$x_k \equiv x_k', \forall k \geq 0.$$

Under this coupling, we have $\theta_k \equiv \theta_k', \forall k \geq 0$.

For the case $x_0 \neq x_0'$, we let the two processes $(x_k, \theta_k)_{k \geq 1}$ and $(x_k', \theta_k')_{k \geq 1}$ evolve independently.

Given the above coupling, we first observe that

$$
\begin{aligned}
\bar{W}_2(\mathcal{L}(x_k, \theta_k), \mathcal{L}(x_k', \theta_k')) &= \mathbb{E}[\bar{W}_2(\mathcal{L}(x_k, \theta_k), \mathcal{L}(x_k', \theta_k'))|x_0 = x_0']\mathbb{P}(x_0 = x_0') \\
&\quad + \mathbb{E}[\bar{W}_2(\mathcal{L}(x_k, \theta_k), \mathcal{L}(x_k', \theta_k'))|x_0 \neq x_0']\mathbb{P}(x_0 \neq x_0') \\
&= \epsilon\mathbb{E}[\bar{W}_2(\mathcal{L}(x_k, \theta_k), \mathcal{L}(x_k', \theta_k'))|x_0 \neq x_0'].
\end{aligned}
$$

The second equality holds since $\mathbb{E}[\bar{W}_2(\mathcal{L}(x_k, \theta_k), \mathcal{L}(x_k, \theta_k'))|x_0 = x_0'] = 0$ and $\mathbb{P}(x_0 \neq x_0') = \epsilon$.

Next, we note the following upper bound of the Wasserstein distance,

$$\bar{W}_2^2(\mathcal{L}(x_k, \theta_k), \mathcal{L}(x_k', \theta_k')) = \inf \mathbb{E}\Big[ \mathbb{1}\{x_k \neq x_k'\} + \|\theta_k - \theta_k'\|^2 \Big]$$

$$\leq 1 + 2\Big(\mathbb{E}[\|\theta_k - \theta^*\|^2] + \mathbb{E}[\|\theta_k' - \theta^*\|^2]\Big).$$

Making use of Proposition 4.2, we have

$$\bar{W}_2^2(\mathcal{L}(x_k, \theta_k), \mathcal{L}(x_k', \theta_k'))$$

$$\leq 2c_{2,1}(1 - \alpha\mu)^k \Big(\mathbb{E}[\|\theta_0 - \theta^*\|^2] + \mathbb{E}[\|\theta_0' - \theta^*\|^2]\Big) + 4c_{2,2}\alpha\tau \cdot \frac{L^2}{\mu} + 1$$

$$\leq 4c_{2,1}(1 - \alpha\mu)^k \mathbb{E}[\|\theta_0 - \theta^*\|^2] + 4c_{2,2}\alpha\tau \cdot \frac{L^2}{\mu} + 1,$$

where the second inequality holds for $\theta_0 = \theta_0'$ by assumption.

Note that the above upper bound to the Wasserstein distance is independent of the choice of $(x_0, x_0')$. Hence, we can conclude that

$$\bar{W}_2(\mathcal{L}(x_k, \theta_k), \mathcal{L}(x_k', \theta_k')) = \epsilon\mathbb{E}[\bar{W}_2(\mathcal{L}(x_k, \theta_k), \mathcal{L}(x_k', \theta_k'))|x_0 \neq x_0']$$

$$\leq \epsilon\Big(4c_{2,1}(1 - \alpha\mu)^k \mathbb{E}[\|\theta_0 - \theta^*\|^2] + 4c_{2,2}\alpha\tau \cdot \frac{L^2}{\mu} + 1\Big)^{1/2}.$$

We complete the proof of the lemma. $\qquad\square$

By Lemma E.5, we see that when $x_0$ is close to its stationary distribution $\pi$, $\theta_k$ would not deviate too much from $\theta_k'$, as if it were initialized from the stationary distribution.

Now we consider a joint process $(x_k, \theta_k)_{k\geq 0}$ with arbitrary initialization. By the property of uniform ergodicity of $(x_k)_{k\geq 0}$, we know that $\|\mathcal{L}(x_k) - \pi\|_{\mathrm{TV}} \leq Rr^k$. Choose time $t_0 \geq 0$. We construct a second Markov chain $(x_k', \theta_k')_{k\geq t_0}$ with the following properties: (1) $x_{t_0}' \sim \pi$ and is maximally coupled to $x_{k_0}$, i.e., $\|\mathcal{L}(x_{t_0}) - \mathcal{L}(x_{t_0}')\|_{\mathrm{TV}} = \mathbb{P}(x_{t_0} \neq x_{t_0}')$ and (2) $\theta_{t_0}' = \theta_{k_0}$. Under this construction, for $k \geq t_0 + \tau$, we have

$$\bar{W}_2(\mathcal{L}(x_k, \theta_k), \bar{\nu}) \leq \bar{W}_2(\mathcal{L}(x_k, \theta_k), \mathcal{L}(x_k', \theta_k')) + \bar{W}_2(\mathcal{L}(x_k', \theta_k'), \bar{\nu})$$

$$\leq Rr^{t_0}\Big(4c_{2,1}(1 - \alpha\mu)^{k-t_0}\mathbb{E}[\|\theta_{t_0} - \theta^*\|^2] + 4c_{2,2}\alpha\tau \cdot \frac{L^2}{\mu} + 1\Big)^{1/2}$$

$$+ 16(1 - \alpha\mu)^{k-t_0}\Big(\mathbb{E}\big[\|\theta_{t_0} - \theta^*\|^2\big] + c_{2,2}'\Big),$$

where the last inequality follows from Lemma E.4 and Lemma E.5.

For each $t$ with $t_0 \geq \tau$, set $t_0 = t/2$. From the above inequality, we obtain that

$$\bar{W}_2(\mathcal{L}(x_t, \theta_t), \bar{\nu})$$

$$\leq Rr^{t/2}\Big(4c_{2,1}(1 - \alpha\mu)^{t/2}\mathbb{E}[\|\theta_{t_0} - \theta^*\|^2] + 4c_{2,2}\alpha\tau \cdot \frac{L^2}{\mu} + 1\Big)^{1/2}$$

$$+ 16(1 - \alpha\mu)^{t/2}\Big(\mathbb{E}\big[\|\theta_{t_0} - \theta^*\|^2\big] + c_{2,2}'\Big)$$

$$\leq Rr^{t/2}\Big(4c_{2,1}(1 - \alpha\mu)^{t/2} \cdot \Big(8(1 - \alpha\mu)^{t_0}\mathbb{E}[\|\theta_0 - \theta^*\|^2] + c_{2,2}\alpha\tau \cdot \frac{L^2}{\mu}\Big) + 4c_{2,2}\alpha\tau \cdot \frac{L^2}{\mu} + 1\Big)^{1/2}$$

$$+ 16(1 - \alpha\mu)^{t/2}\Big(\Big(8(1 - \alpha\mu)^{t_0}\mathbb{E}[\|\theta_0 - \theta^*\|^2] + c_{2,2}\alpha\tau \cdot \frac{L^2}{\mu}\Big) + c_{2,2}'\Big)$$

$$\leq \max(r, 1 - \alpha\mu)^{t/2} \cdot s(\theta_0, \theta^*, \mu, L, R)$$

$$\leq (1 - \alpha\mu)^{t/2} \cdot s(\theta_0, \theta^*, \mu, L, R)$$

where $s(\theta_0, L, \mu)$ denote some constant that depends on the initialization of $\theta_0$, and problem primitives $L, \mu$, but independent of stepsize $\alpha$ and iteration index $t$. Last inequality holds because $\mu \leq 1 - r$ and $\alpha \leq \frac{\mu}{908L^2} \leq 1$.

Therefore, as $t \to \infty$, we obtain that $\bar{W}_2(\mathcal{L}(x_k, \theta_k), \bar{\nu}) \to 0$, which implies that the Markov chain $(x_k, \theta_k)_{k\geq 0}$ with arbitrary initialization converges to the same $\bar{\nu}$. As such, we have proved the desired weak convergence result without the assumption on $x_0 \sim \pi$ initialization.

Additionally, we obtain the following convergence rate. For any initialization $(x_0, \theta_0) \in \mathcal{X} \times \mathbb{R}^d$, we have

$$W_2(\mathcal{L}(\theta_t), \mu) \leq \bar{W}_2(\mathcal{L}(x_t, \theta_t), \bar{\nu}) \leq (1 - \alpha\mu)^{t/2} \cdot s(\theta_0, L, \mu).$$

### E.2.1 Proof of Proposition E.2

First, we present the following lemma that is similar to [17, Lemma 2.3].

**Lemma E.6.** *For any $k_1 < k_2$ satisfying $\alpha(k_2 - k_1) \leq \frac{1}{8L}$, the following six inequalities hold:*

$$\|\theta_{k_2}^{[1]} - \theta_{k_2}^{[2]} - \theta_{k_1}^{[1]} + \theta_{k_1}^{[2]}\| \leq 8\alpha L(k_2 - k_1)\|\theta_{k_2}^{[1]} - \theta_{k_2}^{[2]}\|$$

$$\|\theta_{k_2}^{[1]} - \theta_{k_2}^{[2]} - \theta_{k_1}^{[1]} + \theta_{k_1}^{[2]}\| \leq 8\alpha L(k_2 - k_1)\|\theta_{k_1}^{[1]} - \theta_{k_1}^{[2]}\|$$

$$\|\theta_{k_2}^{[1]} - \theta_{k_1}^{[1]}\| \leq 8\alpha L(k_2 - k_1)\left(\|\theta_{k_2}^{[1]}\| + 1\right)$$

$$\|\theta_{k_2}^{[1]} - \theta_{k_1}^{[1]}\| \leq 8\alpha L(k_2 - k_1)\left(\|\theta_{k_1}^{[1]}\| + 1\right)$$

$$\|\theta_{k_2}^{[2]} - \theta_{k_1}^{[2]}\| \leq 8\alpha L(k_2 - k_1)\left(\|\theta_{k_2}^{[2]}\| + 1\right)$$

$$\|\theta_{k_2}^{[2]} - \theta_{k_1}^{[2]}\| \leq 8\alpha L(k_2 - k_1)\left(\|\theta_{k_1}^{[2]}\| + 1\right).$$

*Proof.* Consider the coupling given by equation (E.3), by Assumption 2 and 6, we have

$$\|\theta_{k+1}^{[1]} - \theta_{k+1}^{[2]}\| - \|\theta_k^{[1]} - \theta_k^{[2]}\| \leq \|\theta_{k+1}^{[1]} - \theta_{k+1}^{[2]} - \theta_k^{[1]} + \theta_k^{[2]}\|$$
$$= \alpha\|g(\theta_k^{[1]}, x_k) + \xi_{k+1}(\theta_k^{[1]}) - g(\theta_k^{[1]}, x_k) - \xi_{k+1}(\theta_k^{[1]})\|$$
$$\leq 2\alpha L\|\theta_k^{[1]} - \theta_k^{[2]}\|.$$

Given $k_1 < k_2$, for $\forall t \in [k_1, k_2]$, since $1 + x \leq e^x$ for $\forall x \in R$, we have

$$\|\theta_t^{[1]} - \theta_t^{[2]}\| \leq \prod_{j=k_1}^{t-1} (1 + 2\alpha L)\|\theta_{k_1}^{[1]} - \theta_{k_1}^{[2]}\|$$
$$\leq \exp(2\alpha(k_2 - k_1)L)\|\theta_{k_1}^{[1]} - \theta_{k_1}^{[2]}\|$$
$$\overset{(i)}{\leq} (1 + 4\alpha(k_2 - k_1)L)\|\theta_{k_1}^{[1]} - \theta_{k_1}^{[2]}\|$$
$$\leq 2\|\theta_{k_1}^{[1]} - \theta_{k_1}^{[2]}\|,$$

where (i) holds for $e^x \leq 1 + 2x$ $\forall x \in [0, \frac{1}{2}]$ and $\alpha(k_2 - k_1) \leq \frac{1}{8L}$.

Then, we have

$$\|\theta_{k_2}^{[1]} - \theta_{k_2}^{[2]} - \theta_{k_1}^{[1]} + \theta_{k_1}^{[2]}\| \leq \sum_{t=k_1}^{k_2-1} \|\theta_{t+1}^{[1]} - \theta_{t+1}^{[2]} - \theta_t^{[1]} + \theta_t^{[2]}\| \leq 4\alpha(k_2 - k_1)L\|\theta_{k_1}^{[1]} - \theta_{k_1}^{[2]}\|.$$

Therefore, following $\alpha(k_2 - k_1) \leq \frac{1}{8L}$, we have

$$\|\theta_{k_2}^{[1]} - \theta_{k_2}^{[2]} - \theta_{k_1}^{[1]} + \theta_{k_1}^{[2]}\| \leq 4\alpha(k_2 - k_1)L\|\theta_{k_1}^{[1]} - \theta_{k_1}^{[2]}\|$$
$$\leq 4\alpha(k_2 - k_1)L(\|\theta_{k_2}^{[1]} - \theta_{k_2}^{[2]} - \theta_{k_1}^{[1]} + \theta_{k_1}^{[2]}\| + \|\theta_{k_2}^{[1]} - \theta_{k_2}^{[2]}\|)$$
$$\leq \frac{1}{2}\|\theta_{k_2}^{[1]} - \theta_{k_2}^{[2]} - \theta_{k_1}^{[1]} + \theta_{k_1}^{[2]}\| + 4\alpha(k_2 - k_1)L\|\theta_{k_2}^{[1]} - \theta_{k_2}^{[2]}\|.$$

Then, by rearranging the terms, we have

$$\|\theta_{k_2}^{[1]} - \theta_{k_2}^{[2]} - \theta_{k_1}^{[1]} + \theta_{k_1}^{[2]}\| \leq 8\alpha(k_2 - k_1)L\|\theta_{k_2}^{[1]} - \theta_{k_2}^{[2]}\|,$$

thereby we have proved the first two inequalities of Lemma E.6.

By Assumption 2 and 6 and [17, Lemma 2.3], we can prove the last four inequalities and we omit the details here. □

Now, we are ready to prove Proposition E.2. We start with the following decomposition. By equation (E.3), we first have

$$\mathbb{E}[\|\theta_{k+1}^{[1]} - \theta_{k+1}^{[2]}\|^2] = \mathbb{E}\left[\|\theta_k^{[1]} - \theta_k^{[2]} + \alpha\left(g(\theta_k^{[1]}, x_k) - g(\theta_k^{[2]}, x_k) + \xi_{k+1}(\theta_k^{[1]}) - \xi_{k+1}(\theta_k^{[2]})\right)\|^2\right]$$

$$=\mathbb{E}\left[\|\theta_k^{[1]}-\theta_k^{[2]}\|^2\right]+2\alpha\underbrace{\mathbb{E}\left[\langle\theta_k^{[1]}-\theta_k^{[2]},g(\theta_k^{[1]},x_k)-g(\theta_k^{[2]},x_k)\rangle\right]}_{T_1}$$

$$+\alpha^2\underbrace{\mathbb{E}\left[\|g(\theta_k^{[1]},x_k)-g(\theta_k^{[2]},x_k)+\xi_{k+1}(\theta_k^{[1]})-\xi_{k+1}(\theta_k^{[2]})\|^2\right]}_{T_2}.$$

For $T_2$, by Assumption 2 and 6, we have

$$T_2\le 4L^2\mathbb{E}\left[\|\theta_k^{[1]}-\theta_k^{[2]}\|^2\right].$$

We denote $\varepsilon(\cdot,x_k):=g(\cdot,x_k)-\bar{g}(\cdot)$ to be the noise function. Then, by Assumption 2, we conclude that $\varepsilon(\cdot,x_k)$ is $2L$-Lipschitz continuous. Therefore, $T_1$ can be rewritten as:

$$T_1=\mathbb{E}\left[\langle\theta_k^{[1]}-\theta_k^{[2]},\varepsilon(\theta_k^{[1]},x_k)-\varepsilon(\theta_k^{[2]},x_k)\rangle\right]+\mathbb{E}\left[\langle\theta_k^{[1]}-\theta_k^{[2]},\bar{g}(\theta_k^{[1]})-\bar{g}(\theta_k^{[2]})\rangle\right].$$

For $\mathbb{E}\left[\langle\theta_k^{[1]}-\theta_k^{[2]},\bar{g}(\theta_k^{[1]})-\bar{g}(\theta_k^{[2]})\rangle\right]$, by Assumption 3, we have

$$\mathbb{E}\left[\langle\theta_k^{[1]}-\theta_k^{[2]},\bar{g}(\theta_k^{[1]})-\bar{g}(\theta_k^{[2]})\rangle\right]\le-\mu\mathbb{E}\left[\|\theta_k^{[1]}-\theta_k^{[2]}\|^2\right].$$

Let $\mathcal{F}_k:=\sigma\left((\theta_t^{[1]},\theta_t^{[2]},x_t)\mid t\le k\right)$. For $\mathbb{E}\left[\langle\theta_k^{[1]}-\theta_k^{[2]},\varepsilon(\theta_k^{[1]},x_k)-\varepsilon(\theta_k^{[2]},x_k)\rangle\right]$, we have

$$\mathbb{E}\left[\langle\theta_k^{[1]}-\theta_k^{[2]},\varepsilon(\theta_k^{[1]},x_k)-\varepsilon(\theta_k^{[2]},x_k)\rangle\right]$$

$$=\mathbb{E}\left[\mathbb{E}\left[\langle\theta_k^{[1]}-\theta_k^{[2]},\varepsilon(\theta_k^{[1]},x_k)-\varepsilon(\theta_k^{[2]},x_k)\rangle\mid\mathcal{F}_{k-\tau}\right]\right]$$

$$=\mathbb{E}\left[\mathbb{E}\left[\langle\theta_{k-\tau}^{[1]}-\theta_{k-\tau}^{[2]},\varepsilon(\theta_{k-\tau}^{[1]},x_k)-\varepsilon(\theta_{k-\tau}^{[2]},x_k)\rangle\mid\mathcal{F}_{k-\tau}\right]\right]\tag{$T_3$}$$

$$+\mathbb{E}\left[\langle\theta_k^{[1]}-\theta_k^{[2]}-\theta_{k-\tau}^{[1]}-\theta_{k-\tau}^{[2]},\varepsilon(\theta_{k-\tau}^{[1]},x_k)-\varepsilon(\theta_{k-\tau}^{[2]},x_k)\rangle\right]\tag{$T_4$}$$

$$+\mathbb{E}\left[\underbrace{\langle\theta_k^{[1]}-\theta_k^{[2]},\varepsilon(\theta_k^{[1]},x_k)-\varepsilon(\theta_k^{[2]},x_k)-\varepsilon(\theta_{k-\tau}^{[1]},x_k)+\varepsilon(\theta_{k-\tau}^{[2]},x_k)\rangle}_{\spadesuit}\right].$$

We assume $\alpha\tau\le\frac{1}{8L}$. For $T_3$, by definition of mixing time $\tau$, we obtain

$$T_3=\mathbb{E}\left[\mathbb{E}\left[\langle\theta_{k-\tau}^{[1]}-\theta_{k-\tau}^{[2]},\varepsilon(\theta_{k-\tau}^{[1]},x_k)-\varepsilon(\theta_{k-\tau}^{[2]},x_k)\rangle\mid\theta_{k-\tau}^{[1]},\theta_{k-\tau}^{[2]},x_{k-\tau}\right]\right]$$

$$\le 2\alpha L\mathbb{E}\left[\|\theta_{k-\tau}^{[1]}-\theta_{k-\tau}^{[2]}\|^2\right]$$

$$\le 4\alpha L\mathbb{E}\left[\|\theta_k^{[1]}-\theta_k^{[2]}\|^2\right]+4\alpha L\mathbb{E}\left[\|\theta_{k-\tau}^{[1]}-\theta_{k-\tau}^{[2]}-\theta_k^{[1]}+\theta_k^{[2]}\|^2\right]$$

$$\le(4\alpha L+256\alpha^3\tau^2 L^3)\mathbb{E}\left[\|\theta_k^{[1]}-\theta_k^{[2]}\|^2\right],$$

where the last inequality holds by Lemma E.6.

For $T_4$, we obtain

$$T_4\le\mathbb{E}\left[\|\theta_k^{[1]}-\theta_k^{[2]}-\theta_{k-\tau}^{[1]}+\theta_{k-\tau}^{[2]}\|\|\varepsilon(\theta_{k-\tau}^{[1]},x_k)-\varepsilon(\theta_{k-\tau}^{[2]},x_k)\|\right]$$

$$\le 16\alpha\tau L^2\mathbb{E}\left[\|\theta_k^{[1]}-\theta_k^{[2]}\|\|\theta_{k-\tau}^{[1]}-\theta_{k-\tau}^{[2]}\|\right]$$

$$\le(16\alpha\tau L^2+128\alpha^2\tau^2 L^3)\mathbb{E}\left[\|\theta_k^{[1]}-\theta_k^{[2]}\|^2\right].$$

Below, we bound term $\spadesuit$ by two different Taylor expansions. One the one hand, there exist $\lambda_1,\lambda_2\in[0,1]$ such that $h_k=\lambda_1\theta_k^{[1]}+(1-\lambda_1)\theta_k^{[2]}$, $h_{k-\tau}=\lambda_2\theta_{k-\tau}^{[1]}+(1-\lambda_2)\theta_{k-\tau}^{[2]}$ and

$$|\spadesuit|=|\langle\theta_k^{[1]}-\theta_k^{[2]},\varepsilon'(h_k,x_k)(\theta_k^{[1]}-\theta_k^{[2]})-\varepsilon'(h_{k-\tau},x_k)(\theta_{k-\tau}^{[1]}-\theta_{k-\tau}^{[2]})\rangle|$$

$$=|\langle\theta_k^{[1]}-\theta_k^{[2]},\varepsilon'(h_{k-\tau},x_k)(\theta_k^{[1]}-\theta_k^{[2]}-\theta_{k-\tau}^{[1]}+\theta_{k-\tau}^{[2]})\rangle$$

$$+ \langle \theta_k^{[1]} - \theta_k^{[2]}, (\varepsilon'(h_k, x_k) - \varepsilon'(h_{k-\tau}, x_k))(\theta_k^{[1]} - \theta_k^{[2]}) \rangle |$$

$$\leq 16\alpha\tau L^2 \|\theta_k^{[1]} - \theta_k^{[2]}\|^2 + |\langle \theta_k^{[1]} - \theta_k^{[2]}, (\varepsilon'(h_k, x_k) - \varepsilon'(h_{k-\tau}, x_k))(\theta_k^{[1]} - \theta_k^{[2]}) \rangle | \qquad \text{(E.6)}$$

$$\leq 16\alpha\tau L^2 \|\theta_k^{[1]} - \theta_k^{[2]}\|^2 + 2L\|\theta_k^{[1]} - \theta_k^{[2]}\|^2 \|h_k - h_{k-\tau}\|$$

$$\leq 16\alpha\tau L^2 \|\theta_k^{[1]} - \theta_k^{[2]}\|^2 + 2L\|\theta_k^{[1]} - \theta_k^{[2]}\|^2 (\|h_k - \theta_k^{[2]}\| + \|\theta_k^{[2]} - \theta_{k-\tau}^{[2]}\| + \|\theta_{k-\tau}^{[2]} - h_{k-\tau}\|)$$

$$\qquad \qquad \text{(E.7)}$$

$$\leq 16\alpha\tau L^2 \|\theta_k^{[1]} - \theta_k^{[2]}\|^2 + 2L\|\theta_k^{[1]} - \theta_k^{[2]}\|^2 (\|\theta_k^{[1]} - \theta_k^{[2]}\| + \|\theta_k^{[2]} - \theta_{k-\tau}^{[2]}\| + \|\theta_{k-\tau}^{[1]} - \theta_{k-\tau}^{[2]}\|)$$

$$\leq 16\alpha\tau L^2 \|\theta_k^{[1]} - \theta_k^{[2]}\|^2 + 2L\|\theta_k^{[1]} - \theta_k^{[2]}\|^2 (2\|\theta_k^{[1]} - \theta_k^{[2]}\| + 8\alpha\tau L(\|\theta_k^{[2]}\| + 1) + 8\alpha\tau L\|\theta_k^{[1]} - \theta_k^{[2]}\|)$$

$$\qquad \qquad \text{(E.8)}$$

$$\leq 16\alpha\tau L^2 \|\theta_k^{[1]} - \theta_k^{[2]}\|^2 + 2L\|\theta_k^{[1]} - \theta_k^{[2]}\|^2 \Big( 3\|\theta_k^{[1]} - \theta_k^{[2]}\| + \underbrace{8\alpha\tau L(\|\theta_k^{[2]}\| + 1)}_{A} \Big) := \text{(Bound1)}.$$

where we note that choosing the second iterates for triangle inequality in equation (E.7) and choosing $\theta_k^{[2]}$ for bounding $\theta_k^{[2]} - \theta_{k-\tau}^{[2]}$ in equation (E.8) are both symmetric, which implies that we can replace the $\theta_k^{[2]}$ in term $A$ with arbitrary one in $\{\theta_{k-\tau}^{[1]}, \theta_k^{[1]}, \theta_{k-\tau}^{[2]}, \theta_k^{[2]}\}$.

On the other hand, there exist $\bar{\lambda}_1, \bar{\lambda}_2 \in [0, 1]$ such that $p_k = \bar{\lambda}_1 \theta_k^{[1]} + (1 - \bar{\lambda}_1)\theta_{k-\tau}^{[1]}$, $q_k = \bar{\lambda}_2 \theta_k^{[2]} + (1 - \bar{\lambda}_2)\theta_{k-\tau}^{[2]}$ and

$$|\spadesuit| = |\langle \theta_k^{[1]} - \theta_k^{[2]}, \varepsilon'(p_k, x_k)(\theta_k^{[1]} - \theta_{k-\tau}^{[1]}) - \varepsilon'(q_k, x_k)(\theta_k^{[2]} - \theta_{k-\tau}^{[2]}) \rangle |$$

$$= |\langle \theta_k^{[1]} - \theta_k^{[2]}, \varepsilon'(p_k, x_k)(\theta_k^{[1]} - \theta_k^{[2]} - \theta_{k-\tau}^{[1]} + \theta_{k-\tau}^{[2]}) \rangle \qquad \text{(E.9)}$$

$$+ \langle \theta_k^{[1]} - \theta_k^{[2]}, (\varepsilon'(p_k, x_k) - \varepsilon'(q_k, x_k))(\theta_k^{[2]} - \theta_{k-\tau}^{[2]}) \rangle |$$

$$\leq 16\alpha\tau L^2 \|\theta_k^{[1]} - \theta_k^{[2]}\|^2 + 2L\|\theta_k^{[1]} - \theta_k^{[2]}\| \|p_k - q_k\| \|\theta_k^{[2]} - \theta_{k-\tau}^{[2]}\|$$

$$\leq 16\alpha\tau L^2 \|\theta_k^{[1]} - \theta_k^{[2]}\|^2 + 16\alpha\tau L^2 \|\theta_k^{[1]} - \theta_k^{[2]}\| \|p_k - q_k\| (\|\theta_k^{[2]}\| + 1), \qquad \text{(E.10)}$$

where adding and subtracting the second iterates in equation (E.7) and choosing $\theta_k^{[2]}$ for bounding $\theta_k^{[2]} - \theta_{k-\tau}^{[2]}$ in equation (E.8) are both symmetric. We have

$$\|p_k - q_k\| = \|\bar{\lambda}_1(\theta_k^{[1]} - \theta_k^{[2]}) + (1 - \bar{\lambda}_1)(\theta_{k-\tau}^{[1]} - \theta_{k-\tau}^{[2]}) + (\bar{\lambda}_1 - \bar{\lambda}_2)(\theta_k^{[2]} - \theta_{k-\tau}^{[2]})\|$$

$$\leq (2 + 8\alpha\tau L)\|\theta_k^{[1]} - \theta_k^{[2]}\| + 8\alpha\tau L(\|\theta_k^{[2]}\| + 1)$$

$$\leq 3\|\theta_k^{[1]} - \theta_k^{[2]}\| + 8\alpha\tau L(\|\theta_k^{[2]}\| + 1).$$

Therefore, we obtain

$$|\spadesuit| \leq 16\alpha\tau L^2 \|\theta_k^{[1]} - \theta_k^{[2]}\|^2$$

$$+ 2L\|\theta_k^{[1]} - \theta_k^{[2]}\| \Big( 3\|\theta_k^{[1]} - \theta_k^{[2]}\| + \underbrace{8\alpha\tau L(\|\theta_k^{[2]}\| + 1)}_{B} \Big) \underbrace{8\alpha\tau L(\|\theta_k^{[2]}\| + 1)}_{C} := \text{(Bound2)},$$

and we can replace the $\theta_k^{[2]}$ in term $B$ and $C$ with arbitrary one in $\{\theta_{k-\tau}^{[1]}, \theta_k^{[1]}, \theta_{k-\tau}^{[2]}, \theta_k^{[2]}\}$.

By (Bound1) and (Bound2), we have

$$|\spadesuit| \leq \min(\text{Bound1}, \text{Bound2})$$

$$\leq 16\alpha\tau L^2 \|\theta_k^{[1]} - \theta_k^{[2]}\|^2 + 48\alpha\tau L^2 \|\theta_k^{[1]} - \theta_k^{[2]}\|^2 (\|\theta_k^{[2]}\| + 1).$$

Below, we discuss the upper bound for $|\spadesuit|$ by three cases.

**Case 1:** If one of $\{\theta_{k-\tau}^{[1]}, \theta_k^{[1]}, \theta_{k-\tau}^{[2]}, \theta_k^{[2]}\}$ has norm that is less than $4\delta_{\alpha\tau L^2}$, where we define $\delta_{\alpha\tau L^2} := \delta(\alpha\tau L^2)$, without loss of generality, we assume $\|\theta_k^{[2]}\| \leq 4\delta_{\alpha\tau L^2}$. Multiply $\mathbb{1}(\|\theta_k^{[2]}\| \leq 4\delta_{\alpha\tau L^2})$ to both sides of the above inequality, and we have

$$|\spadesuit| \mathbb{1}(\|\theta_k^{[2]}\| \leq 4\delta_{\alpha\tau L^2})$$

$$\leq \left(16\alpha\tau L^2\|\theta_k^{[1]} - \theta_k^{[2]}\|^2 + 48\alpha\tau L^2\|\theta_k^{[1]} - \theta_k^{[2]}\|^2(\|\theta_k^{[2]}\| + 1)\right)\mathbb{1}(\|\theta_k^{[2]}\| \leq 4\delta_{\alpha\tau L^2})$$

$$\leq (64 + 192\delta_{\alpha\tau L^2})\alpha\tau L^2\|\theta_k^{[1]} - \theta_k^{[2]}\|^2,$$

where we can replace the $\theta_k^{[2]}$ in the left hand of the above inequality with an arbitrary one in $\{\theta_{k-\tau}^{[1]}, \theta_k^{[1]}, \theta_{k-\tau}^{[2]}, \theta_k^{[2]}\}$ by the similar argument under (Bound1) and (Bound2).

**Case 2:** If $\|\theta_k^{[1]}\| < 2\|\theta_k^{[1]} - \theta_k^{[2]}\|$, we obtain

$$\|\theta_k^{[2]}\| \leq \|\theta_k^{[1]} - \theta_k^{[2]}\| + \|\theta_k^{[1]}\| \leq 3\|\theta_k^{[1]} - \theta_k^{[2]}\|.$$

Therefore, by definition of ♠ and Lemma E.6, we obtain

$$|♠|\mathbb{1}(\|\theta_k^{[1]}\| \leq 2\|\theta_k^{[1]} - \theta_k^{[2]}\|)$$
$$\leq 2L\|\theta_k^{[1]} - \theta_k^{[2]}\|(\|\theta_k^{[1]} - \theta_{k-\tau}^{[1]}\| + \|\theta_k^{[2]} - \theta_{k-\tau}^{[2]}\|)\mathbb{1}(\|\theta_k^{[1]}\| \leq 2\|\theta_k^{[1]} - \theta_k^{[2]}\|)$$
$$\leq 16\alpha\tau L^2\|\theta_k^{[1]} - \theta_k^{[2]}\|(\|\theta_k^{[1]}\| + \|\theta_k^{[2]}\|)\mathbb{1}(\|\theta_k^{[1]}\| \leq 2\|\theta_k^{[1]} - \theta_k^{[2]}\|)$$
$$\leq 80\alpha\tau L^2\|\theta_k^{[1]} - \theta_k^{[2]}\|.$$

**Case 3:** If all four variables in $\{\theta_{k-\tau}^{[1]}, \theta_k^{[1]}, \theta_{k-\tau}^{[2]}, \theta_k^{[2]}\}$ have a norm larger than $4\delta_{\alpha\tau L^2}$, and $\|\theta_k^{[1]}\| \geq 2\|\theta_k^{[1]} - \theta_k^{[2]}\|$, we obtain

$$\|h_k\| = \|\lambda_1\theta_k^{[1]} + (1-\lambda_1)\theta_k^{[2]}\| \geq \|\theta_k^{[1]}\| - \|\theta_k^{[1]} - \theta_k^{[2]}\| \geq \frac{\|\theta_k^{[1]}\|}{2} \geq 2\delta_{\alpha\tau L^2}$$

$$\|h_{k-\tau}\| = \|\lambda_2\theta_{k-\tau}^{[1]} + (1-\lambda_2)\theta_{k-\tau}^{[2]}\|$$
$$\geq \|\theta_{k-\tau}^{[1]}\| - \|\theta_{k-\tau}^{[1]} - \theta_{k-\tau}^{[2]}\|$$
$$\overset{(i)}{\geq} \delta_{\alpha\tau L^2} - (1 + 8\alpha\tau L)\|\theta_k^{[1]} - \theta_k^{[2]}\|$$
$$\geq \delta_{\alpha\tau L^2},$$

where (i) holds by choosing $\alpha\tau \leq \frac{1}{16L}$. Therefore, we have $\|h_k\| \geq \delta_{\alpha\tau L^2}$ and $\|h_{k-\tau}\| \geq \delta_{\alpha\tau L^2}$. Therefore, by equation (E.6) and Assumption 6, we obtain

$$|♠|\mathbb{1}(\text{Case 3})$$
$$\leq 16\alpha\tau L^2\|\theta_k^{[1]} - \theta_k^{[2]}\|^2 + \|\theta_k^{[1]} - \theta_k^{[2]}\|^2 \left(\|\varepsilon'(h_k, x_k) - G(x_k)\| + \|\varepsilon'(h_{k-\tau}, x_k) - G(x_k)\|\right)$$
$$\leq 18\alpha\tau L^2\|\theta_k^{[1]} - \theta_k^{[2]}\|^2.$$

Therefore, we obtain

$$|\mathbb{E}[♠]| \leq \mathbb{E}[|♠|] \leq \mathbb{E}[|♠|\mathbb{1}(\|\theta_{k-\tau}^{[1]}\| \leq \delta_{\alpha\tau L^2})] + \mathbb{E}[|♠|\mathbb{1}(\|\theta_k^{[1]}\| \leq \delta_{\alpha\tau L^2})]$$
$$+ \mathbb{E}[|♠|\mathbb{1}(\|\theta_{k-\tau}^{[2]}\| \leq \delta_{\alpha\tau L^2})] + \mathbb{E}[|♠|\mathbb{1}(\|\theta_k^{[2]}\| \leq \delta_{\alpha\tau L^2})]$$
$$+ \mathbb{E}[|♠|\mathbb{1}(\|\theta_k^{[1]}\| \leq 2\|\theta_k^{[1]} - \theta_k^{[2]}\|)] + \mathbb{E}[|♠|\mathbb{1}(\text{Case 3})]$$
$$\leq (354 + 768\delta_{\alpha\tau L^2})\alpha\tau L^2\mathbb{E}[\|\theta_k^{[1]} - \theta_k^{[2]}\|^2].$$

By Assumption 6, there exists $\kappa_\mu > 0$ such that $\epsilon\delta(\epsilon) \leq \frac{\mu}{3072}$ for $\forall\epsilon \leq \kappa_\mu$. By the above bounds, when $\alpha\tau \leq \min(\frac{c\mu}{L^2}, \frac{\kappa_\mu}{L^2})$, where we specify the constant $c < 1$ later, we obtain

$$\mathbb{E}[\|\theta_{k+1}^{[1]} - \theta_{k+1}^{[2]}\|^2]$$
$$\leq \left(1 + 2\alpha(-\mu + 4\alpha L + 256\alpha^3\tau^2 L^3 + 128\alpha^2\tau^2 L^3 + 784\alpha\tau L^2 + 768\delta_{\alpha\tau L^2}\alpha\tau L^2) + 4\alpha^2 L^2\right)\mathbb{E}[\|\theta_k^{[1]} - \theta_k^{[2]}\|^2]$$
$$= \left(1 + \alpha(-2\mu + 1568\alpha\tau L^2 + 256\alpha^2\tau^2 L^3 + 1536\delta_{\alpha\tau L^2}\alpha\tau L^2) + \alpha^2(8L + 512\alpha^2\tau^2 L^3 + 4L^2)\right)\mathbb{E}[\|\theta_k^{[1]} - \theta_k^{[2]}\|^2]$$
$$\leq \left(1 + \alpha\mu(-2 + 1568c + 256c + \frac{1}{2} + 524c)\right)\mathbb{E}[\|\theta_k^{[1]} - \theta_k^{[2]}\|^2]$$

$$\leq (1-\mu\alpha)\mathbb{E}[\|\theta_k^{[1]} - \theta_k^{[2]}\|^2],$$

where the last inequality holds by choosing $c = \frac{1}{4696}$.

Therefore, for $k \geq \tau$ and $\alpha\tau \leq \min(\frac{\mu}{906L^2}, \frac{\kappa\mu}{L^2})$ we have

$$
\begin{aligned}
\mathbb{E}[\|\theta_k^{[1]} - \theta_k^{[2]}\|^2] &\leq (1-\mu\alpha)^{k-\tau}\mathbb{E}[\|\theta_\tau^{[1]} - \theta_\tau^{[2]}\|^2] \\
&\leq 2(1-\mu\alpha)^{k-\tau}\mathbb{E}[\|\theta_0^{[1]} - \theta_0^{[2]}\|^2 + \|\theta_\tau^{[1]} - \theta_\tau^{[2]} - \theta_0^{[1]} + \theta_0^{[2]}\|^2] \\
&\leq 2(1+8\alpha\tau L)(1-\mu\alpha)^{k-\tau}\mathbb{E}[\|\theta_0^{[1]} - \theta_0^{[2]}\|^2] \\
&\leq 4(1-\mu\alpha)^{k-\tau}\mathbb{E}[\|\theta_0^{[1]} - \theta_0^{[2]}\|^2].
\end{aligned}
$$

### E.3 Proof of Projected SA

Now, we specialize the proof of Theorem E.1 to the projected SA iterates.

We consider the same coupling:

$$
\begin{aligned}
\theta_{k+\frac{1}{2}}^{[1]} &= \theta_k^{[1]} + \alpha\big(g(\theta_k^{[1]}, x_k) + \xi_{k+1}(\theta_k^{[1]})\big), \\
\theta_{k+1}^{[1]} &= \Pi_{B(\beta)}\Big[\theta_{k+\frac{1}{2}}^{[1]}\Big], \\
\theta_{k+\frac{1}{2}}^{[2]} &= \theta_k^{[2]} + \alpha\big(g(\theta_k^{[2]}, x_k) + \xi_{k+1}(\theta_k^{[2]})\big), \\
\theta_{k+1}^{[2]} &= \Pi_{B(\beta)}\Big[\theta_{k+\frac{1}{2}}^{[2]}\Big].
\end{aligned}
$$

We first need to verify that Proposition E.2 holds for projected SA. By the non-expansion property of $\Pi_{B(\beta)}$ with respect to $\|\cdot\|$, we obtain the following inequality:

$$\|\theta_{k+1}^{[1]} - \theta_{k+1}^{[2]}\| - \|\theta_k^{[1]} - \theta_k^{[2]}\| \leq \|\theta_{k+\frac{1}{2}}^{[1]} - \theta_{k+\frac{1}{2}}^{[2]}\| - \|\theta_k^{[1]} - \theta_k^{[2]}\|,$$

which implies Lemma E.6 still holds for projected SA. For Proposition E.2, we can notice that the iterates of projected SA will always satisfy Case 1 with a finite bound $\beta$ and we do not need to discuss Cases 2 and 3.

Therefore, when $\alpha\tau \leq \min(\frac{c\mu}{L^2}, \frac{1}{8L})$, where we specify the constant $c < 1$ later, we obtain

$$
\begin{aligned}
&\mathbb{E}[\|\theta_{k+1}^{[1]} - \theta_{k+1}^{[2]}\|^2] \\
&\leq \Big(1 + \alpha(-2\mu + (160 + 96\beta))\alpha\tau L^2 + 256\alpha^2\tau^2 L^3) + \alpha^2(8L + 512\alpha^2\tau^2 L^3 + 4L^2)\Big)\mathbb{E}[\|\theta_k^{[1]} - \theta_k^{[2]}\|^2] \\
&\leq (1 + \alpha\mu(-2 + (160 + 96\beta)c + 256c + 524c))\,\mathbb{E}[\|\theta_k^{[1]} - \theta_k^{[2]}\|^2] \\
&= (1-\mu\alpha)\mathbb{E}[\|\theta_k^{[1]} - \theta_k^{[2]}\|^2],
\end{aligned}
$$

where we set $c = \frac{1}{940 + 96\beta}$.

Therefore, $\forall k \geq \tau$ and $\alpha\tau \leq \frac{\mu}{(940 + 96\beta)L^2}$, we have

$$\mathbb{E}[\|\theta_k^{[1]} - \theta_k^{[2]}\|^2] \leq 4(1-\mu\alpha)^{k-\tau}\mathbb{E}[\|\theta_0^{[1]} - \theta_0^{[2]}\|^2].$$

Then, still by the non-expansion of $\Pi_{B(\beta)}$ with respect to $\|\cdot\|$, the rest of the proof simply follows the same proof for non-projected SA. As such, we have proven Theorem 4.1.

## F Proof of Corollary 4.4

In this section, we present the proof of Corollary 4.4.

Recall that by Theorem 4.3, we obtain for $k \geq 2\tau$,

$$W_2^2\Big(\mathcal{L}(\theta_k), \nu_\alpha\Big) \leq (1-\alpha\mu)^k \cdot s(\theta_0, \theta^*, \mu, L, R).$$

By [63, Theorem 4.1], there exists a coupling between $\theta_k$ and $\theta_\infty$ such that

$$W_2^2(\mathcal{L}(\theta_k), \nu_\alpha) = \mathbb{E}[\|\theta_k - \theta_\infty\|^2].$$

Applying Jensen's inequality twice, we obtain that

$$\|\mathbb{E}[\theta_k - \theta_\infty]\|^2 \leq (\mathbb{E}[\|\theta_k - \theta_\infty\|])^2 \leq \mathbb{E}[\|\theta_k - \theta_\infty\|^2] \leq (1 - \alpha\mu)^k \cdot s(\theta_0, \theta^*, \mu, L, R).$$

We thus have for all $k \geq 2\tau$,

$$\|\mathbb{E}[\theta_k] - \mathbb{E}[\theta_\infty]\| \leq \mathbb{E}[\|\theta_k - \theta_\infty\|] \leq (1 - \alpha\mu)^{k/2} \cdot s'(\theta_0, \theta^*, \mu, L, R).$$

For the second moment, we first note that

$$\begin{aligned}
&\|\mathbb{E}[\theta_k \theta_k^\top] - \mathbb{E}[\theta_\infty \theta_\infty^\top]\| \\
&= \|\mathbb{E}[(\theta_k - \theta_\infty)(\theta_k - \theta_\infty)^\top] + \mathbb{E}[\theta_\infty(\theta_k - \theta_\infty)^\top] + \mathbb{E}[(\theta_k - \theta_\infty)\theta_\infty^\top]\| \\
&\leq \|\mathbb{E}[(\theta_k - \theta_\infty)(\theta_k - \theta_\infty)^\top]\| + \|\mathbb{E}[\theta_\infty(\theta_k - \theta_\infty)^\top]\| + \|\mathbb{E}[(\theta_k - \theta_\infty)\theta_\infty^\top]\| \\
&\leq \|\mathbb{E}[\|\theta_k - \theta_\infty\|^2]\| + 2\mathbb{E}[\|\theta_\infty\|\|\theta_k - \theta_\infty\|]\| \\
&\leq \|\mathbb{E}[\|\theta_k - \theta_\infty\|^2]\| + 2\sqrt{\mathbb{E}[\|\theta_\infty\|^2\|\theta_k - \theta_\infty\|^2]\|},
\end{aligned} \tag{F.1}$$

where we apply Cauchy-Schwarz to obtain the last inequality.

Meanwhile, we have

$$\mathbb{E}\left[\|\theta_k - \theta_\infty\|^2\right] \leq (1 - \alpha\mu)^k \cdot s(\theta_0, \theta^*, \mu, L, R) \quad \text{and} \quad \mathbb{E}\left[\|\theta_\infty\|^2\right] = \mathcal{O}(1).$$

Substituting the above bounds into the right-hand side of inequality (F.1) yields

$$\left\|\mathbb{E}\left[\theta_k \theta_k^\top\right] - \mathbb{E}\left[\theta_\infty \theta_\infty^\top\right]\right\| \leq (1 - \alpha\mu)^{k/2} \cdot s''(\theta_0, \theta^*, \mu, L, R).$$

## G   Proof of Corollary 4.5

In this section, we prove the CLT result.

*Proof.* Consider the following centered test function $\bar{h} : \mathcal{X} \times \mathbb{R}^d \to \mathbb{R}^d$ defined as

$$\bar{h}(x, \theta) = \theta - \mathbb{E}[\theta_\infty].$$

To prove that the CLT for function $\bar{h}$, we need to verify the Maxwell-Woodroofe condition [49], i.e.,

$$\sum_{n=1}^\infty n^{-3/2} \left\| \sum_{t=0}^{n-1} Q^t h \right\|_{L^2(\bar{\nu})} < \infty,$$

where $Q$ denotes the transition kernel of the joint Markov chain. If we can show the following

$$\left\| \sum_{t=0}^{n-1} Q^t h \right\|_{L^2(\bar{\nu})} = \mathcal{O}(n^r) \tag{G.1}$$

with $r \in [0, 1/2)$, then the Maxwell-Woodroofe condition is verified, as

$$\sum_{n=1}^\infty n^{-3/2} \left\| \sum_{t=0}^{n-1} Q^t h \right\|_{L^2(\bar{\nu})} = \sum_{n=1}^\infty n^{-3/2} \mathcal{O}(n^r) < \infty.$$

We now proceed to prove the desired order in (G.1). For sufficiently large $n \geq 2\tau_\alpha$, we observe

$$\begin{aligned}
\left\| \sum_{t=0}^{n-1} Q^t h \right\|_{L^2(\bar{\nu})} = \mathbb{E}_{\bar{\nu}} \left\| \sum_{t=0}^{n-1} Q^t h \right\|_2 &\leq \sum_{t=0}^{n-1} \mathbb{E}_{\bar{\nu}} \|Q^t h\|_2 \\
&= \underbrace{\sum_{t=0}^{2\tau_\alpha - 1} \mathbb{E}_{\bar{\nu}} \|Q^t h\|_2}_{T_1} + \underbrace{\sum_{t=2\tau_\alpha}^{n-1} \mathbb{E}_{\bar{\nu}} \|Q^t h\|_2}_{T_2}.
\end{aligned}$$

We now show that both terms $T_1$ and $T_2$ are of order $\mathcal{O}(1)$ with respect to the parameter $n$.

For $T_1$, since $Q$ is a transition kernel, so its $\|Q\|_{L^2(\mu)}$ operator norm equals to 1. Hence, $T_1$ can be upper bounded as
$$T_1 \leq \tau_\alpha \mathbb{E}_{\bar{\mu}}[\|h(\theta, x)\|_2^2] = \tau_\alpha \operatorname{Tr}(\operatorname{Var}(\theta_\infty)) < C_1,$$
where the last inequality follows from $\operatorname{Tr}(\operatorname{Var}(\theta_\infty)) \leq \alpha\tau$ established in (H.4) and $\alpha\tau_\alpha^2 \to 0$ by Definition 2.1.

Before proceeding to analyze the summation in $T_2$, we first recall (E.2), that for $t \geq 2\tau_\alpha$,
$$\bar{W}_2(\mathcal{L}(x_t, \theta_t), \bar{\nu}) = \mathcal{O}((1 - \alpha\mu)^{t/2}),$$
which holds for any $(x, \theta) \in \mathcal{X} \times \mathbb{R}^d$. Hence, by the property of Wasserstein distance [63], there always exists a coupling that attains the optimality, i.e.,
$$\mathbb{E}_{\Gamma((x_t, \theta_t), \bar{\nu})}\left[\|\theta_t - \theta'\|_2^2 + \delta_0(x_t \neq x')\right] = \mathcal{O}((1 - \alpha\mu)^t).$$

Making use of this relationship, we can therefore bound $T_2$,
$$T_2 = \sum_{t=\tau_\alpha}^{n-1} \mathbb{E}_{\bar{\nu}}\|Q^t h\|_2 \leq \sum_{t=\tau_\alpha}^{\infty} \mathbb{E}_{\bar{\nu}}\|Q^t h\|_2 = \mathcal{O}\left(\frac{1}{1 - (1 - \alpha\mu)^{1/2}}\right) = \mathcal{O}(1),$$

where the last $\mathcal{O}(\cdot)$ is asymptotic in $n$.

Combining the analysis of $T_1$ and $T_2$, we have shown the desired order in (G.1). Therefore, the Maxwell-Woodroofe condition has been verified and we establish the CLT for averaged nonlinear iterates with constant stepsize and Markovian data. □

# H  Proofs under Minorization Condition

When assuming the perturbed continuous noise condition in Assumption 5, one takes the alternative route to prove weak convergence. This is achieved by establishing the satisfaction of both a minorization condition and a drift condition. In this section, we prove the weak convergence result in Theorem 4.3 by following this alternative approach. The subsequent corollaries of weak convergence, namely the non-asymptotic convergence rate in Corollary 4.4 and the Central Limit Theorem (CLT) in Corollary 4.5, also hold, and we will provide the proofs for these results as well.

## H.1  Proof of Theorem 4.3

In this section, we prove the weak convergence under Assumption 5(a). The proof consists of two major steps. Firstly, built upon the MSE convergence established in Proposition 4.2, we derive a multi-state drift condition. Subsequently, we show that under the minorization condition, the Markov chain is $(x_k, \theta_k)_{k \geq 0}$ is $\varphi$-irreducible. Then, follow [50, Theorem 19.1.3], we can conclude that the Markov chain $(x_k, \theta_k)_{k \geq 0}$ is geometrically ergodic.

For completeness, we include the Theorem 19.1.3 from [50] below.

**Theorem H.1.** *Suppose that $\Phi$ is a $\varphi$-irreducible chain on $\mathcal{X}$, and let $n(x)$ be a measurable function from $\mathcal{X} \to \mathbb{Z}_+$. The chain is geometrically ergodic if it is aperiodic and there exists some petite set $C$, a nonnegative function $V \geq 1$ and bounded on $C$, and positive constants $\lambda < 1$ and $b$ satisfying*
$$\int P^{n(x)}(x, dy)V(y) \leq \lambda^{n(x)}[V(x) + b\mathbb{1}_C(x)]. \tag{H.1}$$

We note that the function $n(x)$ can be interpreted as the number of steps we must wait, starting from any $x$, for the drift to become negative.

**Step 1: Deriving the Drift Condition**  Given the iteration step
$$\theta_{k+1} = \theta_k + \alpha(g(\theta_k, x_k) + \xi_{t+1}(\theta_k)),$$
we have already shown the following convergence rate on the MSE in Proposition 4.2, that
$$\mathbb{E}[\|\theta_k - \theta^*\|^2] \leq c_{2,1}(1 - \alpha\mu)^k\|\theta_0 - \theta^*\|^2 + c_{2,2}\alpha\tau_\alpha\frac{L^2}{\mu}.$$

Inspired by the MSE convergence bound, we define the Lyapunov function $V : \mathcal{X} \times \mathbb{R}^d \to [1, \infty]$

$$V(x, \theta) = \|\theta - \theta^*\|^2 + 1. \tag{H.2}$$

Therefore, the major goal in this step is to obtain the desired drift condition as shown in (H.1).

Therefore, from the above MSE convergence rate, we first obtain that

$$Q^k V(x_0, \theta_0) \leq c_{2,1}(1 - \alpha\mu)^k V(x_0, \theta_0) + c_{2,2} \frac{L^2}{\mu} \alpha\tau + 1. \tag{H.3}$$

Consider $k = \min\{t \geq 0 : c_{2,1}(1 - \alpha\mu)^t < 1\}$. Then, set $\eta = 8(1 - \alpha\mu)^k$, $\beta = \frac{1-\eta}{2}$, and $m = c_{2,2} \frac{L^2}{\mu} \alpha\tau + 1$, and we consider the following bounded sublevel set,

$$C_\Theta = \{\theta : (\|\theta - \theta^*\|^2 + 1) \leq m/\beta\}.$$

From (H.3), we derive that

$$Q^k V(x_0, \theta_0) - V(x_0, \theta_0) \leq -\beta V(x_0, \theta_0) + m\mathbb{1}_{\bar{C}}(x_0, \theta_0),$$

where $\bar{C} = \mathcal{X} \times C_\Theta$. Rewriting the above Lyapunov drift condition, with $b = m/(1 - \beta)$, we obtain

$$Q^k V(x_0, \theta_0) \leq (1 - \beta)\Big(V(x_0, \theta_0) + b\mathbb{1}_{\bar{C}}(x_0, \theta_0)\Big).$$

Setting $\lambda$ such that $\lambda^k = 1 - \beta$, we have

$$Q^k V(x_0, \theta_0) \leq \lambda^k \Big(V(x_0, \theta_0) + b\mathbb{1}_{\bar{C}}(x_0, \theta_0)\Big),$$

which gives the desired multi-step drift condition.

**Step 2: Proving the Minorization Condition**   Now that we have established the desired multi-step drift condition, it remains for us to show that $\bar{C}$ is accessible, small, and aperiodic.

Under this setup of $\xi_t(\theta)$ in Assumption 5(a), it is straightforward to verify the accessibility of $\bar{C}$. For any $(x, \theta) \in \mathcal{X} \times \mathbb{R}^d$, we have

$$Q((x, \theta), \bar{C}) = \mathbb{P}((x', \theta') \in \mathcal{X} \times C_\Theta | (x, \theta))$$

$$= \mathbb{P}(\theta' \in C_\Theta | (x, \theta)) \geq \int_{\theta' \in C_\Theta} \frac{1}{\alpha^d} p_\theta\Big(\frac{\theta' - \theta}{\alpha} - g(x, \theta)\Big) d\theta' > 0.$$

As such, we have shown that $\bar{C}$ is accessible.

Assuming $\bar{C}$ is small, we can directly conclude aperiodicity following the definition of period of an accessible small set, $d(C) = \text{g.c.d.}\Big\{n \in \mathbb{N}^* : \inf_{x \in C} P^n(x, C) > 0\Big\}$.

Therefore, what remains to show is that $\bar{C}$ is small. For $(x, \theta) \in \bar{C}$ and $\bar{A} \in \mathcal{B}(\mathcal{X}) \times \mathcal{B}(\mathbb{R}^d)$, we define the following projection sets,

$$A_x = \{\theta \in \mathbb{R}^d | (x, \theta) \in \bar{A}\} \quad \text{and} \quad A^\theta = \{x \in \mathcal{X} | (x, \theta) \in \bar{A}\}.$$

Therefore,

$$Q^m((x, \theta), \bar{A}) = \int_{\substack{\{(x^{(k)}, \theta^{(k)})\}_{k=1}^{m-1} \\ \in (\mathcal{X} \times \mathbb{R}^d)^{m-1}}} \mathbb{P}((x_m, \theta_m) \in \bar{A} | (x_{m-1}, \theta_{m-1}) = (x^{(m-1)}, \theta^{(m-1)}))$$

$$\cdots \mathbb{P}((x_1, \theta_1) = \mathrm{d}(x^{(1)}, \theta^{(1)}) | (x_0, \theta_0) = (x, \theta))$$

$$\geq \int_{\substack{\{(x^{(k)}, \theta^{(k)})\}_{k=1}^{m-1} \\ \in (\mathcal{X} \times C_\Theta)^{m-1}}} \mathbb{P}((x_m, \theta_m) \in \bar{A} | (x_{m-1}, \theta_{m-1}) = (x^{(m-1)}, \theta^{(m-1)}))$$

$$\cdots \mathbb{P}((x_1, \theta_1) = \mathrm{d}(x^{(1)}, \theta^{(1)}) | (x_0, \theta_0) = (x, \theta))$$

$$= \int_{\substack{\{(x^{(k)}, \theta^{(k)})\}_{k=1}^{m-1} \\ \in (\mathcal{X} \times C_\Theta)^{m-1}}} \Big(\int_{x' \in \mathcal{X}} \mathbb{P}(x_m = \mathrm{d}x' | x_{m-1} = x^{(m-1)}) \mathbb{P}(\theta_m \in A_{x'} | (x_{m-1}, \theta_{m-1}) = (x^{(m-1)}, \theta^{(m-1)}))\Big)$$

$$\mathbb{P}(x_{m-1} = dx^{(m-1)}|x_{m-2} = x^{(m-2)})\mathbb{P}(\theta_{m-1} = d\theta^{(m-1)}|(x_{m-2}, \theta_{m-2}) = (x^{(m-2)}, \theta^{(m-2)})))\Big)$$

$$\cdots$$

$$\mathbb{P}(x_1 = dx^{(1)}|x_0 = x)\mathbb{P}(\theta_1 = d\theta^{(1)}|(x_0, \theta_0) = (x, \theta))$$

$$= \int_{x' \in \mathcal{X}} \int_{\substack{\{x^{(k)}\}_{k=1}^{m-1} \\ \in \mathcal{X}^{m-1}}} \mathbb{P}(x_m = dx'|x_{m-1} = x^{(m-1)})\mathbb{P}(x_{m-1} = dx^{(m-1)}|x_{m-2} = x^{(m-2)}) \cdots \mathbb{P}(x_1 = dx^{(1)}|x_0 = x)$$

$$\left( \int_{\substack{\{\theta^{(k)}\}_{k=1}^{m-1} \\ \in C_\Theta^{m-1}}} \mathbb{P}(\theta_m \in A_{x'}|(x_{m-1}, \theta_{m-1}) = (x^{(m-1)}, \theta^{(m-1)})) \right.$$

$$\mathbb{P}(\theta_{m-1} = d\theta^{(m-1)}|(x_{m-2}, \theta_{m-2}) = (x^{(m-2)}, \theta^{(m-2)})))\Big)$$

$$\left. \cdots \mathbb{P}(\theta_1 = d\theta^{(1)}|(x_0, \theta_0) = (x, \theta)) \right).$$

Next, for $(x, \theta) \in \bar{C}$, we observe that

$$\mathbb{P}(\theta_k = d\theta'|(x_{k-1}, \theta_{k-1}) = (x, \theta)) = \mathbb{P}\Big(\xi_k(\theta) = d\big(\frac{\theta' - \theta}{\alpha} - g(x, \theta))|(x_k, \theta_k) = (x, \theta)\Big)$$

$$\geq \frac{1}{\alpha^d} p_\theta\Big(\frac{\theta' - \theta}{\alpha} - g(x, \theta)\Big) d\theta'$$

$$\geq \frac{1}{\alpha^d} \inf_{\hat\theta \in C_\Theta} p_{\hat\theta}\Big(\frac{\theta' - \theta}{\alpha} - g(x, \theta)\Big) d\theta'.$$

We next recall the linear growth assumption in Assumption 2 that $\|g(x, \theta)\| \leq L(\|\theta\| + 1)$. Hence, given $\theta \in C_\Theta$, $\forall x \in \mathcal{X}$, we have

$$\|\theta + \alpha g(x, \theta)\| \leq \|\theta\| + \alpha\|g(x, \theta)\| \leq (1 + \alpha L)(\|\theta\| + 1)$$

$$\leq (1 + \alpha L)(\sqrt{M/\beta} + \|\theta^*\| + 1) \leq B,$$

for some bounded value $B$.

We now define the measure $\varsigma^\dagger$ on $\mathbb{R}^d$ as $\varsigma^\dagger(A) = \frac{1}{\alpha^d} \inf_{\|z\| \leq B} \int_{\theta' \in A \cap C_\Theta} \inf_{\hat\theta \in C_\Theta} p_{\hat\theta}\Big(\frac{\theta' - z}{\alpha}\Big) d\theta'$.
Hence, it is easy to see that $\varsigma^\dagger(C_\Theta^c) = 0$. Moreover, we note the following property of measure $\varsigma^\dagger$.

**Claim 1.** *For $A \subseteq C_\Theta$ and $\lambda(A) > 0$, $\varsigma^\dagger(A) > 0$, where $\lambda$ denotes the Lebesgue measure.*

We delay the proof to the end. Taking the claim as true, we derive that

$$\int_{\substack{\{\theta^{(k)}\}_{k=1}^{m-1} \\ \in C_\Theta^{m-1}}} \mathbb{P}(\theta_m \in A_{x'}|(x_{m-1}, \theta_{m-1}) = (x^{(m-1)}, \theta^{(m-1)}))$$

$$\mathbb{P}(\theta_{m-1} = d\theta^{(m-1)}|(x_{m-2}, \theta_{m-2}) = (x^{(m-2)}, \theta^{(m-2)})))\Big) \cdots \mathbb{P}(\theta_1 = d\theta^{(1)}|(x_0, \theta_0) = (x, \theta))$$

$$= \int_{\theta^{(m)} \in A_{x'}} \int_{\theta^{(m-1)} \in C_\Theta} \mathbb{P}(\theta_m d\theta^{(m)}|(x_{m-1}, \theta_{m-1}) = (x^{(m-1)}, \theta^{(m-1)}))$$

$$\int_{\theta^{(m-2)} \in C_\Theta} \mathbb{P}(\theta_{m-1} = d\theta^{(m-1)}|(x_{m-2}, \theta_{m-2}) = (x^{(m-2)}, \theta^{(m-2)})))\Big)$$

$$\cdots$$

$$\int_{\theta^{(1)} \in C_\Theta} \mathbb{P}(\theta_2 = d\theta^{(2)}|(x_1, \theta_1) = (x^{(1)}, \theta^{(1)}))\mathbb{P}(\theta_1 = d\theta^{(1)}|(x_0, \theta_0) = (x, \theta))$$

$$\geq \int_{\theta^{(m)} \in A_{x'}} \int_{\theta^{(m-1)} \in C_\Theta} \frac{1}{\alpha^d} \inf_{\hat\theta \in C_\Theta} p_{\hat\theta}\Big(\frac{\theta^{(m)} - \theta^{(m-1)}}{\alpha} - g(x^{(m-1)}, \theta^{(m-1)})\Big) d\theta^{(m)}$$

$$\int_{\theta^{(m-2)} \in C_\Theta} \mathbb{P}(\theta_{m-1} = d\theta^{(m-1)}|(x_{m-2}, \theta_{m-2}) = (x^{(m-2)}, \theta^{(m-2)})))\Big)$$

$$\cdots$$

$$\int_{\theta^{(1)}\in C_\Theta} \mathbb{P}(\theta_2 = \mathrm{d}\theta^{(2)}|(x_1,\theta_1) = (x^{(1)},\theta^{(1)}))\mathbb{P}(\theta_1 = \mathrm{d}\theta^{(1)}|(x_0,\theta_0) = (x,\theta))$$

$$\geq \inf_{(\tilde{x},\tilde{\theta})\in\bar{C}} \int_{\theta^{(m)}\in A_{x'}} \int_{\theta^{(m-1)}\in C_\Theta} \frac{1}{\alpha^d} \inf_{\hat{\theta}\in C_\Theta} p_{\hat{\theta}}\Big(\frac{\theta^{(m)} - \tilde{\theta}}{\alpha} - g(\tilde{x},\tilde{\theta})\Big)\mathrm{d}\theta^{(m)}$$

$$\int_{\theta^{(m-2)}\in C_\Theta} \mathbb{P}(\theta_{m-1} = \mathrm{d}\theta^{(m-1)}|(x_{m-2},\theta_{m-2}) = (x^{(m-2)},\theta^{(m-2)})))\Big)$$

$$\cdots$$

$$\int_{\theta^{(1)}\in C_\Theta} \mathbb{P}(\theta_2 = \mathrm{d}\theta^{(2)}|(x_1,\theta_1) = (x^{(1)},\theta^{(1)}))\mathbb{P}(\theta_1 = \mathrm{d}\theta^{(1)}|(x_0,\theta_0) = (x,\theta))$$

$$\geq \inf_{\|z\|\leq B} \int_{\theta^{(m)}\in A_{x'}} \int_{\theta^{(m-1)}\in C_\Theta} \frac{1}{\alpha^d} \inf_{\hat{\theta}\in C_\Theta} p_{\hat{\theta}}\Big(\frac{\theta^{(m)} - z}{\alpha}\Big)\mathrm{d}\theta^{(m)}$$

$$\int_{\theta^{(m-2)}\in C_\Theta} \mathbb{P}(\theta_{m-1} = \mathrm{d}\theta^{(m-1)}|(x_{m-2},\theta_{m-2}) = (x^{(m-2)},\theta^{(m-2)})))\Big)\cdots$$

$$\int_{\theta^{(1)}\in C_\Theta} \mathbb{P}(\theta_2 = \mathrm{d}\theta^{(2)}|(x_1,\theta_1) = (x^{(1)},\theta^{(1)}))\mathbb{P}(\theta_1 = \mathrm{d}\theta^{(1)}|(x_0,\theta_0) = (x,\theta))$$

$$\geq \varsigma^\dagger(A_{x'}) \int_{\theta^{(m-1)}\in C_\Theta} \int_{\theta^{(m-2)}\in C_\Theta} \mathbb{P}(\theta_{m-1} = \mathrm{d}\theta^{(m-1)}|(x_{m-2},\theta_{m-2}) = (x^{(m-2)},\theta^{(m-2)})))\Big)\cdots$$

$$\int_{\theta^{(1)}\in C_\Theta} \mathbb{P}(\theta_2 = \mathrm{d}\theta^{(2)}|(x_1,\theta_1) = (x^{(1)},\theta^{(1)}))\mathbb{P}(\theta_1 = \mathrm{d}\theta^{(1)}|(x_0,\theta_0) = (x,\theta))$$

$$\geq \varsigma^\dagger(A_{x''})\varsigma^\dagger(C_\Theta)^{m-1}.$$

Therefore, by combining all the analyses, we obtain

$$Q^m((x,\theta),\bar{A})$$

$$\geq \int_{x'\in\mathcal{X}} \int_{\substack{\{x^{(k)}\}_{k=1}^{m-1} \\ \in\mathcal{X}^{m-1}}} \mathbb{P}(x_m = \mathrm{d}x'|x_{m-1} = x^{(m-1)})\mathbb{P}(x_{m-1} = \mathrm{d}x^{(m-1)}|x_{m-2} = x^{(m-2)})$$

$$\cdots \mathbb{P}(x_1 = \mathrm{d}x^{(1)}|x_0 = x)$$

$$\Big( \int_{\substack{\{\theta^{(k)}\}_{k=1}^{m-1} \\ \in C_\Theta^{m-1}}} \mathbb{P}(\theta_m \in A_{x'}|(x_{m-1},\theta_{m-1}) = (x^{(m-1)},\theta^{(m-1)}))$$

$$\mathbb{P}(\theta_{m-1} = \mathrm{d}\theta^{(m-1)}|(x_{m-2},\theta_{m-2}) = (x^{(m-2)},\theta^{(m-2)})))\Big)$$

$$\cdots \mathbb{P}(\theta_1 = \mathrm{d}\theta^{(1)}|(x_0,\theta_0) = (x,\theta))\Big)$$

$$\geq \varsigma^\dagger(C_\Theta)^{m-1} \int_{x'\in\mathcal{X}} \varsigma^\dagger(A_{x'})$$

$$\int_{\substack{\{x^{(k)}\}_{k=1}^{m-1} \\ \in\mathcal{X}^{m-1}}} \mathbb{P}(x_m = \mathrm{d}x'|x_{m-1} = x^{(m-1)})\cdots\mathbb{P}(x_1 = \mathrm{d}x^{(1)}|x_0 = x)$$

$$\geq \varsigma^\dagger(C_\Theta)^{m-1} \int_{x'\in\mathcal{X}} \varsigma^\dagger(A_{x'})\phi(\mathrm{d}x')$$

$$= \zeta \cdot (\phi \times \varsigma^\dagger)(\bar{A}),$$

where $\zeta = \varsigma^\dagger(C_\Theta)^{m-1}$ and $\phi \times \varsigma^\dagger$ being the unique induced product measure on $\mathcal{X} \times \mathbb{R}^d$.

As such, we have proven that $\bar{C}$ is $(m, \phi \times \varsigma^\dagger)$-small, and hence $(\delta_m, \phi \times \varsigma^\dagger)$-petite, and subsequently shown that $(x_k, \theta_k)_{k\geq 0}$ is geometrically ergodic.

By [50, Theorem 16.0.1 (iv)], we can further conclude that the geometrically ergodic $(\theta_t, x_t)_{t \geq 0}$ is also $V$-uniformly ergodic with the same $V$ as defined in (H.2). Therefore, we have the following convergence rate in the $V$-norm $\left\|\mathcal{L}(x_k, \theta_k), -\bar{\nu}_\alpha\right\|_V \leq \kappa \rho^k$, where $\kappa$ and $\rho$ implicitly depend on the stepsize $\alpha$.

Lastly, we provide the proof of Claim 1.

*Proof.* We first recall that for any set $B \subseteq \mathbb{R}^d$ such that $\lambda(B) > 0$, where $\lambda$ refers to the Lebesgue measure, then for $\theta \in C_\Theta$, we know that $\int_{t \in B} p_\theta(t) \mathrm{d}t \geq \int_{t \in B} \inf_{\theta \in C_\Theta} p_\theta(t) \mathrm{d}t > 0$. Moreover, for a given (translation) $z \in \mathbb{R}^d$ and $\theta \in C_\Theta$, we have

$$\int_{t \in B} p_\theta(t - z) \mathrm{d}t = \int_{t' \in B - z} p_\theta(t') \mathrm{d}t' \geq \int_{t' \in B - z} \inf_{\theta \in C} p_\theta(t') \mathrm{d}t' > 0.$$

Following the properties stated above, we define $h(z) = \int_{t \in B} \inf_{\theta \in C} p_\theta(t - z) \mathrm{d}t$, and it is easy to verify that $h(z)$ is a continuous function. Hence, for a bounded set $D$, we have $\inf_{z \in D} h(z) > 0$. Subsequently, we define the measure $\varsigma^\dagger$ induced by $h$ and we verify that

$$\varsigma^\dagger(A) = \inf_{\|z\| \leq B} \int_{t \in A \cap C_\Theta} \inf_{\theta \in C} p_\theta(t - z) \mathrm{d}t > 0.$$

$\square$

## H.2 Proof of Corollary 4.4

As shown in the previous section, the joint process $(x_k, \theta_k)_{k \geq 0}$ is $V$-uniformly ergodic and hence also exhibits a geometric non-asymptotic convergence rate under the $V$-weighted norm. Subsequently, this corresponds to a version of Corollary 4.4 resulting in a different set of convergence rate coefficients. Thus, we restate Corollary 4.4 in the context of the minorization setting and provide the proof below.

**Corollary H.2** (Non-Asymptotic Convergence Rate). *For any initialization of $\theta_0 \in \mathbb{R}^d$, under the setting of Theorem 4.3, we have*

$$\left\|\mathbb{E}[\theta_k] - \mathbb{E}[\theta_\infty^{(\alpha)}]\right\| \leq \kappa \cdot \rho^k \cdot s'(\theta_0, L, \mu), \quad \text{and} \quad \left\|\mathbb{E}[\theta_k \theta_k^\top] - \mathbb{E}[\theta_\infty^{(\alpha)}(\theta_\infty^{(\alpha)})^\top]\right\| \leq \kappa \cdot \rho^k \cdot s''(\theta_0, L, \mu),$$

*where $\kappa$ and $\rho$ are defined in (4.1) and implicitly depend on $\alpha$.*

*Proof.* For $V$-uniformly ergodic Markov chain $(x_k, \theta_k)_{k \geq 0}$, when functions $f : \mathcal{X} \times \mathbb{R}^d \to \mathbb{R}^d$ is dominated by the Lyapunov function, i.e., $\|f\| \leq V$, it enjoys the following convergence property,

$$\|Q^n f(x_0, \theta_0) - \pi f\| \leq \kappa \rho^n V(\theta_0).$$

Consider test function $f(\theta, x) = \theta - \theta^*$. It is easy to see that $\|f\| \leq V$. Hence, we obtain

$$\|\mathbb{E}[\theta_n] - \mathbb{E}[\theta_\infty]\| = \|Q^n f(x, \theta) - \pi f\| \leq \kappa \rho^n V(\theta_0).$$

Next, consider $f'(x) = (\theta - \theta^*)(\theta - \theta^*)^\top$. Clearly, $\|f'\| \leq V$. Therefore,

$$\|\mathbb{E}[(\theta_t - \theta^*)(\theta_t - \theta^*)^\top] - \mathbb{E}[(\theta_\infty - \theta^*)(\theta_\infty - \theta^*)^\top]\| \leq \kappa \rho^n V(\theta_0).$$

For the LHS, we have

$$\begin{aligned} &\|\mathbb{E}[(\theta_t - \theta^*)(\theta_t - \theta^*)^\top] - \mathbb{E}[(\theta_\infty - \theta^*)(\theta_\infty - \theta^*)^\top]\| \\ &= \|\mathbb{E}[\theta_t \theta_t^\top] - \mathbb{E}[\theta_\infty \theta_\infty^\top] - \mathbb{E}[\theta_t - \theta_\infty](\theta^*)^\top - \theta^* \mathbb{E}[(\theta_t - \theta_\infty)^\top]\| \\ &\geq \|\mathbb{E}[\theta_t \theta_t^\top] - \mathbb{E}[\theta_\infty \theta_\infty^\top]\| - 2\|\mathbb{E}[\theta_t - \theta_\infty]\|\|\theta^*\|. \end{aligned}$$

Subsequently,

$$\|\mathbb{E}[\theta_t \theta_t^\top] - \mathbb{E}[\theta_\infty \theta_\infty^\top]\| \leq (2\|\theta^*\| + 1) \kappa \rho^n V(\theta_0).$$

$\square$

The above results imply the convergence of the first two moments. Moreover, we conclude that

$$\mathrm{Var}(\theta_\infty) = \mathrm{Var}(\theta_\infty - \theta^*) \leq \mathbb{E}[\|\theta_\infty - \theta^*\|^2] = \lim_{t \to \infty} \mathbb{E}[\|\theta_t - \theta^*\|^2] \lesssim \alpha \tau.$$

Additionally,

$$\mathbb{E}[\|\theta_\infty - \theta^*\|]^2 \leq \mathbb{E}[\|\theta_\infty - \theta^*\|^2] \lesssim \alpha \tau. \tag{H.4}$$

## H.3 Proof of Corollary 4.5

After establishing the $V$-uniform ergodicity of the joint process $(x_k, \theta_k)_{k \geq 0}$, the central limit theorem for averaged iterates follows as a straightforward consequence.

*Proof.* For any test function $h : \mathcal{X} \times \mathbb{R}^d \to \mathbb{R}^d$ that satisfies $\|h\|^2 \leq V$, by Theorem 17.0.1 in [50], it has the following CLT results,

$$\frac{1}{\sqrt{k}} \Big[ \sum_{t=0}^{k-1} \big( h_t - \mathbb{E}[h_\infty] \big) \Big] \Rightarrow \mathcal{N}(0, \Sigma^{(a)}).$$

Therefore, consider $h(x, \theta) = \theta - \theta^*$, it is easy to see that $\|h\|^2 \leq V$, and hence we naturally obtain the desired CLT result, $\frac{1}{\sqrt{k}} \Big[ \sum_{t=0}^{k-1} \big( \theta_t - \mathbb{E}[\theta_\infty] \big) \Big] \Rightarrow \mathcal{N}(0, \Sigma^{(a)})$ as $k \to \infty$. $\qquad \square$

# I  Proof of Theorem 4.6

We now provide the proof of Theorem 4.6 on characterizing the asymptotic bias of nonlinear SA.

## I.1  BAR and Preliminaries

The proof utilizes the basic adjoint relationship (BAR) approach to study the stationary distribution

$$\mathbb{E}_{\bar{\nu}}[(P - I)h(\theta, x)] = 0,$$

via carefully designed test functions $h$. We refer readers to [33] for the derivation of the following properties of Markovian SA at stationarity,

$$\mathbb{E}[\mathbb{1}\{\theta_\infty \in S\} \mid x_\infty](x) = \mathbb{E}[\mathbb{1}\{\theta_{\infty+1} \in S\} \mid x_{\infty+1}](x), \tag{I.1}$$

$$\mathbb{E}[\theta_\infty \mid x_\infty](x) = \mathbb{E}[\theta_{\infty+1} \mid x_{\infty+1}](x), \tag{I.2}$$

$$\mathbb{E}[(\theta_\infty)^{\otimes 2} \mid x_\infty](x) = \mathbb{E}[(\theta_{\infty+1})^{\otimes 2} \mid x_{\infty+1}](x). \tag{I.3}$$

Following the Borel state space assumption in 1, $\mathbb{E}[\mathbb{1}\{\theta_\infty \in \cdot\} | x_\infty = x]$ induce a regular conditional probability measure, which we denote as $\tilde{\nu}(\cdot, x_\infty = x)$, and hence (I.1) can be reformulated as

$$\tilde{\nu}(\theta_\infty \in S, x_\infty = x) = \tilde{\nu}(\theta_{\infty+1} \in S, x_{\infty+1} = x).$$

Before proceeding to the proof, we introduce the following shorthands and notations. For $x \in \mathcal{X}$,

$$z_i(x) := \mathbb{E}[(\theta_\infty - \theta^*)^{\otimes i} | x_\infty = x],$$
$$\delta_i(x) := z_i(x) - \pi z_i,$$

Following the differentiability assumption of $g$ in Assumption 3, and we can apply Taylor expansion to $g$ and we note the following notation on residuals.

$$g(\theta, x) = g(\theta^*, x) + g'(\theta^*, x)(\theta - \theta^*) + \frac{1}{2} g''(\theta^*, x)(\theta - \theta^*)^{\otimes 2} + R_3(\theta, x) \tag{I.4}$$

$$= g(\theta^*, x) + g'(\theta^*, x)(\theta - \theta^*) + R_2(\theta, x). \tag{I.5}$$

By Assumption 3 and results from Proposition 4.2 and 4.2, we note that the residual $R_n(\theta, x)$ satisfies

$$\sup_{x \in \mathcal{X}, \theta \in \mathbb{R}^d} \left\{ \|R_n(\theta, x)\| / \|\theta - \theta^*\|^n \right\} < +\infty.$$

Hence, we have

$$\|R_n(\theta, x_\infty)\|_{L^2(\pi)} \lesssim \mathbb{E}[\|\theta_\infty - \theta^*\|^n] = \mathcal{O}((\alpha \tau)^{n/2}), \quad n = 2, 3, 4.$$

Lastly, we denote

$$\bar{g}(\theta) := \mathbb{E}_{x \sim \pi}[g(\theta, x)], \quad \bar{g}_2(\theta) := \mathbb{E}_{x \sim \pi}[(g(\theta, x))^{\otimes 2}]$$

$$\bar{g}^{(1)}(\theta) := \mathbb{E}_{x \sim \pi}[g'(\theta, x)], \quad \bar{g}_2^{(1)}(\theta) := \mathbb{E}_{x \sim \pi}[(g'(\theta, x))^{\otimes 2}], \quad \bar{g}^{(2)}(\theta) := \mathbb{E}_{x \sim \pi}[g''(\theta, x)].$$

We are now ready to present our proof. The proof consists of two major steps. For the complexity of this problem, we first set aside the projection constraint and focus on the BAR analysis, and we shall present the asymptotic bias characterization without the projection step. The analysis in this step thus shall work for the minorization proof technique as well. Then, in the second step, we elaborate on the impact brought along by the projection analysis and conclude our proof.

### I.2 Step 1: Bias Characterization without Projection

#### I.2.1 Step 1: First Moment Analysis

Consider test function $h_1(x, \theta) = \theta - \theta^*$. Therefore, we first have

$$\mathbb{E}[\theta_{\infty+1} - \theta^*] = \mathbb{E}[\theta_\infty - \theta^*] + \alpha\Big(\mathbb{E}[g(\theta_\infty, x_\infty)] + \mathbb{E}[\xi_{\infty+1}(\theta_\infty)]\Big),$$

which immediately implies that

$$0 = \mathbb{E}[g(\theta_\infty, x_\infty)]. \tag{I.6}$$

Substituting the Taylor expansion (I.4) back into (I.6), we have

$$0 = \mathbb{E}[g(\theta^*, x)] + \mathbb{E}[g'(\theta^*, x_\infty)(\theta_\infty - \theta^*)] + \frac{1}{2}\mathbb{E}[g''(\theta^*, x_\infty)(\theta_\infty - \theta^*)^{\otimes 2}] + \mathbb{E}[R_3(\theta_\infty, x_\infty)]$$

$$= \mathbb{E}[g'(\theta^*, x_\infty)(\theta_\infty - \theta^*)] + \frac{1}{2}\mathbb{E}[g''(\theta^*, x_\infty)(\theta_\infty - \theta^*)^{\otimes 2}] + \mathcal{O}((\alpha\tau)^{3/2}), \tag{I.7}$$

where we make use of $\mathbb{E}[g(\theta^*, x)] = \bar{g}(\theta^*) = 0$ by definition and the order or $\mathbb{E}[R_3(\theta_\infty, x_\infty)] = \mathcal{O}((\alpha\tau)^{3/2})$ to obtain the second equality.

Next, we proceed to analyze the two terms in (I.7). For the first term, we have

$$\mathbb{E}[g'(\theta^*, x_\infty)(\theta_\infty - \theta^*)] = \mathbb{E}[g'(\theta^*, x_\infty)z_1(x_\infty)]$$

$$= \mathbb{E}[g'(\theta^*, x_\infty)\delta_1(x_\infty)] + \bar{g}^{(1)}(\theta^*)\mathbb{E}[\theta_\infty - \theta^*]. \tag{I.8}$$

We now move on to analyze the second term, and obtain

$$\mathbb{E}[g''(\theta^*, x_\infty)(\theta_\infty - \theta^*)^{\otimes 2}] = \mathbb{E}[g''(\theta^*, x_\infty)z_2(x_\infty)]$$

$$= \mathbb{E}[g''(\theta^*, x_\infty)\delta_2(x_\infty)] + \bar{g}^{(2)}(\theta^*)\mathbb{E}[(\theta_\infty - \theta^*)^{\otimes 2}]. \tag{I.9}$$

Hence, substituting (I.8) and (I.9) back into (I.7), we reorganize the terms and arrive at

$$\mathbb{E}[\theta_\infty - \theta^*]$$

$$= -(\bar{g}^{(1)}(\theta^*))^{-1}\Big(\mathbb{E}[g'(\theta^*, x_\infty)\delta_1(x_\infty)] \tag{I.10}$$

$$+ \frac{1}{2}\Big(\mathbb{E}[g''(\theta^*, x_\infty)\delta_2(x_\infty)] + \bar{g}^{(2)}(\theta^*)\mathbb{E}[(\theta_\infty - \theta^*)^{\otimes 2}]\Big)\Big) \tag{I.11}$$

$$+ \mathcal{O}((\alpha\tau)^{3/2}).$$

To obtain a refined characterization of the asymptotic bias, we carefully analyze the remaining three terms in (I.10)–(I.11). We focus on each term in the next three sections respectively.

#### I.2.2 Step 2: Second Moment Analysis

We start with analyzing $\mathbb{E}[(\theta_\infty - \theta^*)^{\otimes 2}]$ in this section.

Following the BAR approach, we consider the test function $h_2(x, \theta) = (\theta - \theta^*)^{\otimes 2}$ and obtain

$$\mathbb{E}[(\theta_{\infty+1} - \theta^*)^{\otimes 2}] = \mathbb{E}[(\theta_\infty - \theta^* + \alpha(g(\theta_\infty, x_\infty) + \xi_{\infty+1}(\theta_\infty))^{\otimes 2}]$$

$$= \mathbb{E}[(\theta_\infty - \theta^*)^{\otimes 2}] + \alpha^2(\mathbb{E}[(g(\theta_\infty, x_\infty))^{\otimes 2}] + \mathbb{E}[(\xi_{\infty+1}(\theta_\infty))^{\otimes 2}])$$

$$+ \alpha(\mathbb{E}[g(\theta_\infty, x_\infty) \otimes (\theta_\infty - \theta^*)] + \mathbb{E}[(\theta_\infty - \theta^*) \otimes g(\theta_\infty, x_\infty)]).$$

Simplifying the above expression, we have

$$0 = \alpha(\mathbb{E}[(g(\theta_\infty, x_\infty))^{\otimes 2}] + \mathbb{E}[(\xi_{\infty+1}(\theta_\infty))^{\otimes 2}])$$

$$+ (\mathbb{E}[(\theta_\infty - \theta^*) \otimes g(\theta_\infty, x_\infty)] + \mathbb{E}[g(\theta_\infty, x_\infty) \otimes (\theta_\infty - \theta^*)]). \tag{I.12}$$

We adopt a similar approach in analyzing the above relationship that contains $g(\theta_\infty, x_\infty)$ as in the previous step. We make use of the Taylor expansion of $g$ at $\theta^*$ but at a lower order. We substitute the Taylor expansion (I.5) into (I.12) and obtain

$$0 = \alpha\mathbb{E}[(g(\theta^*, x_\infty) + g'(\theta^*, x_\infty)(\theta_\infty - \theta^*) + R_2(\theta_\infty, x_\infty))^{\otimes 2}] + \alpha\mathbb{E}[(\xi_{\infty+1}(\theta_\infty))^{\otimes 2}]$$

$$+ \mathbb{E}[(g(\theta^*, x_\infty) + g'(\theta^*, x_\infty)(\theta_\infty - \theta^*) + R_2(\theta_\infty, x_\infty)) \otimes (\theta_\infty - \theta^*)]$$
$$+ \mathbb{E}[(\theta_\infty - \theta^*) \otimes (g(\theta^*, x_\infty) + g'(\theta^*, x_\infty)(\theta_\infty - \theta^*) + R_2(\theta_\infty, x_\infty))]$$
$$= \mathbb{E}[g(\theta^*, x_\infty) \otimes (\theta_\infty - \theta^*)] + \mathbb{E}[(\theta_\infty - \theta^*) \otimes g(\theta^*, x_\infty)] \tag{I.13}$$
$$+ \mathbb{E}[(g'(\theta^*, x_\infty)(\theta_\infty - \theta^*)) \otimes (\theta_\infty - \theta^*)] + \mathbb{E}[(\theta_\infty - \theta^*) \otimes (g'(\theta^*, x_\infty)(\theta_\infty - \theta^*))] \tag{I.14}$$
$$+ \alpha \mathbb{E}[(g'(\theta^*, x_\infty)(\theta_\infty - \theta^*))^{\otimes 2}] \tag{I.15}$$
$$+ \alpha \mathbb{E}[g(\theta^*, x_\infty) \otimes (g'(\theta^*, x_\infty)(\theta_\infty - \theta^*))] + \alpha \mathbb{E}[(g'(\theta^*, x_\infty)(\theta_\infty - \theta^*)) \otimes g(\theta^*, x_\infty)] \tag{I.16}$$
$$+ \mathbb{E}[R_2(\theta_\infty, x_\infty) \otimes (\theta_\infty - \theta^*)] + \mathbb{E}[(\theta_\infty - \theta^*) \otimes R_2(\theta_\infty, x_\infty)]$$
$$+ \alpha \mathbb{E}[(g(\theta^*, x_\infty))^{\otimes 2}] + \alpha \mathbb{E}[(\xi_{\infty+1}(\theta_\infty))^{\otimes 2}] + \mathcal{O}(\alpha^2 \tau).$$

Therefore, we proceed to analyze the terms in (I.13)–(I.16).

Starting with the terms in (I.13), we have

$$\mathbb{E}[g(\theta^*, x_\infty) \otimes (\theta_\infty - \theta^*)] = \mathbb{E}[g(\theta^*, x_\infty) \otimes (\delta_1(x_\infty) + \pi z_1)]$$
$$= \mathbb{E}[g(\theta^*, x_\infty) \otimes \delta_1(x_\infty)] + \underbrace{\mathbb{E}[g(\theta^*, x_\infty)]}_{=0} \otimes \mathbb{E}[\theta_\infty - \theta^*]$$
$$= \mathbb{E}[g(\theta^*, x_\infty) \otimes \delta_1(x_\infty)].$$

Similarly,

$$\mathbb{E}[(\theta_\infty - \theta^*) \otimes g(\theta^*, x_\infty)] = \mathbb{E}[\delta_1(x_\infty) \otimes g(\theta^*, x_\infty)].$$

Next, for the terms in (I.14), we have

$$\mathbb{E}[(g'(\theta^*, x_\infty)(\theta_\infty - \theta^*)) \otimes (\theta_\infty - \theta^*)] = \mathbb{E}[g'(\theta^*, x_\infty)(\theta_\infty - \theta^*)^{\otimes 2}]$$
$$= \mathbb{E}[g'(\theta^*, x_\infty)(\delta_2(x_\infty) + \pi z_2)]$$
$$= \mathbb{E}[g'(\theta^*, x_\infty)\delta_2(x_\infty)] + \bar{g}^{(1)}(\theta^*)\mathbb{E}[(\theta_\infty - \theta^*)^{\otimes 2}].$$

Similarly,

$$\mathbb{E}[(\theta_\infty - \theta^*) \otimes (g'(\theta^*, x_\infty)(\theta_\infty - \theta^*))] = \mathbb{E}[\delta_2(x_\infty)g'(\theta^*, x_\infty)] + \mathbb{E}[(\theta_\infty - \theta^*)^{\otimes 2}]\bar{g}^{(1)}(\theta^*).$$

Moving on to the second term in (I.15)

$$\mathbb{E}[(g'(\theta^*, x_\infty)(\theta_\infty - \theta^*))^{\otimes 2}] = \mathbb{E}[g'(\theta^*, x_\infty)(\theta_\infty - \theta^*)^{\otimes 2}g'(\theta^*, x_\infty)]$$
$$= \mathbb{E}[g'(\theta^*, x_\infty)(\delta_2(x_\infty) + \pi z_2)g'(\theta^*, x_\infty)]$$
$$= \mathbb{E}[g'(\theta^*, x_\infty)\delta_2(x_\infty)g'(\theta^*, x_\infty)] + \bar{g}_2^{(1)}(\theta^*)\mathbb{E}[(\theta_\infty - \theta^*)^{\otimes 2}].$$

Last, for the terms in (I.16)

$$\mathbb{E}[g(\theta^*, x_\infty) \otimes (g'(\theta^*, x_\infty)(\theta_\infty - \theta^*))] = \mathbb{E}[g(\theta^*, x_\infty) \otimes (g'(\theta^*, x_\infty)(\delta_1(x_\infty) + \pi z_1))]$$
$$= \mathbb{E}[g(\theta^*, x_\infty) \otimes (g'(\theta^*, x_\infty)\delta_1(x_\infty))]$$
$$+ \mathbb{E}[g(\theta^*, x_\infty) \otimes (g'(\theta^*, x_\infty)\mathbb{E}[\theta_\infty - x_\infty])].$$

Similarly,

$$\mathbb{E}[(g'(\theta^*, x_\infty)(\theta_\infty - \theta^*)) \otimes g(\theta^*, x_\infty)] = \mathbb{E}[(g'(\theta^*, x_\infty)\delta_1(x_\infty)) \otimes g(\theta^*, x_\infty)]$$
$$+ \mathbb{E}[(g'(\theta^*, x_\infty)\mathbb{E}[\theta_\infty - x_\infty]) \otimes g(\theta^*, x_\infty)].$$

Lastly, by a second order Taylor expansion around $\theta^*$ of $C(\theta_\infty) = \mathbb{E}[(\xi_{\infty+1}(\theta_\infty))^{\otimes 2}]$, we have

$$C(\theta_\infty) = C(\theta^*) + \mathbb{C}'(\theta^*)\mathbb{E}[\theta_\infty - \theta^*] + \mathbb{E}[R_2'(\theta_\infty)],$$

where $R_2'(\theta)$ satisfies $\sup_{x \in \mathbb{R}^d} \left\{ \|R_2'(\theta)\| / (\|\theta - \theta^*\|^2 + \|\theta - \theta^*\|^{k_\epsilon + 2}) \right\} < \infty.$

Substituting the above analyses of the terms and consolidating the terms, we obtain

$$
\begin{aligned}
&- (\bar{g}^{(1)}(\theta^*) \otimes I + I \otimes \bar{g}^{(1)}(\theta^*)) \mathbb{E}[(\theta_\infty - \theta^*)^{\otimes 2}] \\
&= \alpha \bar{g}_2(\theta^*) + \alpha \mathbb{E}[(\xi_{\infty+1}(\theta^*))^{\otimes 2}] + \alpha \bar{g}_2^{(1)}(\theta^*) \mathbb{E}[(\theta_\infty - \theta^*)^{\otimes 2}] \\
&\quad + \mathbb{E}[g(\theta^*, x_\infty) \otimes \delta_1(x_\infty)] + \mathbb{E}[\delta_1(x_\infty) \otimes g(\theta^*, x_\infty)] \\
&\quad + \mathbb{E}[g'(\theta^*, x_\infty) \delta_2(x_\infty)] + \mathbb{E}[\delta_2(x_\infty) g'(\theta^*, x_\infty)] \\
&\quad + \alpha \mathbb{E}[g(\theta^*, x_\infty) \otimes (g'(\theta^*, x_\infty) \delta_1(x_\infty))] + \alpha \mathbb{E}[g(\theta^*, x_\infty) \otimes (g'(\theta^*, x_\infty) \mathbb{E}[\theta_\infty - x_\infty])] \quad \text{(I.17)} \\
&\quad + \alpha \mathbb{E}[(g'(\theta^*, x_\infty) \delta_1(x_\infty)) \otimes g(\theta^*, x_\infty)] + \alpha \mathbb{E}[(g'(\theta^*, x_\infty) \mathbb{E}[\theta_\infty - x_\infty]) \otimes g(\theta^*, x_\infty)] \\
&\quad + \alpha \mathbb{E}[g'(\theta^*, x_\infty) \delta_2(x_\infty) g'(\theta^*, x_\infty)] \\
&\quad + \mathbb{E}[R_2(\theta_\infty, x_\infty) \otimes (\theta_\infty - \theta^*)] + \mathbb{E}[(\theta_\infty - \theta^*) \otimes R_2(\theta_\infty, x_\infty)] + \alpha C'(\theta^*) \mathbb{E}[\theta_\infty - \theta^*] \\
&\quad + \mathcal{O}(\alpha^2 \tau).
\end{aligned}
$$

By far, we observe that the remaining terms all contain $\delta_1$ and $\delta_2$. Therefore, we conclude our analysis of $\mathbb{E}[(\theta_\infty - \theta^*)^{\otimes 2}]$ at this step and leave the analysis of $\delta_1$ and $\delta_2$ to the next section.

### I.2.3 Step 3: Analysis of the $\delta$-System

In this section, we analyze $\delta_1$ and $\delta_2$.

**Analysis of $\delta_1$.** Starting with $\delta_1$, we first consider the following recursive relationship induced by (I.2).

$$
\begin{aligned}
\mathbb{E}[\theta_{\infty+1} - \theta^* | x_{\infty+1} = s] &= \int_{\mathbb{R}} P^*(s, ds') \mathbb{E}[\theta_{\infty+1} - \theta^* | x_{\infty+1} = s, x_\infty = s'] \\
&\stackrel{(i)}{=} \int_{\mathbb{R}} P^*(s, ds') \mathbb{E}[\theta_\infty - \theta^* + \alpha g(\theta_\infty, x_\infty) | x_\infty = s'] \\
&\stackrel{(ii)}{=} \int_{\mathbb{R}} P^*(s, ds') \mathbb{E}\Big[\theta_\infty - \theta^* + \alpha \Big(g(\theta^*, x_\infty) + g'(\theta^*, x_\infty)(\theta_\infty - \theta^*) + R_2(\theta_\infty, x_\infty)\Big) | x_\infty = s'\Big] \\
&= \int_{\mathbb{R}} P^*(s, ds') \mathbb{E}\Big[\theta_\infty - \theta^* | x_\infty = s'\Big] \\
&\quad + \alpha \int_{\mathbb{R}} P^*(s, ds') \Big(g(\theta^*, s') + g'(\theta^*, s') \mathbb{E}[\theta_\infty - \theta^* | x_\infty = s'] + \mathbb{E}[R_2(\theta_\infty, x_\infty) | x_\infty = s']\Big),
\end{aligned}
$$

where in (i) we make use of the update rule in (2.1), $\mathbb{E}[\xi_{\infty+1}(\theta_\infty)] = 0$ and conditional independence $x_{\infty+1} \perp\!\!\!\perp \theta_\infty | x_\infty$. Next, we substitute the Taylor expansion (I.5) to obtain (ii).

Writing with notation shorthands $z$ and $\delta$, we have

$$
\begin{aligned}
z_1(s) &= \int_{\mathbb{R}} P^*(s, ds') z_1(s') \\
&\quad + \alpha \int_{R} P^*(s, ds') \Big(g(\theta^*, s') + g'(\theta^*, s') z_1(s') + \mathbb{E}[R_2(\theta_\infty, s') | x_\infty = s']\Big).
\end{aligned}
\quad \text{(I.18)}
$$

If we apply $\pi$ to both sides of (I.18), we obtain

$$
\int \pi(ds) P^*(s, ds') \Big(g(\theta^*, s') + g'(\theta^*, s') z_1(s') + \mathbb{E}[R_2(\theta_\infty, s') | x_\infty = s']\Big) = 0.
$$

Analyzing the three terms closely, we observe that

$$
\int \pi(ds) \int P^*(s, ds') g(\theta^*, s') = \int g(\theta^*, s') \underbrace{\int \pi(ds) P^*(s, ds')}_{= \pi(ds')} = \bar{g}(\theta^*) = 0
$$

$$
\int \pi(ds) \int P^*(s, ds') g'(\theta^*, s') z_1(s') = \mathbb{E}[g'(\theta^*, x_\infty) z_1(x_\infty)]
$$

$$\int \pi(\mathrm{d}s) \int P^*(s, \mathrm{d}s') \mathbb{E}[R_2(\theta_\infty, s')|x_\infty = s'] = \mathbb{E}[R_2(\theta_\infty, x_\infty)],$$

and hence we first obtain

$$\mathbb{E}[g'(\theta^*, x_\infty)z_1(x_\infty)] + \mathbb{E}[R_2(\theta_\infty, x_\infty)] = 0. \tag{I.19}$$

We now subtract $\pi z_1$ on both sides of (I.18), and obtain

$$
\begin{aligned}
\delta_1(s) &= \int_{\mathbb{R}} P^*(s, \mathrm{d}s')\delta_1(s') \\
&\quad + \alpha \int_R P^*(s, \mathrm{d}s')\Big(g(\theta^*, s') + g'(\theta^*, s')z_1(s') + \mathbb{E}[R_2(\theta_\infty, s')|x_\infty = s']\Big) \\
&= \int_{\mathbb{R}} \Big(P^*(s, \mathrm{d}s') - \pi(\mathrm{d}s')\Big)\delta_1(s') \\
&\quad + \alpha \int_R \Big(P^*(s, \mathrm{d}s') - \pi(\mathrm{d}s')\Big)\Big(g(\theta^*, s') + g'(\theta^*, s')z_1(s') + \mathbb{E}[R_2(\theta_\infty, s')|x_\infty = s']\Big).
\end{aligned}
$$

Consolidating the terms, we have

$$
\begin{aligned}
&(I - P^* + \Pi)\delta_1(s) \\
&= \alpha \int_R \Big(P^*(s, \mathrm{d}s') - \pi(\mathrm{d}s')\Big)\Big(g(\theta^*, s') + g'(\theta^*, s')z_1(s') + \mathbb{E}[R_2(\theta_\infty, s')|x_\infty = s']\Big).
\end{aligned} \tag{I.20}
$$

We next note the following properties,

$$
\begin{aligned}
&\|\mathbb{E}[R_2(\theta_\infty, x_\infty)|x_\infty = s]\|_{L^2(\pi)}^2 \le \mathbb{E}[\|R_2(\theta_\infty, x_\infty)^2\|]] = \mathcal{O}((\alpha\tau)^2), \\
&\|z_1(s)\|_{L^2(\pi)}^2 \le \mathbb{E}[\|\theta_\infty - \theta^*\|^2] = \mathcal{O}(\alpha\tau) = \mathcal{O}(1).
\end{aligned}
$$

Therefore, we can first conclude that $\|\delta\|_{L^2(\pi)} = \mathcal{O}(\alpha)$.

Subsequently, from (I.19), we can derive that

$$
\begin{aligned}
0 &= \mathbb{E}[g'(\theta^*, x_\infty)z_1(x_\infty)] + \mathbb{E}[R_2(\theta_\infty, x_\infty)] \\
&= \mathbb{E}[g'(\theta^*, x_\infty)\delta_1(x_\infty)] + \bar{g}^{(1)}(\theta^*)\mathbb{E}[\theta_\infty - \theta^*] + \mathbb{E}[R_2(\theta_\infty, x_\infty)] \\
\mathbb{E}[\theta_\infty - \theta^*] &= -(\bar{g}^{(1)}(\theta^*))^{-1}\Big(\mathbb{E}[g'(\theta^*, x_\infty)\delta_1(x_\infty)] + \mathbb{E}[R_2(\theta_\infty, x_\infty)]\Big). \tag{I.21}
\end{aligned}
$$

Hence, together with the relationship between $\delta_1$ and $z_1$, we have

$$z_1 = \delta_1 - (\bar{g}^{(1)}(\theta^*))^{-1}\Big(\mathbb{E}[g'(\theta^*, x_\infty)\delta_1(x_\infty)] + \mathbb{E}[R_2(\theta_\infty, x_\infty)]\Big). \tag{I.22}$$

Lastly, we substitute (I.22) back into (I.20) and obtain

$$
\begin{aligned}
&(I - P^* + \Pi)\delta_1(s) \\
&= \alpha \int_R \Big(P^*(s, \mathrm{d}s') - \pi(\mathrm{d}s')\Big)g(\theta^*, s') \\
&\quad + \alpha \int_R \Big(P^*(s, \mathrm{d}s') - \pi(\mathrm{d}s')\Big)g'(\theta^*, s')\Big(\delta_1(s') - (\bar{g}^{(1)}(\theta^*))^{-1}\mathbb{E}[g'(\theta^*, x_\infty)\delta_1(x_\infty)]\Big) \\
&\quad - \alpha \int_R \Big(P^*(s, \mathrm{d}s') - \pi(\mathrm{d}s')\Big)g'(\theta^*, s')\Big((\bar{g}^{(1)}(\theta^*))^{-1}\mathbb{E}[R_2(\theta_\infty, x_\infty)]\Big) \\
&\quad + \alpha \int_R \Big(P^*(s, \mathrm{d}s') - \pi(\mathrm{d}s')\Big)\mathbb{E}[R_2(\theta_\infty, x_\infty)|x_\infty = s']\Big) \\
&= \alpha v(\theta^*, s) + \mathcal{O}(\alpha^2\tau),
\end{aligned}
$$

where

$$v(\theta^*, s) = (I - P^* + \Pi)^{-1}(P^* - \Pi)g_{\theta^*}(s) = \int_R (I - P^* + \Pi)^{-1}(P^* - \Pi)(s, \mathrm{d}s')g(\theta^*, s').$$

Therefore, for the terms that involve $\delta_1$, we can conclude that

$$\mathbb{E}[g(\theta^*, x_\infty) \otimes \delta_1(x_\infty)] = \alpha M + \mathcal{O}(\alpha^2 \tau) \quad \text{and} \quad \mathbb{E}[\delta_1(x_\infty) \otimes g(\theta^*, x_\infty)] = \alpha M + \mathcal{O}(\alpha^2 \tau),$$

$$\mathbb{E}[g'(\theta^*, x_\infty) \delta_1(x_\infty)] = \alpha v' + \mathcal{O}(\alpha^2 \tau), \tag{I.23}$$

where $M$ and $v$ are independent of $\alpha$.

Note that (I.23) together with (I.21) implies that

$$\mathbb{E}[\theta_\infty - \theta^*] = \alpha v^{(1)} + \mathcal{O}(\alpha \tau).$$

**Analysis of $\delta_2$.** For $z_2$, we first note that

$$|\mathbb{E}[(\theta_\infty - \theta^*)^{\otimes 2} | x_\infty]\|_{L^2(\pi)}^2 \le \mathbb{E}[\|\theta_\infty - \theta^*\|^4] = \mathcal{O}((\alpha \tau)^2),$$

and hence this implies that

$$\|z_2\|_{L^2(\pi)} = \mathcal{O}(\alpha \tau).$$

Next, following (I.3), we obtain the following recursive relationship.

$$\mathbb{E}[(\theta_{\infty+1} - \theta^*)^{\otimes 2} | x_{\infty+1} = s'] = \int P^*(s', \mathrm{d}s) \mathbb{E}[(\theta_{\infty+1} - \theta^*)^{\otimes 2} | x_{\infty+1} = s', x_\infty = s]$$

$$\stackrel{(i)}{=} \int P^*(s', \mathrm{d}s) \mathbb{E}[(\theta_\infty - \theta^* + \alpha(g(\theta_\infty, x_\infty) + \xi_{\infty+1}(\theta_\infty)))^{\otimes 2} | x_\infty = s]$$

$$\stackrel{(ii)}{=} \int P^*(s', \mathrm{d}s) \mathbb{E}[(\theta_\infty - \theta^*)^{\otimes 2} | x_\infty = s]$$

$$+ \alpha^2 \int P^*(s', \mathrm{d}s) \mathbb{E}[(g(\theta_\infty, s))^{\otimes 2} | x_\infty = s] + \alpha^2 \int P^*(s', \mathrm{d}s) \mathbb{E}[(\xi_{\infty+1}(\theta_\infty))^{\otimes 2} | x_\infty = s]$$

$$+ \alpha \int P^*(s', \mathrm{d}s) \Big( \mathbb{E}[(\theta_\infty - \theta^*) \otimes g(\theta_\infty, s) | x_\infty = s] + \mathbb{E}[g(\theta_\infty, s) \otimes (\theta_\infty - \theta^*) | x_\infty = s] \Big),$$

where in (i) we make use of the update rule in (2.1) and conditional independence $x_{\infty+1} \perp\!\!\!\perp \theta_\infty | x_\infty$. Next, we substitute the Taylor expansion (I.5) to obtain (ii).

Writing with $z_2$ shorthand, we have

$$z_2(s') = \int P^*(s', \mathrm{d}s) z_2(s)$$

$$+ \alpha^2 \int P^*(s', \mathrm{d}s) \mathbb{E}[(g(\theta_\infty, s))^{\otimes 2} | x_\infty = s] + \alpha^2 \int P^*(s', \mathrm{d}s) \mathbb{E}[(\xi_{\infty+1}(\theta_\infty))^{\otimes 2} | x_\infty = s]$$

$$+ \alpha \int P^*(s', \mathrm{d}s) \Big( \mathbb{E}[(\theta_\infty - \theta^*) \otimes g(\theta_\infty, s) | x_\infty = s] + \mathbb{E}[g(\theta_\infty, s) \otimes (\theta_\infty - \theta^*) | x_\infty = s] \Big).$$

Making use of the relationship $(P^* - \Pi)z_2 = (P^* - \Pi)\delta_2$, we have

$$\delta_2(s') = \int \Big( P^*(s', \mathrm{d}s) - \pi(\mathrm{d}s) \Big) \delta_2(s)$$

$$+ \alpha^2 \int P^*(s', \mathrm{d}s) \mathbb{E}[(g(\theta_\infty, s))^{\otimes 2} | x_\infty = s] + \alpha^2 \int P^*(s', \mathrm{d}s) \mathbb{E}[(\xi_{\infty+1}(\theta_\infty))^{\otimes 2} | x_\infty = s]$$

$$+ \alpha \int P^*(s', \mathrm{d}s) \Big( \mathbb{E}[(\theta_\infty - \theta^*) \otimes g(\theta_\infty, s) | x_\infty = s] + \mathbb{E}[g(\theta_\infty, s) \otimes (\theta_\infty - \theta^*) | x_\infty = s] \Big).$$

Hence,

$$(I - P^* + \Pi)\delta_2(s')$$

$$= \alpha \int P^*(s', \mathrm{d}s) \Big( \mathbb{E}[(\theta_\infty - \theta^*) \otimes g(\theta_\infty, s) | x_\infty = s] + \mathbb{E}[g(\theta_\infty, s) \otimes (\theta_\infty - \theta^*) | x_\infty = s]$$

$$+ \alpha^2 \int P^*(s', \mathrm{d}s) \Big( \mathbb{E}[g((\theta_\infty, s))^{\otimes 2} | x_\infty = s] + \mathbb{E}[(\xi_{\infty+1}(\theta_\infty))^{\otimes 2} | x_\infty = s] \Big).$$

To analyze the above system of $\delta_2$, we make use of the Taylor expansion of $g(\theta_\infty, s)$ and $\xi_{\infty+1}(\theta_\infty)$. Starting with the first term, we have

$$\mathbb{E}[(\theta_\infty - \theta^*) \otimes g(\theta_\infty, s)|x_\infty = s]$$
$$= \mathbb{E}\Big[\Big(\theta_\infty - \theta^*\Big) \otimes \Big(g(\theta^*, s) + g'(\theta^*, s)(\theta_\infty - \theta^*) + R_2(\theta, s)\Big)|x_\infty = s\Big]$$
$$= z_1(s) \otimes g(\theta^*, s) + z_2(s)g'(\theta^*, s) + \mathbb{E}[(\theta_\infty - \theta^*) \otimes R_2(\theta, s)|x_\infty = s]$$
$$= \Big(\delta_1(s) - (\bar{g}^{(1)}(\theta^*))^{-1}\Big(\mathbb{E}[g'(\theta^*, x_\infty)\delta_1(x_\infty)] + \mathbb{E}[R_2(\theta_\infty, x_\infty)]\Big)\Big) \otimes g(\theta^*, s)$$
$$+ z_2(s)g'(\theta^*, s) + \mathbb{E}[(\theta_\infty - \theta^*) \otimes R_2(\theta, s)|x_\infty = s].$$

Similarly, we have the following relationship for the second term,

$$\mathbb{E}[g(\theta_\infty, s) \otimes (\theta_\infty - \theta^*)|x_\infty = s]$$
$$= g(\theta^*, s) \otimes \Big(\delta_1(s) - (\bar{g}^{(1)}(\theta^*))^{-1}\Big(\mathbb{E}[g'(\theta^*, x_\infty)\delta_1(x_\infty)] + \mathbb{E}[R_2(\theta_\infty, x_\infty)]\Big)\Big)$$
$$+ g'(\theta^*, s)z_2(s) + \mathbb{E}[R_2(\theta, s) \otimes (\theta_\infty - \theta^*)|x_\infty = s].$$

Next, we proceed to analyze the third term.

$$\mathbb{E}[(g(\theta_\infty, s))^{\otimes 2}|x_\infty = s]$$
$$= \mathbb{E}\Big[\Big(g(\theta^*, s) + g'(\theta^*, s)(\theta_\infty - \theta^*) + R_2(\theta_\infty, s)\Big)^{\otimes 2}|x_\infty = s\Big]$$
$$= \mathbb{E}[(g(\theta^*, s))^{\otimes 2}|x_\infty = s] + \mathbb{E}[(g'(\theta^*, s)(\theta_\infty - \theta^*))^{\otimes 2}|x_\infty = s]$$
$$+ \mathbb{E}[g(\theta^*, s) \otimes (g'(\theta^*, s)(\theta_\infty - \theta^*))|x_\infty = s] + \mathbb{E}[(g'(\theta^*, s)(\theta_\infty - \theta^*)) \otimes g(\theta^*, s)|x_\infty = s]$$
$$+ \mathbb{E}[g(\theta^*, s) \otimes R_2(\theta_\infty, s)|x_\infty = s] + \mathbb{E}[R_2(\theta_\infty, s) \otimes g(\theta^*, s)|x_\infty = s]$$
$$+ \mathbb{E}[(g'(\theta^*, s)(\theta_\infty - \theta^*)) \otimes R_2(\theta_\infty, s)|x_\infty = s]$$
$$+ \mathbb{E}[R_2(\theta_\infty, s) \otimes (g'(\theta^*, s)(\theta_\infty - \theta^*))|x_\infty = s] + \mathbb{E}[(R_2(\theta_\infty, s))^{\otimes 2}|x_\infty = s].$$

Lastly, for the noise term, we derive that

$$\mathbb{E}[(\xi_{\infty+1}(\theta_\infty))^{\otimes 2}|x_\infty = s] = \mathbb{E}[(\xi_{\infty+1}(\theta^*))^{\otimes 2}] + C'(\theta^*)\mathbb{E}[\theta_\infty - \theta^*|x_\infty = s] + \mathbb{E}[R'_2(\theta_\infty)|x_\infty = s].$$

Leveraging on the respective orders, we can conclude that

$$\|\delta_2\|_{L^2(\pi)} = \mathcal{O}(\alpha^2\tau).$$

Therefore, for second-moment cross-terms, we have the following orders

$$\mathbb{E}[g'(\theta^*, x_\infty)\delta_2(x_\infty)] = \mathcal{O}(\alpha^2\tau) \quad \text{and} \quad \mathbb{E}[\delta_2(x_\infty)g'(\theta^*, x_\infty)] = \mathcal{O}(\alpha^2\tau),$$
$$\mathbb{E}[g''(\theta^*, x_\infty)\delta_2(x_\infty)] = \mathcal{O}(\alpha^2\tau). \tag{I.24}$$

### I.2.4   Step 4: Bias Characterization

Finally, we are ready to consolidate the above analyses and conclude the characterization of the asymptotic bias.

We recall that we have already shown the following expansion of the asymptotic bias

$$\mathbb{E}[\theta_\infty - \theta^*]$$
$$= -(\bar{g}^{(1)}(\theta^*))^{-1}\Big(\mathbb{E}[g'(\theta^*, x_\infty)\delta_1(x_\infty)] + \frac{1}{2}\Big(\mathbb{E}[g''(\theta^*, x_\infty)\delta_2(x_\infty)] + \bar{g}^{(2)}(\theta^*)\mathbb{E}[(\theta_\infty - \theta^*)^{\otimes 2}]\Big)\Big)$$
$$+ \mathcal{O}((\alpha\tau)^{3/2}).$$

By our analyses above, we have shown that

$$\delta_1 = \alpha(I - P^* + \Pi)^{-1}(P^* - \Pi)g^*_\theta + \mathcal{O}(\alpha^2\tau).$$

Hence, we derive that

$$\mathbb{E}[g'(\theta^*, x_\infty)\delta_1(x_\infty)] = \alpha\mathbb{E}[g'(\theta^*, x_\infty)(I - P^* + \Pi)^{-1}(P^* - \Pi)g^*_\theta(x_\infty)] + \mathcal{O}(\alpha^2\tau).$$

For the second term, we simply use the order shown in (I.24).

Lastly, we substitute our analyses of $\delta_1$ and $\delta_2$ into the expansion of MSE (I.17) and derive that

$$
\begin{aligned}
& - (\bar{g}^{(1)}(\theta^*) \otimes I + I \otimes \bar{g}^{(1)}(\theta^*)) \mathbb{E}[(\theta_\infty - \theta^*)^{\otimes 2}] \\
& = \alpha \bar{g}_2(\theta^*) + \alpha \mathbb{E}[(\xi_{\infty+1}(\theta^*))^{\otimes 2}] \\
& \quad + \alpha \mathbb{E}[g(\theta^*, x_\infty) \otimes (I - P^* + \Pi)^{-1}(P^* - \Pi) g_{\theta^*}(x_\infty)] \\
& \quad + \alpha \mathbb{E}[(I - P^* + \Pi)^{-1}(P^* - \Pi) g_{\theta^*}(x_\infty) \otimes g(\theta^*, x_\infty)] + \mathcal{O}(\alpha^2 \tau).
\end{aligned}
$$

Therefore, combining all the analyses, we have shown the bias characterization in Theorem 4.6,

$$
\begin{aligned}
& \mathbb{E}[\theta_\infty - \theta^*] \\
& = -\alpha \cdot (\bar{g}^{(1)}(\theta^*))^{-1} \mathbb{E}[g'(\theta^*, x_\infty) h(\theta^*, x_\infty)] \\
& \quad + \alpha \cdot \frac{1}{2} (\bar{g}^{(1)}(\theta^*))^{-1} (\bar{g}^{(1)} A (\bar{g}_2(\theta^*) + \alpha \mathbb{E}[(\xi_{\infty+1}(\theta^*))^{\otimes 2}])) \\
& \quad + \alpha \cdot \frac{1}{2} (\bar{g}^{(1)}(\theta^*))^{-1} A (\mathbb{E}[g(\theta^*, x_\infty) \otimes h(\theta^*, x_\infty)] + \mathbb{E}[h(\theta^*, x_\infty) \otimes g(\theta^*, x_\infty)]) + \mathcal{O}((\alpha\tau)^{3/2}).
\end{aligned}
$$

where

$$
\begin{aligned}
A &= (\bar{g}^{(1)}(\theta^*) \otimes I + I \otimes \bar{g}^{(1)}(\theta^*))^{-1}, \\
h(\theta^*, s) &= (I - P^* + \Pi)^{-1}(P^* - \Pi) g_{\theta^*}(s) \\
&= \int_{\mathcal{X}} (I - P^* + \Pi)^{-1}(P^* - \Pi)(s, ds') g(\theta^*, s').
\end{aligned}
$$

Therefore, we see that assuming weak convergence without projection, the bias admits a leading term of order $\alpha$. We emphasize that the expansion holds as equality, rather than an upper bound.

## I.3  Step 2: Impact of Projection on Bias

Now, we proceed to analyze the impact of having the additional projection step on the asymptotic bias characterization. In the following, we use the shorthand $\theta_{t+1/2}$ to denote the iterate we obtain before the projection step, i.e.,

$$
\theta_{t+1/2} = \theta_t + \alpha(g(\theta_t, x_t) + \xi_{t+1}(\theta_t)) \quad \text{and} \quad \theta_{t+1} = \Pi_{B(\beta)} \theta_{t+1/2}.
$$

Therefore, we see that our analysis from Step 1 can be understood as the analysis for $\theta_{\infty+1/2}$.

Starting with the first moment analysis with test function $h_1(x, \theta) = \theta - \theta^*$, we have

$$
\begin{aligned}
\mathbb{E}[\theta_{\infty+1} - \theta^*] &= \mathbb{E}[\theta_{\infty+1/2} - \theta^*] + \mathbb{E}[\theta_{\infty+1} - \theta_{\infty+1/2}] \\
&= \mathbb{E}[\theta_\infty - \theta^*] + \alpha(\mathbb{E}[g(\theta_\infty, x_\infty)] + \mathbb{E}[\xi_{\infty+1}(\theta_\infty)]) + \mathbb{E}[\theta_{\infty+1} - \theta_{\infty+1/2}],
\end{aligned}
$$

which implies that

$$
-\frac{1}{\alpha} \mathbb{E}[\theta_{\infty+1} - \theta_{\infty+1/2}] = \mathbb{E}[g'(\theta^*, x_\infty)(\theta_\infty - \theta^*)] + \frac{1}{2} \mathbb{E}[g''(\theta^*, x_\infty)(\theta_\infty - \theta^*)^{\otimes 2}] + \mathcal{O}((\alpha\tau)^{3/2}).
\tag{I.25}
$$

Therefore, we turn our focus to analyzing $\mathbb{E}[\theta_{\infty+1} - \theta_{\infty+1/2}]$.

$$
\begin{aligned}
& \mathbb{E}[\theta_{\infty+1} - \theta_{\infty+1/2}] \\
& = \mathbb{E}[(\theta_{\infty+1} - \theta_{\infty+1/2}) \mathbb{1}\{\|\theta_{t+1/2} - \theta^*\| < \beta\}] + \mathbb{E}[(\theta_{\infty+1} - \theta_{\infty+1/2}) \mathbb{1}\{\|\theta_{t+1/2} - \theta^*\| \geq \beta\}] \\
& = \mathbb{E}[(\theta_{\infty+1} - \theta_{\infty+1/2}) \mathbb{1}\{\|\theta_{t+1/2} - \theta^*\| \geq \beta\}],
\end{aligned}
$$

where we note that when $\|\theta_{t+1/2} - \theta^*\| \leq \beta$ implies that $\|\theta_{t+1/2}\| \leq \beta + \|\theta^*\| \leq 2\beta$ and hence $\theta_{\infty+1} = \theta_{t+1/2}$ in this case. To analyze the remaining term, we use Hölder's inequality and obtain

$$
\begin{aligned}
& \|\mathbb{E}[(\theta_{\infty+1} - \theta_{\infty+1/2}) \mathbb{1}\{\|\theta_{t+1/2} - \theta^*\| \geq \beta\}]\| \\
& \leq \mathbb{E}^{1/p}[\|(\theta_{\infty+1} - \theta^*) - (\theta_{\infty+1/2} - \theta^*)\|^p] \mathbb{E}^{1/q}[\mathbb{1}\{\|\theta_{t+1/2} - \theta^*\| \geq \beta\}]
\end{aligned}
$$

$$\leq 2\mathbb{E}^{1/p}[\|\theta_{\infty+1/2} - \theta^*\|^p]\big(\mathbb{P}(\|\theta_{t+1/2} - \theta^*\| \geq \beta)\big)^{1/q}.$$

Setting $p = 6$ and $q = 6/5$, and making use of the property that $\mathbb{E}[\|\theta_{\infty+1/2} - \theta^*\|^6] = \mathcal{O}((\alpha\tau)^3)$ from Proposition 4.2, we have

$$\mathbb{E}^{1/6}[\|\theta_{\infty+1/2} - \theta^*\|^6]\big(\mathbb{P}(\|\theta_{t+1/2} - \theta^*\| \geq R)\big)^{5/6} \lesssim (\alpha\tau)^{3/6}\Big(\mathbb{E}[\|\theta_{t+1/2} - \theta^*\|^6]/R^6\Big)^{5/6}$$
$$\lesssim (\alpha\tau)^3.$$

Hence, we can conclude that

$$\mathbb{E}[\theta_{\infty+1} - \theta_{\infty+1/2}] = \mathcal{O}((\alpha\tau)^3).$$

Substituting this order information back into (I.25), we can see that $\mathbb{E}[\theta_{\infty+1} - \theta_{\infty+1/2}]/\alpha = \mathcal{O}(\alpha^2\tau^3)$. Recall that $\tau = \mathcal{O}(\log(1/\alpha))$, and hence we can assimilate this residual order from projection into the existing $\mathcal{O}((\alpha\tau)^{3/2})$ residual term.

For the remaining terms, we follow the existing analysis in Section I.2.1 and again obtain

$$\mathbb{E}[\theta_\infty - \theta^*] = -(\bar{g}^{(1)}(\theta^*))^{-1}\Big(\mathbb{E}[g'(\theta^*, x_\infty)\delta_1(x_\infty)]$$
$$+ \frac{1}{2}\Big(\mathbb{E}[g''(\theta^*, x_\infty)\delta_2(x_\infty)] + \bar{g}^{(2)}(\theta^*)\mathbb{E}[(\theta_\infty - \theta^*)^{\otimes 2}]\Big)\Big)$$
$$+ \mathcal{O}((\alpha\tau)^{3/2}).$$

Now, we proceed to analyze $\mathbb{E}[(\theta_\infty - \theta^*)^{\otimes 2}]$ and examine the impact of projection. Consider test function $h_2(x, \theta) = (\theta - \theta^*)^{\otimes 2}$ and follow a similar strategy as the first moment analysis, we have

$$\mathbb{E}[(\theta_{\infty+1} - \theta^*)^{\otimes 2}] = \mathbb{E}[(\theta_{\infty+1/2} - \theta^*)^{\otimes 2}] + \mathbb{E}[(\theta_{\infty+1} - \theta^*)^{\otimes 2} - (\theta_{\infty+1/2} - \theta^*)^{\otimes 2}]$$
$$= \mathbb{E}[(\theta_\infty - \theta^*)^{\otimes 2}] + \alpha^2(\mathbb{E}[(g(\theta_\infty, x_\infty))^{\otimes 2}] + \mathbb{E}[(\xi_{\infty+1}(\theta_\infty))^{\otimes 2}])$$
$$+ \alpha(\mathbb{E}[g(\theta_\infty, x_\infty) \otimes (\theta_\infty - \theta^*)] + \mathbb{E}[(\theta_\infty - \theta^*) \otimes g(\theta_\infty, x_\infty)])$$
$$+ \mathbb{E}[(\theta_{\infty+1} - \theta^*)^{\otimes 2} - (\theta_{\infty+1/2} - \theta^*)^{\otimes 2}].$$

Hence, by reorganizing the terms, we have

$$-\frac{1}{\alpha}\mathbb{E}[(\theta_{\infty+1} - \theta^*)^{\otimes 2} - (\theta_{\infty+1/2} - \theta^*)^{\otimes 2}]$$
$$= \alpha(\mathbb{E}[(g(\theta_\infty, x_\infty))^{\otimes 2}] + \mathbb{E}[(\xi_{\infty+1}(\theta_\infty))^{\otimes 2}])$$
$$+ (\mathbb{E}[g(\theta_\infty, x_\infty) \otimes (\theta_\infty - \theta^*)] + \mathbb{E}[(\theta_\infty - \theta^*) \otimes g(\theta_\infty, x_\infty)]).$$

Therefore, we turn our focus to analyzing $\mathbb{E}[(\theta_{\infty+1} - \theta^*)^{\otimes 2} - (\theta_{\infty+1/2} - \theta^*)^{\otimes 2}]$. Similar as the first moment analysis, we have

$$\mathbb{E}[(\theta_{\infty+1} - \theta^*)^{\otimes 2} - (\theta_{\infty+1/2} - \theta^*)^{\otimes 2}]$$
$$= \mathbb{E}[((\theta_{\infty+1} - \theta^*)^{\otimes 2} - (\theta_{\infty+1/2} - \theta^*)^{\otimes 2})\mathbb{1}\{\|\theta_{t+1/2} - \theta^*\| \geq \beta\}].$$

To analyze the term on the right-hand side, we again make use of Hölder's inequality and obtain

$$\|\mathbb{E}[((\theta_{\infty+1} - \theta^*)^{\otimes 2} - (\theta_{\infty+1/2} - \theta^*)^{\otimes 2})\mathbb{1}\{\|\theta_{t+1/2} - \theta^*\| \geq \beta\}]\|$$
$$\leq \mathbb{E}^{1/p}[\|(\theta_{\infty+1} - \theta^*)^{\otimes 2} - (\theta_{\infty+1/2} - \theta^*)^{\otimes 2}\|^p]\mathbb{E}^{1/q}[\mathbb{1}\{\|\theta_{t+1/2} - \theta^*\| \geq \beta\}]$$
$$\leq 2\mathbb{E}^{1/p}[\|\theta_{\infty+1/2} - \theta^*\|^{2p}]\big(\mathbb{P}(\|\theta_{t+1/2} - \theta^*\| \geq \beta)\big)^{1/q}.$$

Setting $p = 3$ and $q = 3/2$, and making use of the property that $\mathbb{E}[\|\theta_{\infty+1/2} - \theta^*\|^6] = \mathcal{O}((\alpha\tau)^3)$ from Proposition 4.2, we have

$$\mathbb{E}^{1/3}[\|\theta_{\infty+1/2} - \theta^*\|^6]\big(\mathbb{P}(\|\theta_{t+1/2} - \theta^*\| \geq R)\big)^{2/3} \lesssim (\alpha\tau)\Big(\mathbb{E}[\|\theta_{t+1/2} - \theta^*\|^6]/R^6\Big)^{2/3}$$
$$\lesssim (\alpha\tau)^3.$$

Hence, we can conclude that

$$\mathbb{E}[(\theta_{\infty+1} - \theta^*)^{\otimes 2} - (\theta_{\infty+1/2} - \theta^*)^{\otimes 2}] = \mathcal{O}((\alpha\tau)^3).$$

From the analyses above, we can also conclude that

$$\|\mathbb{E}[\theta_{\infty+1} - \theta_{\infty+1/2}|x_\infty]\|_{L^2(\pi)} = \mathcal{O}((\alpha\tau)^{3/2})$$
$$\|\mathbb{E}[(\theta_{\infty+1} - \theta^*)^{\otimes 2} - (\theta_{\infty+1/2} - \theta^*)^{\otimes 2}|x_\infty]\|_{L^2(\pi)} = \mathcal{O}((\alpha\tau)^{3/2}).$$

Therefore, combining the analyses above, we see that the projection only introduces error terms of order $\mathcal{O}(\alpha^2\tau^3)$. Hence, combining the analysis from Section I.2.1, we can conclude the same desired order that

$$\mathbb{E}[\theta_\infty^{(\alpha)} - \theta^*] = \alpha b + \mathcal{O}((\alpha\tau)^{3/2}).$$

# J  Additional Insights on TA and RR

In this section, we present more detailed results that characterize the first and second moment of Polayk-Ruppert (PR) tail-averaged iterates and Richardson-Romberg (RR) extrapolated iterates.

The following corollary provides non-asymptotic characterization for the first two moments of PR tail-averaged iterates $\bar\theta_{k_0,k}$.

**Corollary J.1** (Tail Averaging). *Under the setting of Theorem 4.6, the tail-averaged iterates satisfy the following bounds for all $k > k_0 + 2\tau$ and $k_0 \geq \tau + \frac{1}{\alpha\mu}\log\left(\frac{1}{\alpha\tau_\alpha}\right)$:*

$$\mathbb{E}[\bar\theta_{k_0,k}^{(\alpha)} - \theta^*] = \alpha b + \mathcal{O}\left((\alpha\tau_\alpha)^{3/2}\right) + \mathcal{O}\left(\frac{(1-\alpha\mu)^{k_0/2}}{\alpha(k - k_0)}\right) \quad and \tag{J.1}$$

$$\mathbb{E}\left[(\bar\theta_{k_0,k}^{(\alpha)} - \theta^*)(\bar\theta_{k_0,k}^{(\alpha)} - \theta^*)^\top\right] = \alpha^2 bb^T + \mathcal{O}(\alpha \cdot (\alpha\tau_\alpha)^{3/2}) + \mathcal{O}\left(\frac{\tau_\alpha}{k - k_0} + \frac{(1-\alpha\mu)^{k_0/2}}{\alpha(k - k_0)^2}\right). \tag{J.2}$$

With this result, taking the trace on both sides of (J.2) recovers Corollary 4.7.

*Proof.* First, we have

$$\mathbb{E}[\bar\theta_{k_0,k}^{(\alpha)} - \theta^*] = (\mathbb{E}[\theta_\infty] - \theta^*) + \frac{1}{k - k_0}\sum_{t=k_0}^{k-1}\mathbb{E}[\theta_t - \theta_\infty].$$

By Corollary 4.4, we obtain

$$\|\mathbb{E}[\theta_t] - \mathbb{E}[\theta_\infty]\| \leq (1-\alpha\mu)^{\frac{t}{2}} \cdot s'(\theta_0, L, \mu).$$

Hence, it follows that

$$\left\|\sum_{t=k_0}^{k-1}\mathbb{E}[\theta_t - \theta_\infty]\right\| \leq \sum_{t=k_0}^{k-1}\|\mathbb{E}[\theta_t] - \mathbb{E}[\theta_\infty]\|$$
$$\leq s'(\theta_0, L, \mu) \cdot (1-\alpha\mu)^{\frac{k_0}{2}}\frac{1}{1 - \sqrt{(1-\alpha\mu)}}$$
$$\leq s'(\theta_0, L, \mu) \cdot (1-\alpha\mu)^{\frac{k_0}{2}}\frac{2}{\alpha\mu}.$$

Together with Theorem 4.6, we have

$$\mathbb{E}\left[\bar\theta_{k_0,k}^{(\alpha)}\right] - \theta^* = \alpha b + \mathcal{O}\left((\alpha\tau_\alpha)^{3/2}\right) + \mathcal{O}\left(\frac{(1-\alpha\mu)^{k_0/2}}{\alpha(k - k_0)}\right),$$

thereby finishing the proof of the first moment.

To bound the second moment of the tail-averaged iterate, we follow the proof technique in [33, Section A.6.2] . We notice that

$$
\mathbb{E}\left[\left(\bar{\theta}_{k_0,k}^{(\alpha)} - \theta^*\right)\left(\bar{\theta}_{k_0,k}^{(\alpha)} - \theta^*\right)^\top\right]
$$

$$
= \mathbb{E}\left[\left(\bar{\theta}_{k_0,k}^{(\alpha)} - \mathbb{E}\left[\theta_\infty\right] + \mathbb{E}\left[\theta_\infty\right] - \theta^*\right)\left(\bar{\theta}_{k_0,k}^{(\alpha)} - \mathbb{E}\left[\theta_\infty\right] + \mathbb{E}\left[\theta_\infty\right] - \theta^*\right)^\top\right]
$$

$$
= \underbrace{\mathbb{E}\left[\left(\bar{\theta}_{k_0,k}^{(\alpha)} - \mathbb{E}\left[\theta_\infty\right]\right)\left(\bar{\theta}_{k_0,k}^{(\alpha)} - \mathbb{E}\left[\theta_\infty\right]\right)^\top\right]}_{T_1} + \underbrace{\mathbb{E}\left[\left(\bar{\theta}_{k_0,k}^{(\alpha)} - \mathbb{E}\left[\theta_\infty\right]\right)\left(\mathbb{E}\left[\theta_\infty\right] - \theta^*\right)^\top\right]}_{T_2}
$$

$$
+ \underbrace{\mathbb{E}\left[\left(\mathbb{E}\left[\theta_\infty\right] - \theta^*\right)\left(\bar{\theta}_{k_0,k}^{(\alpha)} - \mathbb{E}\left[\theta_\infty\right]\right)^\top\right]}_{T_3} + \underbrace{\mathbb{E}\left[\left(\mathbb{E}\left[\theta_\infty\right] - \theta^*\right)\left(\mathbb{E}\left[\theta_\infty\right] - \theta^*\right)^\top\right]}_{T_4} .
$$

For $T_2$, we have

$$
T_2 = \frac{1}{k - k_0}\left(\sum_{t=k_0}^{k-1} \mathbb{E}\left[\theta_t - \theta_\infty\right]\right)\left(\mathbb{E}[\theta_\infty] - \theta^*\right)^\top
$$

$$
= \mathcal{O}\left(\frac{(1 - \alpha\mu)^{k_0/2}}{\alpha(k - k_0)}\right) \cdot (\alpha b + \mathcal{O}((\alpha\tau_\alpha)^{3/2})) = \mathcal{O}\left(\frac{(1 - \alpha\mu)^{k_0/2}}{(k - k_0)}\right) .
$$

The term $T_3$ is similar to $T_2$ and obeys the same bound.

For $T_4$, we have

$$
T_4 = (\alpha b + \mathcal{O}((\alpha\tau_\alpha)^{3/2}))(\alpha b + \mathcal{O}((\alpha\tau_\alpha)^{3/2}))^T = \alpha^2 bb^T + \mathcal{O}(\alpha \cdot (\alpha\tau_\alpha)^{3/2}).
$$

For $T_1$, we have

$$
T_1 = \frac{1}{(k - k_0)^2}\mathbb{E}\left[\left(\sum_{t=k_0}^{k-1}(\theta_t - \mathbb{E}[\theta_\infty])\right)\left(\sum_{t=k_0}^{k-1}(\theta_t - \mathbb{E}[\theta_\infty])\right)^\top\right]
$$

$$
= \frac{1}{(k - k_0)^2}\sum_{t=k_0}^{k-1}\mathbb{E}\left[(\theta_t - \mathbb{E}[\theta_\infty])(\theta_t - \mathbb{E}[\theta_\infty])^\top\right] \tag{J.3}
$$

$$
+ \frac{1}{(k - k_0)^2}\sum_{t=k_0}^{k-1}\sum_{l=t+1}^{k-1}\mathbb{E}\left[(\theta_t - \mathbb{E}[\theta_\infty])(\theta_l - \mathbb{E}[\theta_\infty])^\top\right] \tag{J.4}
$$

$$
+ \frac{1}{(k - k_0)^2}\sum_{t=k_0}^{k-1}\sum_{l=t+1}^{k-1}\mathbb{E}\left[(\theta_l - \mathbb{E}[\theta_\infty])(\theta_t - \mathbb{E}[\theta_\infty])^\top\right] . \tag{J.5}
$$

By Corollary 4.4 and Proposition 4.2 we have

$$
\mathbb{E}\left[(\theta_t - \mathbb{E}[\theta_\infty])(\theta_t - \mathbb{E}[\theta_\infty])^\top\right]
$$

$$
= \left(\mathbb{E}\left[\theta_t \theta_t^\top\right] - \mathbb{E}\left[\theta_\infty \theta_\infty^\top\right]\right) + \left(\mathbb{E}\left[\theta_\infty \theta_\infty^\top\right] - \mathbb{E}[\theta_\infty]\mathbb{E}\left[\theta_\infty^\top\right]\right)
$$

$$
- \left(\mathbb{E}\left[\theta_t\right]\mathbb{E}\left[\theta_\infty^\top\right] + \mathbb{E}[\theta_\infty]\mathbb{E}\left[\theta_t^\top\right] - 2\mathbb{E}[\theta_\infty]\mathbb{E}\left[\theta_\infty^\top\right]\right)
$$

$$
= \left(\mathbb{E}\left[\theta_t \theta_t^\top\right] - \mathbb{E}\left[\theta_\infty \theta_\infty^\top\right]\right) + \mathrm{Var}\left(\theta_\infty\right) - \mathbb{E}[\theta_t - \theta_\infty]\mathbb{E}\left[\theta_\infty^\top\right] - \mathbb{E}[\theta_\infty]\mathbb{E}\left[(\theta_t - \theta_\infty)^\top\right]
$$

$$
= \mathcal{O}\left((1 - \alpha\mu)^{\frac{t}{2}} + \alpha\tau_\alpha\right),
$$

where we bound $\mathrm{Var}\left(\theta_\infty\right)$ with $\mathcal{O}(\alpha\tau_\alpha)$ by Proposition 4.2 and Fatou's lemma.

Then, for (J.3), we have

$$(\text{J.3}) = \frac{1}{(k-k_0)^2} \sum_{t=k_0}^{k-1} \mathcal{O}\left( (1-\alpha\mu)^{\frac{t}{2}} + \alpha\tau_\alpha \right)$$

$$= \mathcal{O}\left( \frac{1}{(k-k_0)^2} \sum_{t=k_0}^{\infty} (1-\alpha\mu)^{\frac{t}{2}} \right) + \mathcal{O}\left( \frac{\alpha\tau_\alpha}{k-k_0} \right)$$

$$= \mathcal{O}\left( \frac{(1-\alpha\mu)^{k_0/2}}{\alpha(k-k_0)^2} + \frac{\alpha\tau_\alpha}{k-k_0} \right).$$

We restate the following claim, whose proof closely resembles Claim 4 in [33].

**Claim 2.** *For $t \geq \tau + \frac{1}{\alpha\mu} \log\left(\frac{1}{\alpha\tau_\alpha}\right)$ and $l \geq t + 2\tau_\alpha$, we have*

$$\left\| \mathbb{E}\left[ (\theta_t - \mathbb{E}[\theta^{(\alpha)}])(\theta_l - \mathbb{E}[\theta^{(\alpha)}])^\top \right] \right\| = \mathcal{O}\left( (\alpha\tau_\alpha) \cdot (1-\alpha\mu)^{\frac{(l-t)}{2}} \right).$$

Then, by [33, Claim 4], we have term $(\text{J.4}) = \mathcal{O}\left(\frac{\tau_\alpha}{k-k_0}\right)$. Similarly, we have term $(\text{J.5}) = \mathcal{O}\left(\frac{\tau_\alpha}{k-k_0}\right)$. Therefore, we have

$$T_1 = \mathcal{O}\left( \frac{(1-\alpha\mu)^{k_0/2}}{\alpha(k-k_0)^2} \right) + \frac{\tau_\alpha}{k-k_0}. \tag{J.6}$$

By adding $T_1$–$T_4$ together, we obtain

$$\mathbb{E}\left[ \left( \bar{\theta}_{k_0,k}^{(\alpha)} - \theta^* \right) \left( \bar{\theta}_{k_0,k}^{(\alpha)} - \theta^* \right)^\top \right] = \alpha^2 bb^T + \mathcal{O}(\alpha \cdot (\alpha\tau_\alpha)^{3/2}) + \mathcal{O}\left( \frac{(1-\alpha\mu)^{k_0/2}}{(k-k_0)} \right)$$

$$+ \mathcal{O}\left( \frac{(1-\alpha\mu)^{k_0/2}}{\alpha(k-k_0)^2} + \frac{\tau_\alpha}{k-k_0} \right)$$

$$= \alpha^2 bb^T + \mathcal{O}(\alpha \cdot (\alpha\tau_\alpha)^{3/2}) + \mathcal{O}\left( \frac{\tau_\alpha}{k-k_0} + \frac{(1-\alpha\mu)^{k_0/2}}{\alpha(k-k_0)^2} \right).$$

$\square$

Next, we present the following corollary formalizes the non-asymptotic characterization for the first two moments of the RR-extrapolated iterate $\widetilde{\theta}_{k_0,k}^{(\alpha)}$.

**Corollary J.2** (Richardson-Romberg Extrapolation)**.** *Under the setting of Theorem 4.6, the RR extrapolated iterates with stepsizes $\alpha$ and $2\alpha$ satisfy the following bounds for all $k > k_0 + 2\tau_\alpha$ and $k_0 \geq \tau_\alpha + \frac{1}{\alpha\mu} \log\left(\frac{1}{\alpha\tau_\alpha}\right)$:*

$$\mathbb{E}[\tilde{\theta}_{k_0,k}^{(\alpha)} - \theta^*] = \mathcal{O}\left((\alpha\tau_\alpha)^{3/2}\right) + \mathcal{O}\left( \frac{(1-\alpha\mu)^{k_0/2}}{\alpha(k-k_0)} \right), \quad and \tag{J.7}$$

$$\mathbb{E}\left[ (\tilde{\theta}_{k_0,k}^{(\alpha)} - \theta^*)(\tilde{\theta}_{k_0,k} - \theta^*)^\top \right] = \mathcal{O}\left((\alpha\tau_\alpha)^3\right) + \mathcal{O}\left( \frac{\tau_\alpha}{k-k_0} + \frac{(1-\alpha\mu)^{k_0/2}}{\alpha(k-k_0)^2} \right). \tag{J.8}$$

*Proof.* By equation (J.1), we obtain

$$\mathbb{E}\left[ \tilde{\theta}_{k_0,k}^{(\alpha)} \right] - \theta^* = \mathbb{E}\left[ 2\bar{\theta}_{k_0,k}^{(\alpha)} - \bar{\theta}_{k_0,k}^{(2\alpha)} \right] - \theta^* = 2\mathbb{E}\left[ \bar{\theta}_{k_0,k}^{(\alpha)} - \theta^* \right] - \mathbb{E}\left[ \bar{\theta}_{k_0,k}^{(2\alpha)} - \theta^* \right]$$

$$= 2\left( \alpha b + \mathcal{O}\left((\alpha\tau_\alpha)^{3/2}\right) + \mathcal{O}\left( \frac{(1-\alpha\mu)^{k_0/2}}{\alpha(k-k_0)} \right) \right)$$

$$- \left( 2\alpha b + \mathcal{O}\left((2\alpha\tau_{2\alpha})^{3/2}\right) + \mathcal{O}\left( \frac{(1-2\alpha\mu)^{k_0/2}}{\alpha(k-k_0)} \right) \right)$$

$$= \mathcal{O}\left((\alpha\tau_\alpha)^{3/2}\right) + \mathcal{O}\left( \frac{(1-\alpha\mu)^{k_0/2}}{\alpha(k-k_0)} \right).$$

Let $u_1 := \bar{\theta}^{(\alpha)}_{k_0,k} - \mathbb{E}\left[\theta^{(\alpha)}_\infty\right]$, $u_2 := \bar{\theta}^{(2\alpha)}_{k_0,k} - \mathbb{E}\left[\theta^{(2\alpha)}_\infty\right]$ and $v := 2\mathbb{E}\left[\theta^{(\alpha)}_\infty\right] - \mathbb{E}\left[\theta^{(2\alpha)}_\infty\right] - \theta^*$.

With these notations, $\tilde{\theta}_{k_0,k} - \theta^* = 2u_1 - u_2 + v$. We then have the following bound

$$\left\|\mathbb{E}\left[\left(\tilde{\theta}^{(\alpha)}_{k_0,k} - \theta^*\right)\left(\tilde{\theta}^{(\alpha)}_{k_0,k} - \theta^*\right)^\top\right]\right\| \leq \mathbb{E}\left[\|2u_1 - u_2 + v\|^2\right]$$
$$\leq 3\mathbb{E}\|2u_1\|^2 + 3\mathbb{E}\|u_2\|^2 + 3\|v\|^2.$$

By equation (J.6), we have

$$\mathbb{E}\|u_1\|^2 = \mathrm{Tr}\left(\mathbb{E}\left[u_1 u_1^\top\right]\right) = \mathcal{O}\left(\frac{(1-\alpha\mu)^{k_0/2}}{\alpha\left(k - k_0\right)^2} + \frac{\tau_\alpha}{k - k_0}\right).$$

Similarly, we have

$$\mathbb{E}\|u_2\|_2^2 = \mathcal{O}\left(\frac{(1-2\alpha\mu)^{k_0/2}}{\alpha\left(k - k_0\right)^2} + \frac{\tau_{2\alpha}}{k - k_0}\right).$$

By Theorem 4.6, we have $\|v\|_2^2 = \mathcal{O}\left((\alpha\tau_\alpha)^3\right)$.

Combining these bounds, we have

$$\mathbb{E}\left[\left(\tilde{\theta}_{k_0,k} - \theta^*\right)\left(\tilde{\theta}_{k_0,k} - \theta^*\right)^\top\right] = \mathcal{O}\left((\alpha\tau_\alpha)^3\right) + \mathcal{O}\left(\frac{\tau_\alpha}{k - k_0} + \frac{(1-\alpha\mu)^{k_0/2}}{\alpha\left(k - k_0\right)^2}\right).$$

$\square$

