# OpenReview forum: "The Collusion of Memory and Nonlinearity in Stochastic Approximation With Constant Stepsize"
_NeurIPS.cc/2024/Conference — NeurIPS 2024 spotlight_

### Official Review · Reviewer_9yxV · 2024-07-10

**Soundness:** 3
**Presentation:** 3
**Contribution:** 3
**Rating:** 7
**Confidence:** 4

**Summary:**

This paper studies constant step-size stochastic approximation algorithms with Markovian noise. It is known that in this case, the error of a stochastic does not vanish asymptotically, even when consider averaging. This is \theta_n has a bias.  Previous work have show how to study the bias of such algorithm for linear stochastic approximation. In this paper, the authors tackle the more challenging case of non-linear stochastic approximation. Most of the paper is devoted to proving that the bias is of order O(\alpha). Some discussion on the applicability of results is presented.

**Strengths:**

The paper proposes a characterization of the bias under the challenging setting of Markovian noise plus non-linear drifts.

The analysis is asymptotically tight as the step size \alpha converges to 0, i.e., the expression for the bias is not a bound but an equality plus smaller order terms.

The implications of the results are discussed.

**Weaknesses:**

Most of the assumptions needed to obtain the results are relatively mild but two conditions are quite strong:
1. strong monotonicity (A3)
2. Smoothness (A2)
I think that the paper should discuss in more details these assumption and highlight that they are really limiting the applicability of the result.

The assumption developed in part 4.2 to avoid assuming that the iterates are bounded seems quite strong, not verified in many practical cases (for instance any instance of stochastic gradient descent where the noise has a finite support would not satisfy this assumption), and very artificial (is the only purpose of this assumption to

The paper is extremely long (54 pages including proofs and references). To me, this says that there is either too much content or that the results are too diluted. As a result, the paper is very technical and hard to read. There should be more effort to make the paper more readable. Some suggestions:
- consider a slightly less general setting
- avoid considering sub-cases (like section 4.2) that are a bit orthogoal to the paper

The practical applications of the results are unclear. Some potential applications are presented in Section 4.5 / 4.6 but there are no experiments or simulations to confirm that this actually work (these sections are interesting, though).

There is a lot of papers on the subject. This can be seen as a good sign (this is an active area of research), but at the same time, it is hard for me to really assess the novelty of the results. Note that there are some (recent) papers that seem related to these work and that might be cited. This last point is not a weakness since some of them are extremely recent (available online after the submission deadline):
- Computing the Bias of Constant-step Stochastic Approximation with Markovian Noise. S Allmeier, N Gast (2024)
- Bias in Stochastic Approximation Cannot Be Eliminated With Averaging, Caio Kalil Lauand, Sean P. Meyn (2022)
- Revisiting Step-Size Assumptions in Stochastic Approximation. Caio Kalil Lauand, Sean Meyn (2024)
A comparison with these papers might be useful (even of this is not mandatory for the two 2024 papers).

**Questions:**

There are some questions / comments in the limit that would deserve some comments from the authors.

The Markovian noise (x_k) is assumed exogeneous (it does not depend on \theta). For some application (like Q-learning with a navigating policy derived from \theta), this would not be satisfied. Could this assumption be lifted?

The next order term is a O(\alpha^{3/2}): is it the sharpest bound or could O(\alpha^2) be obtained?

**Limitations:**

The limitation of the assumptions (see "weaknesses") should be better discussed.

---

> ### Author Rebuttal · Authors · 2024-08-07
>
> We thank the reviewer for recognizing the strengths and contribution of our paper, and for the constructive comments. Below we provide our responses to the comments. In the following, we use [1] [2] etc. to refer to papers cited in our submission, and [a] [b] etc.
> for new references, with bibliographic information given at the end of this reply.
>
> **Comment: Discussion on the limitation of Assumption 2 (smoothness) and Assumption 3 (strong convexity/monotonicity).**
>
> We thank the reviewer for raising this point. We refer the reviewer to the **global rebuttal** for additional discussion on the smoothness and strong convexity/monotonicity assumptions.
>
> **Comment: Discussion on Assumption 5 on noise minorization.**
>
> We address this comment in the **global rebuttal**, as it is an important point.
>
> **Comment: paper structure and presentation**
>
> We thank the reviewer for their feedback on the paper's structure. We will incorporate the suggestions in the revision to make our paper more succinct.
>
> **Comment: practical application and numerical experiments.**
>
> We thank the reviewer for the constructive comment. We briefly remark on the practical application of our results. As discussed in Section 4.6, many GLMs satisfy the conditions of our results. Our theory suggests that by running SGD with two stepsizes in parallel and tracking the PR-averaged iterates for each, we can use RR-extrapolation to obtain a reduced-bias estimate. Additionally, we can construct confidence intervals under our CLT guarantee.
>
> We also include numerical experiments in the **global rebuttal** to demonstrate our results. Specifically, we performed a set of experiments with $L_2$-regularized logistic regression with Markovian data to support our theoretical findings: the presence of bias and its reduction through Richardson-Romberg (RR) extrapolation, as well as the central limit theorem (CLT) of Polyak-Ruppert (PR) averaged iterates. Employing $L_2$-regularized logistic regression in these experiments also showcases the practical applicability of our results within the scope of generalized linear models (GLMs). We will add the numerical experiments in the revised paper.
>
>
> **Comment: Additional reference.**
>
> We thank the reviewer for pointing out the additional related works. All three works examine general (non-linear) SA under Markovian noise with a constant stepsize. Both [b] and our work prove weak convergence, but with different techniques. [b] provides an **upper bound** for the asymptotic bias, while we offer an equality and closed-form solution for the leading-order bias. [a] presents an upper bound for the PR-averaged iterates and, similar to our results, demonstrates the effectiveness of RR-extrapolation in reducing bias. Besides constant stepsize, [c] also explores diminishing stepsizes and examines the impact of the stepsize decay rate on the asymptotic statistics of PR-averaged SA. We will discuss these related works in our revised literature review.
>
> **Comment: Assumption on exogenous Markovian noise.**
>
> We acknowledge that our current model of Markovian noise does not account for dependence on $\theta$. In fact, the very recent work (pointed out by the reviewer) [a] considers the Markovian model that incorporates such dependence. We are confident that our work can be extended to adopt a similar modeling approach for the Markovian noise and achieve comparable results.
>
> **Comment: Tighter higher order in the bias.**
>
> We agree with the reviewer's comment that $O(\alpha^{3/2})$ might not be the tightest next order in the bias characterization, and we believe that $O(\alpha^2)$ can be obtained. Improving the next highest order to $O(\alpha^2)$ requires a more refined characterization of the asymptotic second order $E[(\theta_\infty-\theta^*)^{\otimes2}]$ by following a similar strategy as our current approach. Thus, we leave this refinement out of the scope of the current paper. We conjecture that, with appropriate assumptions on smoothness and noise moment, we can prove refined characterizations of higher orders of $E[(\theta_\infty-\theta^*)^{\otimes p}]$, and subsequently we obtain $E[\theta_\infty]=\theta^*+\sum_{n=1}^m\alpha^nb_n+O(\alpha^{n+1})$.
>
>
>
> References:
>
> [a] S. Allmeier and N. Gast. Computing the Bias of Constant-step Stochastic Approximation with Markovian Noise. 2024.
>
> [b] C. K. Lauand and S. Meyn. Bias in Stochastic Approximation Cannot Be Eliminated With Averaging. 2022.
>
> [c] C. K. Lauand and S. Meyn. Revisiting Step-Size Assumptions in Stochastic Approximation. 2024.

---

> > ### Comment · Reviewer_9yxV · 2024-08-11
> >
> > Thank you for this detailed answer. I have no further questions.

---

### Official Review · Reviewer_gRG8 · 2024-07-12

**Soundness:** 2
**Presentation:** 2
**Contribution:** 3
**Rating:** 7
**Confidence:** 4

**Summary:**

The present paper obtains a representation for the asymptotic bias of constant step-size nonlinear stochastic approximation with Markovian noise. In particular, this characterization makes the hindering effect of memory and nonlinearity explicit in terms of the algorithm's performance. Moreover, the authors establish ergodicity of the parameter-noise joint process (with and without projection of estimates), obtain finite-time bounds on L_p moments of the estimation error and establish a central limit theorem for the constant gain algorithm. Finally, a bias attenuation technique based upon the Richardson-Romberg extrapolation is proposed for the nonlinear algorithm.

**Strengths:**

The contributions and assumptions are clearly identified. To the best of the reviewer's knowledge, the results are novel and exciting: it is great to see the hindering effect caused by the interplay between memory and nonlinearity in SA.

This paper is well written, but could use some polishing.

I did not have enough time to review all proofs in detail, but the analysis seems correct.

**Weaknesses:**

One of the weaknesses of this paper is the fact that no numerical experiments are provided to illustrate any of the main results. Although a discussion on how the theory fits within Generalized Linear Models is given before the conclusions, I encourage the authors to include a simple toy example to illustrate some of their main results such as the CLT or the bias attenuation technique.

**Questions:**

-	Could the authors clarify if assuming strong monotonicity and uniform boundedness of g and its Lipschitz constant over x are indeed needed for the result in Thm. 4.6? I can see their importance for finite-time bounds, but if there are needed for the asymptotic bias bound, I believe that the authors should mention the employment of stronger assumptions when comparing their work with previous research on asymptotic results.

-	I am not sure I understand what the authors mean by ``fine-grained’’  when talking about the result in Thm. 4.6. Upon further inspection, it seems that equation (4.2) is an extension of the result in [40] for nonlinear recursions: it consists a representation for the dominant bias term plus an upper bound as in this previous work. Is the fine-grained part related to the upper bound for \alpha?

-	I might have missed this in the text, but could the authors clarify if the results in Sections 4.3 and 4.5 require projection?

**Limitations:**

The authors addressed the limitations in their work through a clear list of assumptions and discussions after presenting the main results.

---

> ### Author Rebuttal · Authors · 2024-08-07
>
> We thank the reviewer for recognizing our paper’s contribution and strengths, and for the constructive comments. We provide our detailed responses below. In the following, we use [1] [2] etc. to refer to papers cited in our submission, and [a] [b] etc. for new references, with bibliographic information given at the end of this reply.
>
> **Comment: numerical experiments.**
>
> We thank the reviewer for the suggestion. We include numerical experiments in the **global rebuttal** to demonstrate our results. In particular, we conducted two sets of experiments using $L_2$-regularized logistic regression with Markovian data to verify our theoretical results: the existence of bias and its attenuation via Richardson-Romberg (RR) extrapolation, and the central limit theorem (CLT) of Polyak-Ruppert (PR) averaged iterates. Using $L_2$-regularized logistic regression in the experiment also demonstrates the practicality of our results in the context of generalized linear models (GLMs). We will add the numerical experiments in the revised paper.
>
>
> **Comment: Clarification of Assumption 3 (boundedness and smoothness of $g$) and Assumption 4 (strong monotonicity) for Theorem 4.6.**
>
> We thank the reviewer for raising this point. For discussions on strong convexity and Lipschitz smoothness in proving weak convergence, we refer the reviewer to the global rebuttal. Here, we discuss the role of these two assumptions in bias characterization.
>
> The key technique in bias characterization is Taylor expansion around $\theta^\ast$. The strong monotonicity and Lipschitz smoothness together ensures that $E\|\theta_\infty-\theta^*\|^{2p}$ is of order $O((\alpha\tau)^p)$, which subsequently controls the order of the residual term in Taylor expansion. Moreover, the algebraic manipulation in bias characterization involves the inversion of $\bar{g}'(\theta^*)$ (in the context of SGD, this is equivalent to the Hessian of the objective function). Therefore, the strong monotonicity (convexity) ensures the validity of this inversion.
>
> We believe that we could potentially relax the strong monotonicity assumption to Hurwitz $\bar{g}'(\theta^\ast)$ to ensure the validity of such a matrix inversion. It is unclear what our approach would imply if we only have a positive semi-definite (not full-rank) $\bar{g}'(\theta^*)$. We conjecture that the joint process would still converge, but the bias may exhibit drastically different behaviors that we do not yet fully understand. This conjecture is based on [a], where the SA update is strongly convex and Lipschitz-smooth but non-differentiable at $\theta^*$. [a] shows that the bias admits a very different behavior, with the leading term scaling with $\sqrt{\alpha}$ instead of $\alpha$. Therefore, we currently do not have a clear conjecture for this case without the strong monotonicity assumption at $\theta^*$.
>
>
> **Comment: "Fine-grained" characterization of bias in Theorem 4.6.**
>
> We clarify how our bias characterization (4.2) -- (4.5) in Theorem 4.6 differs from the result (8) in [40]. We first remark that characterization in [40] is an \textbf{upper bound} ($\limsup$) on the averaged iterate's bias, not the asymptotic bias of the limiting random variable $E[\theta_\infty]-\theta^*$, since [40] does not prove weak convergence for general SA. Additionally, the result in [40] only shows that the leading term of bias is of $\alpha$-order. In contrast, we provide a closed-form expression in (4.2) -- (4.5) for the leading term of bias, which is computable if the underlying Markovian noise is known or can be approximated. Moreover, we show that the leading term can be decomposed into three components: Markovian noise $b_m$, non-linearity $b_n$, and the compound effect $b_c$, which cannot be implied from [40]. Therefore, our result is a more "fine-grained" characterization.
>
>
> **Comment: Clarification on projection for results in Section 4.3/4.5.**
>
> We thank the reviewer for raising this point. The results in Section 4.3 (non-asymptotic result) and 4.5 (CLT), as well as 4.4 (asymptotic bias characterization), are follow-up results to the weak convergence results in Sections 4.1 and 4.2 (weak convergence with or without projection). In obtaining these follow-up results, projection for uniform boundedness of iterates $\theta_t$ is not required. If we have weak convergence without projection, these results subsequently do not require projection. For discussions on removing the projection and minorization noise assumption, please refer to the **global rebuttal**.
>
> References:
>
> [a] Y. Zhang, D. L. Huo, Y. Chen, Q. Xie. Prelimit Coupling and Steady-State Convergence of Constant-stepsize Nonsmooth Contractive SA. 2024.

---

> > ### Comment · Reviewer_gRG8 · 2024-08-11
> >
> > I thank the authors for their clarifying and detailed responses. I also thank them for including an experiment that validates their theorems. Also, it is interesting to see the experiment with i.i.d. data performing worst than the experiment with Markovian data.
> >
> > Here a few more observations:
> >
> > **On a bias equation** I apologize If I was not clear in my first response. When I mentioned [40] I was referring to their equation (10) where there are no limsups and not (8).
> >
> > However, it is very clear to me that this does not affect the value of our work since they do not incorporate the effects of nonlinearity. Still, I think I would be beneficial to mention that a similar fine-grained expression was obtained for the more restrictive case of linear $g$ in the final version.
> >
> > **Strong Monotonicity** Maybe it would be beneficial to provide a bit more discussion regarding Assumption 3 in the final version like in the responses provided (e.g. when it is satisfied/ or if it could be lifted).

---

> > > ### Author Response · Authors · 2024-08-11
> > >
> > > Thank you for the clarification and response. We will include those discussions in our revised paper.

---

### Official Review · Reviewer_o1Ug · 2024-07-17

**Soundness:** 2
**Presentation:** 3
**Contribution:** 2
**Rating:** 5
**Confidence:** 3

**Summary:**

This paper considers a nonlinear stochastic approximation (SA) problem with Markov noise (MC). It is assumed that the MC is uniformly geometrically ergodic. Instead of the standard iterative procedure, a projection onto a bounded set is additionally introduced (the latter can be relaxed under the additional assumption of the existence of a positive density). Under these assumptions the authors manage to write a decomposition that characterizes the bias. At the same time, they manage to identify three factors influencing the bias: the factor of MC, the factor of nonlinearity of the procedure, and the factor of interaction between MC and nonlinearity. In addition, bounds for the Polyak-Ruppert averaging and the Richardson Romberg procedure are given.

**Strengths:**

- Decomposition that characterizes the bias with explicit dependence on MC, non-linearity and interactions between MC and non-linearity.

**Weaknesses:**

It would be good to obtain:
- high probability bounds instead of the MSE
- remove additional projection step and assumption on the density (it could be useful to consider convergence in the weighted W distance instead of V norm)
-  explicit dependence on the asymptotic variance of MC in the $O(\tau/(k - k_0))$.

**Questions:**

Could you please comment on the fact that there is no dependence between stepsize and number of iterates in the Polyak Ruppert avaraging?

**Limitations:**

-

---

> ### Author Rebuttal · Authors · 2024-08-07
>
> We thank the reviewer for recognizing our paper’s contribution and strengths, and for the constructive comments. We provide our detailed responses below. In what follows, we use [1] [2] etc. to refer to papers cited in our submission, and [a] [b] etc. for new references, with bibliographic information given at the end of this response.
>
> **Comment: High probability bounds instead of MSE.**
>
> We are grateful to the reviewer for this suggestion. We believe that we can derive high probability bounds by applying the Markov inequality to our Proposition 4.2 (convergence of $E\|\theta_k-\theta^*\|^{2p}$). Our bias characterization allows these bounds to be further tightened. If one would like to prove exponential tail bounds, stronger assumptions on the noise sequence, such as having exponential tails, are necessary as shown in [a] for iid data.
>
> **Comment: Discussion on projection and Assumption 5 on minorization.**
>
> We address this comment in the **global rebuttal**, as it is an important point.
>
> **Comment: Discussion on explicit dependence on variance of Markov chain.**
>
> We thank the reviewer for raising this point. We believe that it is straightforward to extend our results to characterize the dependence on the variance of the Markovian noise in the second moment of Polyak-Ruppert (PR) averaged iterates. We conjecture that by quantifying the variance introduced by the Markovian noise under an additional assumption that $\|g(\theta,x)-g(\theta,y)\|\leq \sigma_d$ for any $x,y\in\mathcal{X}$ and $\theta\in R^d$ (similar assumption in [46]), the explicit dependence of $\sigma$ would be $O(\sigma_d^2\tau_\alpha/(k-k_0))$.
>
> **Comment: Discussion on dependence between stepsize and number of iterates.**
>
> We thank the reviewer for bringing up this question. Our result allows for independent choices of stepsize $\alpha$ and the number of iterates $k$, provided that $\alpha$ is sufficiently small and $k$ sufficiently large. The first two terms of the second moment bound of the PR averaged iterates do not depend on the number of iterates, which verifies the presence of asymptotic bias with a leading term proportional to the stepsize $\alpha$. The remaining two terms depend on the number of iterates and will vanish as the number of iterates increases $k\to\infty$. The third term corresponds to the asymptotic variance and decays at the rate of $1/k$, and the fourth term corresponds to the optimization error. Our result also allows for optimizing the stepsize $\alpha$ if the number of iterates $k$ is known a priori. Suppose $k$ is fixed and $k_0=k/2$, the optimized $\alpha$ is obtained by balancing the first $\alpha^2$ term and the last $(1-\alpha\mu)^{k_0/2}/(\alpha (k-k_0)^2)$ term. The optimized order is $\alpha = O(k^{-2/3})$, which matches the order in [46], and, to the best of our knowledge, provides the tightest dependence for Markovian linear SA with constant stepsize. We would appreciate it if the reviewer could clarify any remaining questions to ensure our response fully addresses the question on the dependence between stepsize $\alpha$ and the number of iterates $k$ in the PR-averaging bound.
>
> References:
>
> [a] I. Merad and S. Gaïffas. Convergence and concentration properties of constant step-size SGD through Markov chains. 2023.

---

> > ### Comment · Reviewer_o1Ug · 2024-08-12
> >
> > I thank the authors for their response. I retain my current score.

---

### Official Review · Reviewer_3y4U · 2024-07-18

**Soundness:** 3
**Presentation:** 3
**Contribution:** 3
**Rating:** 7
**Confidence:** 2

**Summary:**

This paper investigates stochastic approximation (SA) with Markovian data and nonlinear updates under constant stepsize, and establishes the weak convergence of the joint process $(x_t, \theta_t)$. It also presents a precise characterization of the asymptotic bias of the SA iterates.

**Strengths:**

I find this paper well-written. The analysis appears to be correct (although details are not checked). Also, both the literature review and motivation are very clear. It seems to me that this paper has solved a challenging problem, caused by Markovian data and nonlinear updates.

**Weaknesses:**

I find the presentation of this paper a bit technical for people that are not very familiar with this area. In addition, perhaps some numerical experiments should be performed to better illustrate the theory.

**Questions:**

1. About Assump. 3: for many GLMs, we actually do not have strong convexity. I feel this assumption is a bit strong.
2. Page 4, line 147: is the notation superscript "\cross 2" defined anywhere? I assume this denotes the outer product of a vector.
3. Similar to Assump. 3, Assump. 4 should be justified further as well.

---

> ### Author Rebuttal · Authors · 2024-08-07
>
> We thank the reviewer for recognizing our paper’s contribution and strengths, and for the constructive comments. We provide our detailed responses below.
>
> **Comment: Numerical experiments.**
>
> We thank the reviewer for the suggestion. We conduct a set of experiments, running SGD on $L_2$-regularized logistic regression with Markovian data, to verify our theoretical results: the existence of bias, bias reduction via Richardson-Romberg (RR) extrapolation, and the central limit theorem (CLT) of Polyak-Ruppert (PR) averaged iterates. Using $L_2$-regularized logistic regression also demonstrates the practicality of our results in the context of generalized linear models (GLMs). For a more detailed discussion and the experiment results, please refer to the **global rebuttal** and the figures in the one-page PDF.
> We will add the numerical experiments in the revised paper.
>
> **Comment: Discussion on Assumption 3 on strong convexity/monotonicity.**
>
> We thank the reviewer for raising this point, and we refer the reviewer to the **global rebuttal** for clarification on the strong convexity/monotonicity assumption.
>
> **Comment: the $\otimes$ notation.**
>
> We use "$u\otimes v$'' to denote the tensor product of the two vectors $u$ and $v$, and "$u^{\otimes k}$" to denote the $k$-th tensor power of vector $u$. When $k=2$, it is simply the outer product of $M$. We thank the reviewer for pointing this out, and we will add this definition to the paper.
>
> **Comment: Discussion on Assumption 4 on the noise sequence.**
>
> We would like to further clarify Assumption 4, in which we assume the existence of the $2p$ moment of the noise sequence. We note that it is standard in the SA literature to assume the existence of the $2p$ moment of the noise sequence to control the $2p$ moment of the limiting random variable, as seen in [18,53,60] (cited in our submission). Moreover, we believe that this assumption is necessary; without a finite $2p$ moment for the noise sequence, the limiting random variable $\theta_\infty$ might not have a finite $2p$ moment. For example, consider the simple case  $\theta_{t+1}=\theta_t-\alpha(\theta_t-w_t)$, with $\theta_0=0$ and iid noise $w_t$. Note that we have $\theta_T=\alpha\sum_{t=0}^{T-1}(1-\alpha)^{T-t}w_t$. Therefore, it is easy to see that if the noise sequence does not have a finite $2p$ moment, $\theta_\infty$ would not have a finite $2p$ moment.

---

> > ### Comment · Reviewer_3y4U · 2024-08-07
> >
> > The authors have done a good job in responding to my previous comments. I find their responses clear and detailed. I have no further questions.

---

### Author Rebuttal · Authors · 2024-08-07

We thank all reviewers for their insightful feedback. In this global rebuttal, we discuss smoothness, strong monotonicity, the use of projection, and the minorization assumption of the noise sequence. We also supplement our theoretical results with a set of numerical experiments, with results in the one-page PDF.

In what follows, we use [1] [2] etc. to refer to papers cited in our submission, and [a] [b] etc. for new references with bibliographic information given at the end of this response.

**Differentiability and strong monotonicity:** In proving weak convergence, we assume that the stochastic approximation (SA) update operator $g$ is sufficiently smooth (three-times differentiable) and strongly monotone (for SGD this implies strong convexity of the objective function). This ensures controlled evolution of the iterates with Markovian/correlated data. The differentiability supports a Taylor series expansion of $g$ up to the second order with a bounded remainder, crucial for analyzing the $\spadesuit$ term in the convergence proof in Section 3 and for bias characterization where Taylor expansion around $\theta^\ast$ is the key technique. Some form of differentiability assumption is standard in SA literature, such as [18,40,a], particularly when one seeks a fine-grained characterization of the iterates' distributional property beyond MSE bounds. Such an assumption is satisfied by many GLMs, such as logistic regression and Poisson regression. When $g$ is not differentiable, SA behavior can differ significantly (even with iid data) [g], which is beyond our scope.

The strong monotonicity assumption is common in SA literature. Together with smoothness, it allows us to establish geometric distributional convergence. While some GLMs by themselves do not satisfy this condition, applying $L_2$-regularization (equivalently, weight decay) ensures strong convexity and improves statistical performance. It is a standard calculation that one can appropriately choose the regularization parameter to derive tight results for non-strongly-convex functions.
We believe that it is possible to relax the strong monotonicity assumption to weaker conditions, Hurwitz $\bar{g}'(\theta^\ast)$ [a,40], or even to non-convex problems satisfying structural properties like dissipativity or generalized Polyak-Lojasiewicz (PL) inequality [60].

**Projection and minorization:** Projection steps have a longstanding presence in SA literature for tractability in convergence theory, as seen in many analyses of SGD [b,c,d,e,5]. Although not an algorithmic proposal, this additional projection step does not incur much computational cost in practice, as it only involves rescaling the iterates, and the projection radius can be estimated a priori. Before our work, no studies had proven weak convergence for non-linear SA with Markovian data and constant stepsize, with or without the projection. Thus, our result is valuable, as it is the first to prove detailed weak convergence in this setting.

In Theorem 4.3, we provide an alternative proof of weak convergence using the Drift and Minorization technique, which does not require a projection. We acknowledge that the minorization assumption is not satisfied by some noise models, but argue that it is easily met by adding a small noise with a continuous distribution, a common practice for promoting exploration and privacy.

In addition, it is possible to extend our current results and prove weak convergence without the minorization noise or the projection. This can be achieved by employing the established Drift and Contraction technique [f]. As discussed in Section 3, when $\|\theta_{k-\tau}^{[1]}-\theta^*\|^2+\|\theta_{k-\tau}^{[2]}-\theta^\ast\|^2$ is too large, obtaining a contraction in $E\|\theta_k^{[1]}-\theta_k^{[2]}\|^2$ may not be feasible. In this case, the drift and contraction technique suggests using the drift of $E\|\theta_k-\theta^*\|^2$ in Proposition 4.2 to carefully balance the distance $E\|\theta_k^{[1]}-\theta_k^{[2]}\|^2\lesssim (E\|\theta_k^{[1]}-\theta_k^{[2]}\|^2)^r(E\|\theta_k^{[1]}-\theta^*\|^2+E\|\theta_k^{[2]}-\theta^*\|^2)^{1-r}$ and obtain a contraction.

**Numerical experiments:** We run SGD on $L_2$-regularized logistic regression with constant step sizes, without projection. Figure 1 confirms the Central Limit Theorem (CLT) result for averaged iterates. Figure 2(a) verifies the presence of an asymptotic bias approximately proportional to the stepsize $\alpha$, and illustrates the effectiveness of Richardson-Romberg (RR) extrapolation in reducing this bias. We also compare the bias under Markovian data ($x_{t+1}\sim P(\cdot|x_t)$) and iid data ($x_t\sim\pi$) in Figure 2(b). Interestingly, Figure 2(b) reveals that Markovian data does not necessarily lead to a larger bias than iid data. This is consistent with our theory, as the three bias terms $b_m,b_n,b_c$ may have opposite signs leading to cancellation. This result suggests that in the presence of nonlinearity, one should not avoid Markovian data simply for the sake of reducing bias. Rather, RR extrapolation may be more effective for bias reduction.


References:

[a] S. Allmeier and N. Gast. Computing the Bias of Constant-step Stochastic Approximation with Markovian Noise. 2024.

[b] A. Nemirovski, A. Juditsky, G. Lan, and A. Shapiro. Robust Stochastic Approximation Approach to Stochastic Programming. 2009.

[c] H. Kushner. Stochastic approximation: a survey. 2010.

[d] S. Lacoste-Julien, M. Schmidt, and F. Bach. A simpler approach to obtaining an O(1/t) convergence rate for the projected stochastic subgradient. 2012.

[e] S. Bubeck. Convex Optimization: Algorithms and Complexity. 2015.

[f] Q. Qin and J. P. Hobert. Geometric convergence bounds for Markov chains in Wasserstein distance based on generalized drift and contraction conditions. 2022.

[g] Y. Zhang, D. L. Huo, Y. Chen, Q. Xie. Prelimit Coupling and Steady-State Convergence of Constant-stepsize Nonsmooth Contractive SA. 2024.

---

### Decision · Program_Chairs · 2024-09-25

**Decision:**

Accept (spotlight)

**Comment:**

The reviewers were in unanimous agreement that the paper solves a challenging problem caused by Markovian data and nonlinear updates. Also the interplay between memory and nonlinearity in stochastic approximations is particularly striking and good for the community to know. A clear accept.